# Lysoptosis is an evolutionarily conserved cell death pathway moderated by intracellular serpins

Cliff J. Luke [1,2,10✉], Stephanie Markovina [2,3,10], Misty Good [1], Ira E. Wight[1], Brian J. Thomas [1], John M. Linneman[1], Wyatt E. Lanik [1], Olga Koroleva[1], Maggie R. Coffman[1], Mark T. Miedel[4], Qingqing Gong[1], Arlise Andress[3], Marlene Campos Guerrero [3], Songyan Wang[3], LiYun Chen [3], Wandy L. Beatty[5], Kelsey N. Hausmann[5], Frances V. White[6], James A. J. Fitzpatrick [7,8], Anthony Orvedahl [1], Stephen C. Pak [1] & Gary A. Silverman [1,2,7,9✉]

Lysosomal membrane permeabilization (LMP) and cathepsin release typifies lysosome-dependent cell death (LDCD). However, LMP occurs in most regulated cell death programs suggesting LDCD is not an independent cell death pathway, but is conscripted to facilitate the final cellular demise by other cell death routines. Previously, we demonstrated that *Caenorhabditis elegans* (*C. elegans*) null for a cysteine protease inhibitor, *srp-6*, undergo a specific LDCD pathway characterized by LMP and cathepsin-dependent cytoplasmic proteolysis. We designated this cell death routine, lysoptosis, to distinguish it from other pathways employing LMP. In this study, mouse and human epithelial cells lacking *srp-6* homologues, *mSerpinb3a* and *SERPINB3*, respectively, demonstrated a lysoptosis phenotype distinct from other cell death pathways. Like in *C. elegans*, this pathway depended on LMP and released cathepsins, predominantly cathepsin L. These studies suggested that lysoptosis is an evolutionarily-conserved eukaryotic LDCD that predominates in the absence of neutralizing endogenous inhibitors.

[1] Departments of Pediatrics, Washington University School of Medicine and the Children's Discovery Institute of St. Louis Children's Hospital, St. Louis, MO, USA. [2] Siteman Cancer Center, and Washington University School of Medicine and the Children's Discovery Institute of St. Louis Children's Hospital, St. Louis, MO, USA. [3] Radiation Oncology, Washington University School of Medicine and the Children's Discovery Institute of St. Louis Children's Hospital, St. Louis, MO, USA. [4] Department of Computational and Systems biology, Drug Discovery Institute, University of Pittsburgh, Pittsburgh, PA, USA. [5] Molecular Microbiology, Washington University School of Medicine and the Children's Discovery Institute of St. Louis Children's Hospital, St. Louis, MO, USA. [6] Department of Pathology and Immunology, Washington University School of Medicine and the Children's Discovery Institute of St. Louis Children's Hospital, St. Louis, MO, USA. [7] Cell Biology and Physiology, Washington University School of Medicine and the Children's Discovery Institute of St. Louis Children's Hospital, St. Louis, MO, USA. [8] Neuroscience, and Washington University School of Medicine and the Children's Discovery Institute of St. Louis Children's Hospital, St. Louis, MO, USA. [9] Genetics, Washington University School of Medicine and the Children's Discovery Institute of St. Louis Children's Hospital, St. Louis, MO, USA. [10] These authors contributed equally: Cliff J. Luke, Stephanie Markovina. ✉email: cjluke@wustl.edu; gsilverman@wustl.edu

Since Christian De Duve's discovery of lysosomes in 1955, it has been debated whether lysosomal membrane permeabilization (LMP) compromises cell survival, or heralds the onset of postmortem autolysis. Nonetheless, lysosome-dependent cell death (LDCD) is one of a dozen regulated cell death (RCD) subroutines identified by the Nomenclature Committee on Cell Death[1]. LDCD is defined by the presence of lysosomal membrane permeabilization (LMP) followed by the release of cathepsins into the cytosol[2]. While LDCD has a discernable role in a few physiological processes; its contribution to other pathological stressors is incompletely understood[3–6].

The role of lysosomes and LMP in mammalian cell death pathways has been difficult to ascertain. LMP and cathepsin release are detected in most cell death routines including apoptosis[1,7], mitochondrial permeability transition-driven necrosis (MPT-DN)[8,9], ferroptosis[10], pyroptosis[11], and necroptosis[12–14]. Moreover, morphology cannot always be used to determine the role of LMP in RCD[15], as cells dying by LDCD can exhibit both apoptotic[1,2,16,17] and necrotic morphological hallmarks[18–23]. Additionally, extracellular and intracellular stressors activate signaling pathways that are interconnected and molecular crosstalk simultaneously activates several pro-death and/or pro-survival pathways[1,2,24]. Therefore, molecular events contributing to LDCD could simultaneously trigger parallel death signaling cascades making it difficult to discern which pathway served as the principal executioner. Furthermore, the lysosomal cysteine proteases are highly processive with broad substrate specificity[25]. Cytosolic cathepsins can destroy evidence of pre-existing cell death routines by degrading signaling molecules associated with different forms of RCD[25–28]. The prevalence of these associations brings into question whether LDCD serves as a primary or stand-alone RCD pathway or is an epiphenomenon associated with the terminal stages of all cell deaths.

Although it has been difficult to establish the contribution of LDCD or LMP within the hierarchy of RCD pathways in mammalian systems[11,17,28–31], simpler model organisms have been instrumental in defining the core components of cell death routines[32–34]. In particular, C. elegans also undergoes several different forms of LDCD independent of caspase activation[35,36]. Neurons expressing a gain-of-function mutation in one of several different types of ion channels undergo a form of excitotoxic cell death with a necrotic phenotype[37–40]. In a different C. elegans model of LDCD, we discovered that animals null for the intracellular serine/cysteine protease inhibitor (serpin), srp-6, an inhibitor of calpains and lysosomal cysteine cathepsins, are sensitized to intestinal cell necrosis leading to organismal death[41]. Several exogenous stressors trigger a rapid rise in intracellular calcium, followed by calpain activation, LMP, cathepsin release, loss of plasma membrane integrity, and death within minutes to hours[41]. Since the apocalyptic loss of lysosomal membrane integrity is the focal point for somatic cell death in C. elegans, and this mechanism is the predominant manner of death occurring in response to multiple stressors[41–45], we have now re-defined this core death pathway as lysoptosis to distinguish it from events where LMP is a secondary event triggered by other death routines.

Lysoptosis in C. elegans represents an example of an autonomous LDCD pathway. Lysoptosis occurs independently of other known C. elegans cell death pathways[41] and is not confounded by the co-activation of necroptosis and pyroptosis, as the machinery of these death routines are lacking in nematodes[46,47]. Based on the molecular and morphological features of C. elegans lysoptosis, we determined whether the signature of this ancient LDCD pathway was conserved and/or embedded within mammalian cell death routines. Using lysoptosis sensitizing loss-of-function mutations in mouse and human homologs of srp-6 (mouse

Serpinb3a and human SERPINB3), we found that LDCD is conserved in mammals and can be activated by RCD inducers. Interestingly, low-level LMP could be detected in most forms of cell death, especially apoptosis. However, lysoptosis emerged as the dominant form of cell death when the intracellular protease-inhibitor balance was compromised.

## Results

**The srp-6 homologs, mouse Serpinb3a, and human SERPINB3, protected C. elegans from lysoptosis.** Nematodes null for srp-6 (srp-6(ok319)) are sensitized to lysoptosis due to the loss of cysteine peptidase inhibitory activity in the cytosol[41]. We generated transgenic lines to determine whether the expression of certain mammalian intracellular (Clade B) serpins could prevent death[48]. C. elegans expression vectors, containing a single cDNA from one of several different intracellular serpins ligated between the srp-6 native promoter and a C-terminal GFP gene (Fig. S1a), were co-injected with a visual selection head marker (P_{myo2}::mCherry) into srp-6(ok319) animals. Serpin expression was similar to the native srp-6 gene[41] and similar to each other (Fig. S1b–d). Transgenic lines were subjected to hypotonic stress and scored for death (Fig. S2)[41]. Human SERPINB3, SERPINB4, and the mouse orthologue, mSerpinb3a, but not SERPINB6 or SERPINB13, protected srp-6(ok319) animals from hypotonic stress (Fig. S2a)[49]. As controls, we included SERPINB1 (a serine peptidase inhibitor) and SERPINB3 with a classical inactivating P14 mutation (A341R)[50]. Neither of these serpins suppressed lysoptosis in C. elegans (Fig. S2a). This result indicated that the cysteine protease inhibitory activity of some, but not all, of the human intracellular Clade B serpins provided a pro-survival function similar to that of C. elegans SRP-6. These results also suggested that mSerpinb3a and SERPINB3, and to a lesser degree SERPINB4, could serve a pro-survival function in vertebrates. However, SERPINB4, a potent chymotrypsin serine protease inhibitor, which shares >90% amino acid sequence identity with SERPINB3, but diverges primarily within the C-terminal reactive site loop (RSL), demonstrated relatively poor inhibitory kinetics against lysosomal cysteine proteases in vitro[51,52].

Extensive LMP is the hallmark of lysoptosis in C. elegans. To obtain a qualitative and quantitative assessment of LMP after hypotonic stress, we labeled lysosomes in C. elegans with different molecular mass fluorescent dextrans (10–70 kDa)[53] and exposed the animals to hypotonic stress. srp-6(ok319) animals showed a significant graded and time-dependent lysosomal loss of all the dextrans (Fig. S2u, v), whereas N2 (wild-type) animals displayed almost no LMP, except the low 3 kDa molecular weight dextran was slightly but significantly decreased (Fig. S2t, v) compared to controls (Fig. S2r, s). These data suggested that there was discernible LMP in wild-type animals exposed to hypotonic stress, but lysosomal permeability was dramatically enhanced in srp-6(ok319) animals.

**Mouse Serpinb3a protected fetal intestinal epithelial cells from lysoptosis-like death.** mSerpinb3a is highly expressed in the lung, skin, and gastrointestinal tract of mice[54,55]. Therefore, we hypothesized that mSerpinb3a loss would predispose mouse intestinal cells to stress-induced death similar to that observed in our C. elegans model. To determine if this was the case, we established fetal intestinal explant cultures (FIECs)[56] (Fig. S3a) from mSerpinb3a^{+/+} (wild-type BALB/c) and mSerpinb3a^{−/−} mSerpinb3a knockout (mSerpinb3a^{−/−}) mice[57]. After culture for ≥7 days mSerpinb3a^{+/+}, FIECs expressed mSerpinb3a, as assessed by immunofluorescence and reverse-transcriptase PCR (Fig. S3b, c).

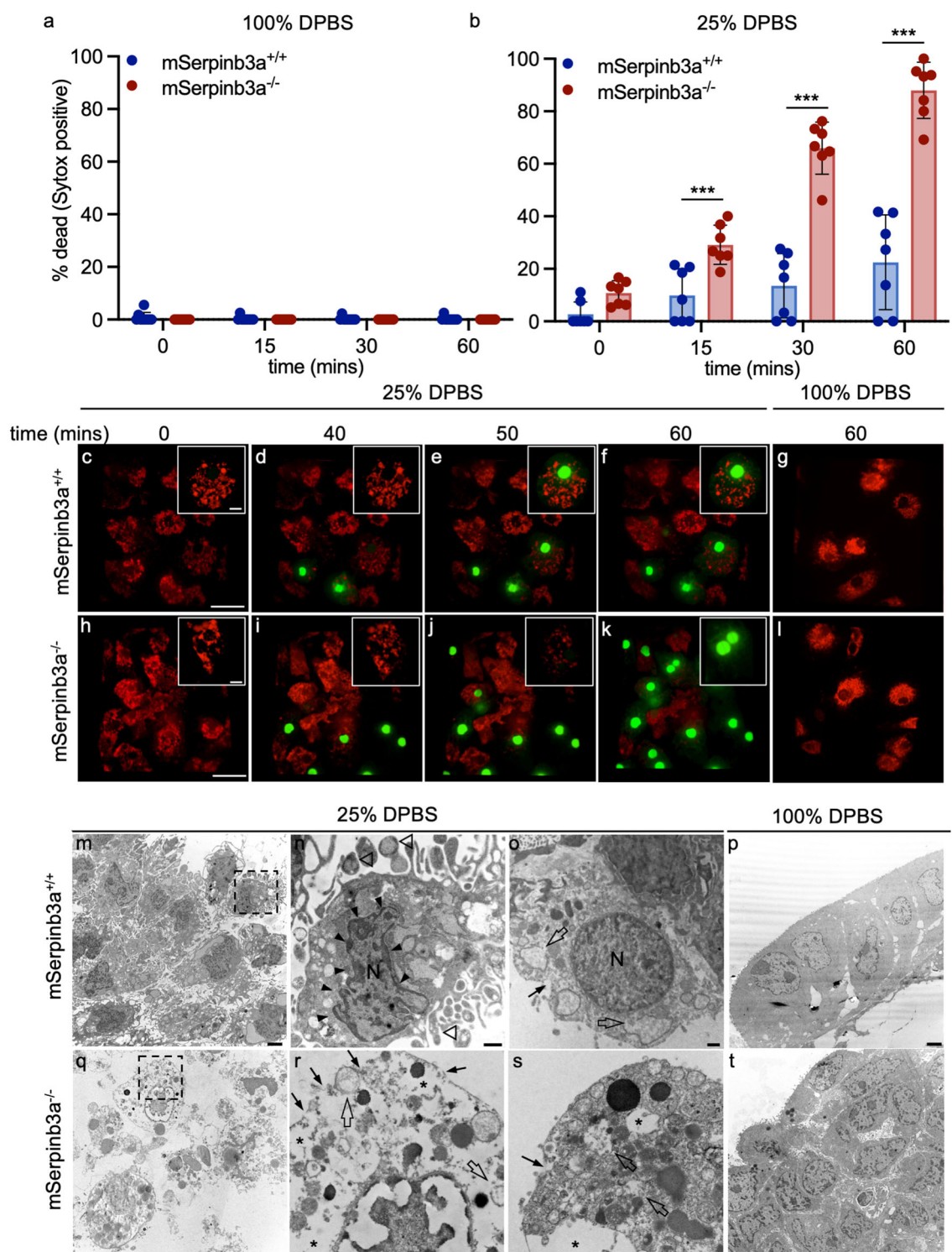

To determine if *mSerpinb3a*$^{-/-}$ FIECs were predisposed to lysoptosis-like cell death, we measured their susceptibility to hypotonic stress compared to wild-types. Neither *mSerpinb3a*$^{+/+}$ nor *mSerpinb3a*$^{-/-}$ FIECs showed Sytox™ Green (SG) uptake in 100% DPBS (Fig. 1a, g, l and Supplementary movies 1, 2). However, 30 min after incubation in 25% DPBS, ~10% of *mSerpinb3a*$^{+/+}$, and ~70% of *mSerpinb3a*$^{-/-}$ FIECs died (Fig. 1b). This mortality was more striking at 60 min, where ~20% of the controls, but ~90% of *mSerpinb3a*$^{-/-}$ FIECs were SG positive

(Fig. 1b). To determine whether extensive LMP occurred prior to cell death, as observed with *C. elegans* intestinal cells undergoing lysoptosis, FIECs were labeled with tetramethylrhodamine (TMR)-labeled 10 kDa dextran (red) and exposed to 25% DPBS in the presence of SG. Over time, a few of the *mSerpinb3a*$^{+/+}$ FIECs died during hypotonic stress exposure and there was a partial loss of lysosomal content during or after the uptake of SG (Fig. 1c–f and Supplementary Movie 3). In contrast, almost all of *mSerpinb3a*$^{-/-}$ FIECs died with rapid disappearance of

**Fig. 1 Mouse *Serpinb3a* protected fetal intestinal epithelial cells (FIECs) from lysoptosis-like death. a, b** *mSerpinb3a*[+/+] (blue bars with individual data points marked) or *mSerpinb3a*[−/−] (red bars with individual data points marked) FIECs were incubated in 100% (**a**) or 25% (**b**) DPBS for the indicated time points. The percent dead (Sytox positive) was calculated as (# Sytox positive nuclei/# of DAPI positive nuclei) × 100. The mean ± SD were from counts covering at least seven fields and compared using a two-tailed *t*-test (***$P < 0.001$). A representative experiment is shown. **c–l** Live-cell, confocal microscopy assessed lysosomal content (red, 10 K TMR-labeled dextran), and plasma membrane permeability (green, SG) in *mSerpinb3a*[+/+] (**c–g**) or *mSerpinb3a*[−/−] (**h–l**). FIECs incubated in 25% (**c–f**, **h–k**) or 100% DPBS (**g** and **l**). Z series were obtained every 2 min for 1 h, indicated time points are shown (scale bar = 25 µm); inset scale bar = 5 µm). Note marked loss of endolysosomal staining in SG positive *mSerpinb3a*[−/−] versus *mSerpinb3a*[+/+] FIECs (**k** vs. **f**). FIECs incubated in 100% DPBS did not show SG uptake or loss of TMR-labeled dextran (**g**, **l**). Representative transmission electron micrographs (TEM) of *mSerpinb3a*[+/+] (**m–p**) or *mSerpinb3a*[−/−] (**q–t**) FIECs after incubation in 25% DPBS (**m**, **n**, **o** and **q**, **r**, **s**) or 100% DPBS (**p** and **t**). Low magnification (scale bar = 2 µm) TEM images of *mSerpinb3a*[+/+] or *mSerpinb3a*[−/−] FIECs (**m**, **p** and **q**, **t**, respectively) show cell morphology. High magnification (scale bar = 100 nm) TEM images (**n**, **o** and **r**, **s**, respectively) demonstrate cell morphology. **n** and **r** were magnified cells from within the hashed box in **m** and **q**, respectively. **o** and **s** were individual cells from a different field of view. Some *mSerpinb3a*[+/+] FIECs exhibit hallmarks of apoptotic morphology (**n**), including chromatin condensation (black arrowheads) and plasma membrane budding (open arrowheads), or necrotic morphology (**o**), including plasma membrane breaks (black arrows) and organelle degeneration (open arrows). *mSerpinb3a*[−/−] FIECs displayed a necrotic cell death morphology (black and open arrows as above) along with severe vacuolization and cytoplasmic clearing (asterisks) (**r**, **s**). Original imaging data files can be found at https://data.mendeley.com/datasets/hk2t9x7d6x/1.

endolysosomal staining prior to SG uptake (Fig. 1h–k and Supplementary Movie 4).

Transmission electron microscopy (TEM) ultrastructural changes of FIECs incubated in 25% DPBS for 1 h indicated more cell death in *mSerpinb3a*[−/−] FIECs. An occasional dead cell from *mSerpinb3a*[+/+] FIECs showed apoptosis with membrane budding and chromatin condensation (Fig. 1m, n), or necrosis with cytoplasmic vacuolization and loss of plasma membrane integrity (Fig. 1o)[15]. In contrast, *mSerpinb3a*[−/−] FIECs exclusively showed a necrotic-like phenotype consisting of nucleocytoplasmic and organellar swelling, vacuolization, and loss of lysosomal and plasma membrane integrity (Fig. 1q–s)[15]. FIECs from both strains treated with 100% DPBS appeared normal (Fig. 1p, t). We concluded that *mSerpinb3a*[−/−] FIECs were sensitized to lysoptosis-like cell death similar to that observed in *srp-6(ok319) C. elegans* intestinal cells exposed to hypotonic stress[41].

**Lysoptosis-like death in *mSerpinb3a*[−/−] FIECs required LMP and lysosomal cysteine peptidases, but not executioner caspases.** In *srp-6(ok319) C. elegans*, lysoptosis occurred in the absence of caspases, but required LMP and the cytoplasmic release of lysosomal cysteine peptidases[41]. To determine if *mSerpinb3a*[−/−] FIECs exposed to hypotonic stress were undergoing apoptosis or lysoptosis, we utilized fluorescent markers of lysosomal injury and different protease inhibitors. FIECs were incubated with the endolysosomal marker, Alexafluor[647] conjugated to 10 kDa dextran (dextran[647]), and with either the pan-cysteine protease inhibitor, E64d, or the executioner caspase-3/7 inhibitor, DEVD-CHO. To determine whether hypotonic stress might also activate apoptosis, cultures were assayed for evidence of apoptosis with the phosphatidylserine marker, FITC-labeled Annexin V (green), and the necrosis marker, propidium iodide (PI, red). Minimal cell death (<10%) was detected in the *mSerpinb3a*[+/+] FIECs incubated in 25% DPBS (Fig. 2a–l, y). Neither DMSO (diluent), E64d, nor DEVD-CHO had an effect on cell viability (Fig. 2c, y). Lysosomal staining also remained intact (Fig. 2d, h, l, z). In contrast, *mSerpinb3a*[−/−] FIECs showed increased PI uptake, but not annexin V staining, upon exposure to 25% DPBS (Fig. 2m-x,y). Cell death was associated with the complete loss of endolysosomal staining (dextran[647], Fig. 2p, z) and was mostly blocked by pre-incubation with E64d, but not DEVD-CHO (Fig. 2t, x, z). These data suggested that the cysteine proteases triggered an increase in both lysosomal and plasma membrane permeability in *mSerpinb3a*[−/−] FIECs exposed to hypotonic stress. Moreover, neither caspase activation nor plasma membrane externalization of phosphatidylserine (markers of apoptosis) were associated with this lysoptosis-like phenotype.

**Necrostatin-1 fails to inhibit lysoptosis-like death in *mSerpinb3a*[−/−] FIECs.** To determine whether *mSerpinb3a*[+/+] or *mSerpinb3a*[−/−] FIECs exposed to hypotonic stress were undergoing necroptosis, cells were treated with nec-1 prior to incubation in 25% DPBS (Fig. 2i–xiv). We found ~20% of *mSerpinb3a*[+/+] FIECs died, as defined by PI uptake (Fig. 2i–vi, xiii). Nec-1 provided partial protection in this experiment, although overall death was minimal (Fig. 2xii). However, there was no decrease in the lysosomal content between the nec-1 treated or untreated *mSerpinb3a*[+/+] FIECs (Fig. 2iii, vi, xiv). This lack of effect may be attributed to the relatively late-appearing LMP associated with necroptosis[13]. In contrast, ~60% *mSerpinb3a*[−/−] FIECs died (Fig. 2vii–xiii). Nec-1 neither protected these cells from death, nor blocked the marked LMP associated with PI uptake (Fig. 2vii, ix, xi–xiv).

**Apoptosis and necroptosis inducers preferentially triggered lysoptosis-like cell death in *mSerpinb3a*[−/−] FIECs.** Since DEVD-CHO failed to block lysoptosis-like cell death in *mSerpinb3a*[−/−] FIECs, we assessed whether this inhibitor was ineffective in FIECs, or the executioner phase of apoptosis was nonoperational under these conditions. We induced apoptosis and necroptosis in *mSerpinb3a*[+/+] and *mSerpinb3a*[−/−] FIECs by treating the cultures with the protein kinase C inhibitor, STS and STS combined with the pan-caspase inhibitor, z-VAD-fmk, respectively[58,59]. After treatment with STS, *mSerpinb3a*[+/+] FIECs showed increased annexin V staining without PI uptake (Fig. 3a–c, g–i, m–o, s, t). These results are consistent with the induction of apoptosis in *mSerpinb3a*[+/+] FIECs. Surprisingly, the *mSerpinb3a*[−/−] FIECs showed little annexin V staining in response to STS, but a significant increase in PI uptake (Fig. 3d–f, j–l, s, t). Moreover, the PI uptake in *mSerpinb3a*[−/−] FIECs was comparable to that observed in FIECs incubated in 25% DPBS for 2 h (Fig. 3p–r, t). To ensure that annexin V staining had not escaped detection due to accelerated necrosis, we performed live-cell time-lapsed imaging of *mSerpinb3a*[−/−] FIECs during incubation with STS (Fig. S4). Minimal annexin V staining was detected, although PI-positive cells were detected as early as 2 h after treatment. In contrast, *mSerpinb3a*[+/+] FIECs were annexin V positive as early as 2 h with a few PI-positive cells detected by 8 h (Fig. S4).

*mSerpinb3a*[+/+] FIECs exposed to STS and z-VAD-fmk showed increased necrotic-like cell death as shown by PI uptake, and death was inhibited markedly by the RIPK1 inhibitor, nec-1[60] (Fig. 3u, w, y). *mSerpinb3a*[−/−] FIECs treated with STS and z-VAD-fmk also showed significant necrotic-like cell death as evident by PI uptake (Fig. 3v, x, y). Unlike *mSerpinb3a*[+/+] FIECs, *mSerpinb3a*[−/−] FIECs were not protected from cell death by nec-1 (Fig. 3w, x, y). We concluded that *mSerpinb3a*[+/+] FIECs were capable of undergoing apoptosis and necroptosis, but that

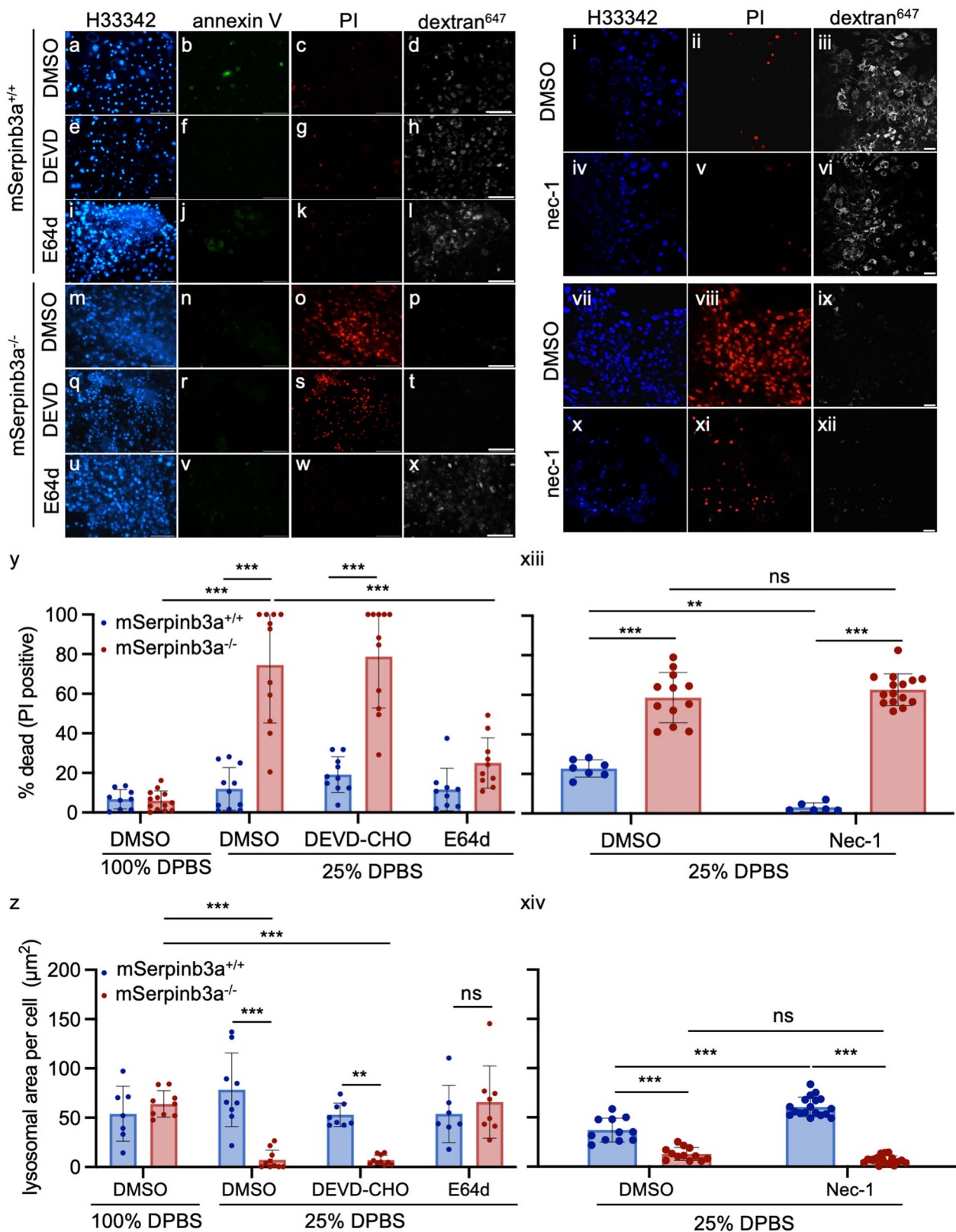

*mSerpinb3a*$^{−/−}$ FIECs preferentially demonstrated a lysoptosis-like death phenotype in response to the same cell death inducers. Moreover, *mSerpinb3a*$^{−/−}$ FIECs were more sensitive to classic apoptosis- and necrosis-inducing agents as evidenced by higher cell death under equivalent conditions.

To determine whether apoptosis or necroptosis contributed to cell death in *mSerpinb3a*$^{−/−}$ FIECs, or if a lysoptosis-like pathway was executed before either RCD routine became fully activated, required additional investigation. Unfortunately, biochemical studies in FIECs are impractical due to their small colony sizes and fastidious culture conditions. To overcome this limitation, we re-assessed these RCD routines in human immortalized tumor cell lines that expressed high levels of SERPINB3 (human mSerpinb3a orthologue). Two different human cervical carcinoma cell lines, "wild-type" (SW756$^{B3-WT}$ and

**Fig. 2 LMP and lysosomal cysteine peptidases, but not executioner caspases or RIPK1, induced lysoptosis-like death in *mSerpinb3a*[−/−] FIECs.**
**a–x** Representative confocal fluorescence images of *mSerpinb3a*[+/+] (a-l) or *mSerpinb3a*[−/−] (**m-x**) FIECs pretreated with diluent (DMSO; **a-d**; **m-p**), 2 μM DEVD-CHO (**e-h**; **q-t**) or 2 μM E64d (**i-l**; **u-x**) and then incubated in 25% DPBS for 1 h (scale bar = 40 μm). FIECs were stained with Hoescht 33342 (H33342; blue), annexin V-FITC (annexin V, green), propidium iodide (PI, red), and Alexafluor[647] conjugated dextran (dextran[647], white). **y, z** Quantification of dead cells (PI-positive; **y**) or lysosomal area (dextran[647] staining; **z**) from different representative experiments (*n* = 3) over multiple fields (*n* > 5). **i–xii** Confocal maximum intensity projections of fluorescently labeled lysosomes from *mSerpinb3a*[+/+] (i–vi) or *mSerpinb3a*[−/−] (vii–xii) FIECs (scale bar = 25 μm). Cultures were pretreated with DMSO (**i-iii**; **vii-ix**) or nec-1 (**iv-vi**; **x-xii**) for 1 h prior to incubation in 25% DPBS. Cell numbers, cell death, and lysosomes were quantitated using H33342 (blue), PI (red), and dextran[647] (white), respectively. Representative images are from the same experiment, which was repeated three times. **xii, xiv** Quantification of PI positive (**xii**) or lysosomal (**xiv**) staining from multiple fields (*n* ≥ 6). The means ± SD of a representative experiment were compared using a two-way ANOVA with Tukey's multiple comparisons (**$P < 0.01$, ***$P < 0.001$). Original data files can be found at https://data.mendeley.com/datasets/scgbb3s333/1.

---

HT3[B3-WT]) or null (SW756[B3-KO] and HT3[B3-KO]) for SERPINB3, were generated using CRISPR/Cas9 technology (Fig. S5)[61].

**SERPINB3 null human tumor cell lines exposed to hypotonic stress undergo lysoptosis-like cell death.** HT3[B3-WT] and HT3[B3-KO] cell lines were incubated in 10% DPBS and examined for cell death by PI uptake. HT3[B3-KO] cells showed >50% mortality compared to the HT3[B3-WT] controls (Fig. 4a, b). Although the HT3[B3-WT] cell death was relatively small and variable (<20%), it was still detectable (compare Figs. 4a, b). This suggested that both cell lines were susceptible to hypotonic stress, but as in *C. elegans* and mouse FIECs, serpin loss exacerbated the cell death phenotype. Similar results were obtained with the SW756[B3-WT] and SW756[B3-KO] cell lines (Fig. S6). Moreover, the expression of SERPINB3, but not SERPINB3[A341R 50], nor, in this case, SERPINB4 rescued the HT3[B3-KO] cells from hypotonic stress (Fig. S7). Other mechanistically different cysteine protease inhibitors of various efficacy; including cystatin A, cystatin B[62], SERPINB1[63], SERPINB6[64], and SERPINB13[65], were detected by quantitative RT-PCR in both tumor cell lines (Fig. S8a). However, only SERPINB1 was detectable in all the cells lines by immunoblotting (Fig. S8b). Considering the inability of SERPINB1 to rescue *srp-6* null *C. elegans* (Fig. S2), these rescue data suggested that the loss of SERPINB3 alone was responsible for enhanced sensitivity to hypotonic stress. Cell death was blocked in the HT3[B3-KO] cells by pre-incubation with E64d, but not DEVD-CHO (Fig. 4a, b), and characterized by a nearly complete loss of endolysosomal staining (Fig. 4c–n). Similar to *srp-6(ok319)* animals, LMP assessed by the loss of fluorescently labeled lysosomal dextrans, ranging from 3–70 kDa, showed a significant acceleration in HT3[B3-KO] cells compared to HT3[B3-WT] cells (Figs. 4o–s, S9). LMP can be induced by ROS[66], and a mouse serpin, Spi2A, protects mouse embryonic fibroblasts from cell death by inhibition of cathepsin B (CTSB)[67]. However, after treatment of either HT3[B3-WT] and HT3[B3-KO] cells with hypotonic stress, no significant increase in intracellular ROS was detected by the cell-permeant fluorescent ROS indicator, 2',7'-dichlorodihydro-fluorescein diacetate (H2-DCFDA; Fig. S10a). As a control, treatment with the ROS producer, tBOOH, increased H2-DCFDA fluorescence intensity in both cell lines (Fig. S10a). tBOOH-induced fluorescence exceeded that observed after hypotonic DPBS exposure, although no death was detected after 2% tBOOH treatment. Thus, the marginal (statistically non-significant) increase in ROS detected after hypotonic DPBS exposure did not, in itself, appear to be toxic to either the HT3[B3-WT] or HT3[B3-KO] cells (Fig. S10b). Upon exposure to 15% DPBS, and compared to HT3[B3-WT] cells, HT3[B3-KO] cells examined by TEM showed cytoplasmic and nuclear swelling, loss of organellar architecture, and degeneration of the plasma membrane (Figs. 4t–x and S11). HT3[B3-WT] cells showed similar morphology, but the damage was neither as severe nor as frequent compared to the HT3[B3-KO] cells (Figs. 4t–x and S11). We concluded that this cell death phenotype resembled the lysoptosis-like cell death observed in *mSerpinb3a*[−/−] FIECs exposed to hypotonic stress.

**The lysoptosis-like cell death pathway involves promiscuous cytoplasmic proteolytic activity in human cell lines.** During lysoptosis in *C. elegans*, we detected cytosolic cysteine protease activity using the quenched fluorogenic substrate zFR-R110, which fluoresces green upon cleavage by proteolytic enzymes[41]. To determine whether lysosomal proteases released into the cytosol were involved in the lysoptosis-like cell death pathway, we first determined which lysosomal proteases (including the most common cysteine proteases, CTSL and CTSB; and the cysteine protease processing aspartic protease, CTSD) were detectable in parental HT3 and SW756 cell lines using quantitative RT-PCR and immunoblotting (Fig. S12). We detected CTSB, -D, and -L transcripts and protein in both cell types regardless of SERPINB3 status (Fig. S12a–e, h, i). CTSS, -K, or -V transcripts were undetectable by quantitative RT-PCR (Fig. S12a). Neither CTSS nor -K was detected by immunoblotting (Fig. S12–n-u). A scant amount of CTSV protein was present in the SW756 cell line (Fig. S12j, k). These data suggested that CTSL and -B were the major cysteine CTSs expressed in the HT3 and SW756 cell lines. To monitor the release of cysteine CTS activity into the cytosol, we utilized a differential cell lysis procedure that facilitates the diffusion of cytosolic contents into the media without affecting lysosomal membrane integrity[68,69]. In HT3 cells, low concentration digitonin (<25 μg/ml) permeabilized the high cholesterol-containing plasma membrane, whereas higher concentrations (>25 μg/ml) also perforated subcellular vesicle membranes, including the lysosomes (Fig. S13a, b). As expected, cytosolic GAPDH was detected in all culture supernatants obtained from both HT3[B3-WT] and HT3[B3-KO] cells, regardless if they were treated with either 10 or 100% DPBS and exposed to either low (20 μg/ml) or high (200 μg/ml) digitonin concentrations. Similarly, the cytosolic protein, SERPINB3, was also detected in these same supernatants, except not those from HT3[B3-KO] cells. As a marker for LMP, we analyzed CTSL content by immunoblotting. Unsurprisingly, CTSL was detected in supernatants obtained after 200 μg/ml digitonin treatment in both HT3[B3-WT] and HT3[B3-KO] cells incubated in 100% DPBS (Fig. 5a). In contrast, incubation in 10% DPBS, but not 100% DPBS, and the low digitonin concentration, revealed the presence of CTSL in the supernatant from both HT3[B3-WT] and HT3[B3-KO] cells (Fig. 5a). To assess for actual leaked cytosolic protease activity, we assayed for cleavage of the quenched fluorogenic cysteine protease substrate, zFR-R110. There was relatively more activity detected in supernatants from HT3[B3-KO] than HT3[B3-WT] cells treated with 10% DPBS and the low concentration of digitonin (Fig. 5b). Taken together, we concluded that cellular stress-induced LMP and the release of active lysosomal cysteine proteases into the cytosol of both HT3[B3-WT] and HT3[B3-KO] cell lines. However, it was the absence of SERPINB3 neutralizing activity that led to unchecked lysosomal derived cytosolic proteolysis, followed by subcellular injury and, eventually, the lysoptosis-like cell death phenotype.

In *C. elegans*, the lysosomal proteases, ASP-1, CPR-6, and CPL-1 (human CTSD, CTSB, and CTSL homologs, respectively) are necessary for lysoptosis[41] and all three human homologs were

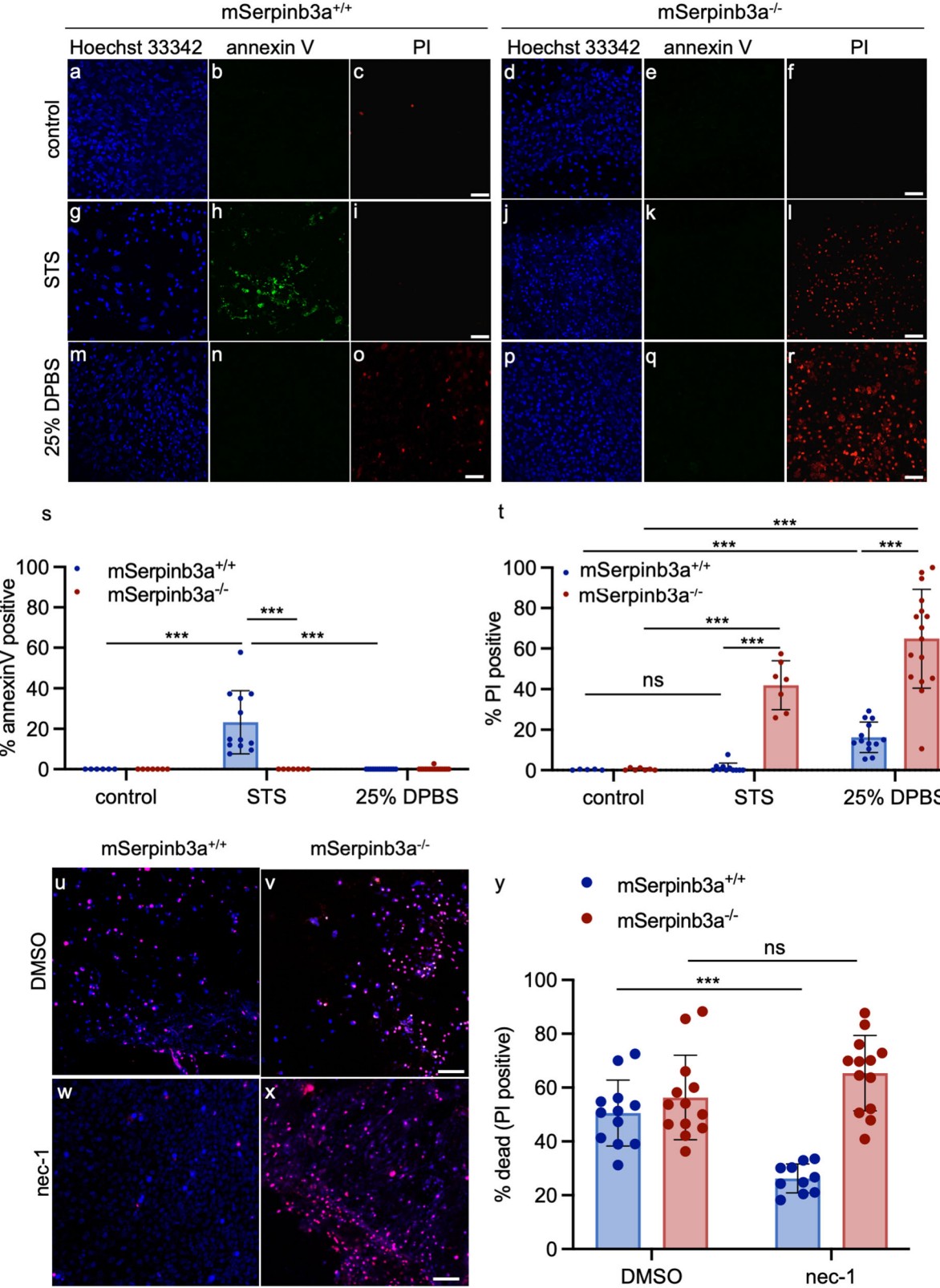

present in the SW756 cell lines (Fig. S12). However, CTSD is an aspartic protease family member and both human SERPINB3 and mouse Serpinb3a failed to inhibit CTSB in vitro[52,70]. Thus, we reasoned that the CTSL protease would be the most likely target for SERPINB3 and be required for lysoptosis-like cell death in human cells. We generated CTSL knockout cell cultures in both SW756[B3-WT] and SW756[B3-KO] cells using CRISPR/cas9 technology (Table S1). As

expected, the SW756[B3-KO;CTSL-WT] cell lines were more sensitive to 10% DPBS treatment than SW756[B3-WT;CTSL-WT] (Fig. 5c). However, the SW756[B3-KO;CTSL-KO] cell line significantly protected the SW756[B3-KO] cells from death (Fig. 5c). Moreover, pretreatment of the HT3[B3-KO] cell line with the selective cathepsin L inhibitor, CAA0225[71], for 1 h at 10 μM prior to treatment with hypotonic stress protected the cells from death, similar to the E64d control at the same

**Fig. 3 Apoptosis or necroptosis inducers preferentially triggered lysoptosis-like cell death in _mSerpinb3a_[−/−] FIECs. a–r** To assess apoptosis, mSerpinb3a[+/+] or mSerpinb3a[−/−] FIECs were treated with 100% PBS (**a–f**), 1 μM staurosporine (STS) for 16 h (**g–l**), or 25% PBS for 2 h (**m–r**). Representative confocal fluorescence maximum intensity projections were of cells stained with Hoechst 33342 (blue), annexin V-FITC (green), and PI (red; scale bars = 25 μm). **s–t** Quantification of annexin V (**s**) and PI (**t**) staining in multiple fields (n ≥ 5) from a representative of three experiments using mSerpinb3a[+/+] (blue) or mSerpinb3a[−/−] (red) FIECs treated with 100% PBS (control), 1 μM STS for 16 h or 25% PBS for 2 h. **u–x** To induce necroptosis, mSerpinb3a[+/+] (**u**, **w**), or mSerpinb3a[−/−] (**v**, **x**) FIECs were pretreated with either DMSO or 5 μM nec-1 and then incubated with 1 μM STS with 10 μM z-VAD-fmk for 16 h. Representative merged confocal fluorescence maximum intensity projections were of cells stained with H33342 (blue) and PI (red, scale bar = 25 μm). Dually labeled nuclei indicated dead cells and were depicted as magenta. **y** Quantification of PI staining in multiple fields from a representative of three experiments of mSerpinb3a[+/+] or mSerpinb3a[−/−] FIECS pretreated with DMSO or 5 μM nec-1 and induced for necroptosis by treatment with 10 μM STS with 1 μM z-VAD-fmk for 16 h. For data in **s**, **t**, and **y**, the means ± SD were compared using a two-tailed t-test (***P < 0.001, **P < 0.01). Original data files can be found https://data.mendeley.com/datasets/4gfrf8xkyf/1.

---

concentration (Fig. 5d). Similar to _C. elegans_[41], these data suggested that CTSL, a major target of SERPINB3 in vitro, was an essential component of the mammalian lysoptosis-like cell death pathway.

**Induction of apoptosis in SERPINB3 null tumor cells activated executioner caspases, but a lysoptosis-like cell death phenotype.** Since _mSerpinb3a_[−/−] FIECs treated with STS demonstrated a cell death phenotype more consistent with lysoptosis, rather than apoptosis, we treated both HT3[B3-KO] and HT3[B3-WT] cells with STS and assayed for cell death. Both cell lines showed a comparable increase in caspase-3/7 activation, as measured by an increase in fluorescence of a quenched executioner caspase substrate (NucView) (Fig. 6a). This activity was blocked by the caspase inhibitor, DEVD-CHO, but not the pan-cysteine peptidase inhibitor, E64d (Fig. 6a). Consistent with induction of apoptosis by STS, caspase-3 and PARP cleavage were detected in both cell lines (Fig. 6b). HT3[B3-WT] cells also showed annexin V staining (Fig. 6c–e, i–k, o) without an increase in PI or SG uptake (Fig. 6c–e, i–k, p). In contrast, HT3[B3-KO] cells showed minimal annexin V staining, but a marked increase in PI or SG uptake (Fig. 6f–h, l–n, p, q). This observation indicated an early loss of plasma membrane integrity, which is atypical for the early stages of apoptosis. Of note, SG uptake in HT3[B3-KO] cells was decreased after pre-incubation with E64d, but not DEVD-CHO (Fig. 6q).

We next examined lysosomal membrane integrity in STS treated cells. HT3[B3-WT] cells showed a modest but significant decrease in Lysotracker[TM] staining, but no increase in SG uptake (Fig. S14). In contrast, HT3[B3-KO] cells showed nearly a complete loss of Lysotracker[TM] staining concomitant with increased SG uptake (Fig. S14). We next assessed LMP by the loss of fluorescently labeled lysosomal dextrans. Interestingly, both cell lines showed a rapid and comparable loss of the 3 kDa dextran (Fig. 6r–v). However, only the HT3[B3-KO] cells showed a significant loss of the higher molecular mass forms, reminiscent of HT3[B3-KO] cells exposed to hypotonic stress (Fig. 6r–v compared to Fig. 4o–s). TEMs of STS treated cells, showed typical morphological features of apoptosis in HT3[B3-WT] cells, with membrane budding, and chromatin condensation and fragmentation (Fig. 6w, y). In contrast, HT3[B3-KO] cells showed an obvious necrotic-like phenotype with swollen nuclei, cytoplasmic vacuolization, and loss of plasma membrane integrity (Fig. 6x, z). Taken together, these studies suggested that STS induced canonical caspase-3/7-dependent apoptotic death in HT3[B3-WT] cells with minimal LMP. In HT3[B3-KO] cells exposed to STS, caspase-3/7 was activated. However, the lack of protection by DEVD-CHO and necrotic morphology suggested that the intrinsic apoptosis signaling by STS led to a subtle perturbation of lysosomal membrane integrity (loss of 3 kDa dextran), which rapidly deteriorated in the absence of SERPINB3. The net result was the emergence of a lysoptosis-like cell death phenotype in HT3[B3-KO] cells.

**Lysoptosis-like cell death is a distinct form of regulated necrosis.** Mitochondrial permeability transition-driven necrosis (MPT-DN), necroptosis, ferroptosis, and pyroptosis are four distinct, well-characterized regulated necrosis pathways that are differentiated by their mechanisms of induction, signaling cascades, pharmacologic or molecular inhibitors, and death effectors[1,2]. Since lysosomal membrane instability has been associated with all death routines[11], we determined whether the loss of SERPINB3 predisposed cells to lysoptosis-like cell death or sensitized them to one or more regulated necrosis pathways.

**MPT-DN.** The necrotic-like cell death pathway, MPT-DN, results from the opening of the calcium-gated mitochondrial permeability transition pore complex (PTPC) which is inhibited by Cyclosporin A (CsA)[72]. The role of LMP in MPT-DN is ill-defined, however, the neutralization of lysosomal activity by chloroquine can block MPT-DN[73]. To determine the role of LMP in MPT-DN, we treated HT3[B3-KO] and HT3[B3-WT] cells with the MPT-DN inducer, hydrogen peroxide ($H_2O_2$). To ensure that $H_2O_2$ induced MPT, we used the fluorescence probe, JC-1 which fluoresces red in healthy mitochondria but, upon depolarization of the mitochondrial membrane, allows the escape of green fluorescent JC-1 monomers into the cytoplasm[74]. After treatment with $H_2O_2$, both cell lines showed comparable increases in MPT indicated by increased JC-1 green fluorescence (Fig. 7a–n). However, the degree of cell death, as indicated by SG uptake, was significantly increased in HT3[B3-KO] cells compared to HT3[B3-WT] cells (Fig. 7o). CsA protected HT3[B3-WT] cells from $H_2O_2$ toxicity, but had no effect on HT3[B3-KO] cells death (Fig. 7p). In contrast, E64d failed to protect significantly HT3[B3-WT], whereas cell death was significantly decreased in HT3[B3-KO] cells (Fig. 7p). Since lysosomal integrity by Lysotracker staining was more compromised in HT3[B3-KO] cells as compared to HT3[B3-WT] cells treated with $H_2O_2$ (Fig. S15), we assessed LMP using fluorescently labeled dextrans. HT3[B3-WT] cells showed appreciable release of only 3 kDa dextran, whereas HT3[B3-KO] cells demonstrated a significant decay of the 3–70 kDa dextrans (Fig. 7q–u). These data suggested that LMP was not a dominant feature in the HT3[B3-WT] cells undergoing MPT-DN. TEMs showed evidence of necrosis with pan-cellular swelling and vacuolization in both cell lines, but was more pronounced in HT3[B3-KO] cells (Fig. 7v–z). Comparable mitochondrial degeneration was observed in both cells (Fig. 7v–y, aa). We concluded that $H_2O_2$ triggered upstream events in the MPT-DN pathway (i.e., mitochondrial injury) in both the HT3[B3-WT] and HT3[B3-KO] cell lines. However, in the HT3[B3-KO] cells, the loss of SERPINB3 enhanced the effects of LMP resulting in a lysoptosis phenotype that killed the cells. Lysoptosis likely interrupted the MPT-DN pathway before it was fully activated and explains why E64, but not CsA, was partially protective in HT3[B3-KO] cells.

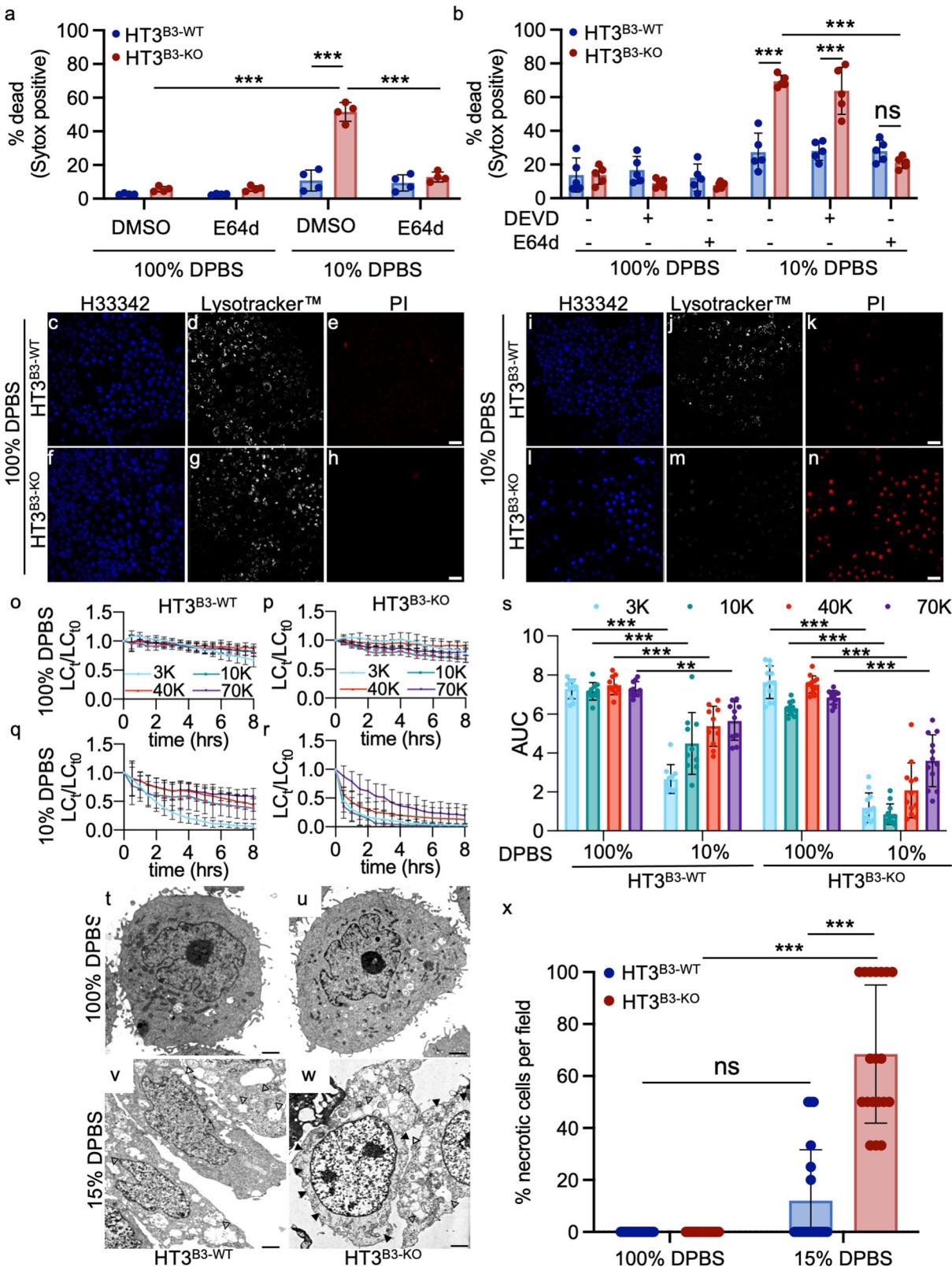

**Ferroptosis.** Reactive oxygen species (ROS) generated by the iron-dependent Fenton reaction, depletion of glutathione, and lipid peroxidation result in ferroptosis. Morphologically ferroptosis is distinguished by mitochondrial condensation, increased membrane density and lost cristae[75]. Ferroptosis is triggered by either erastin or RSL3[76,77] and is inhibited by ferrostatin-1 (fer-1)[78]. Although evidence of LMP has been described[10,79], lysosomal iron stores and proteases appear to be dispensable for ferroptosis[75]. To determine the role, if any, of SERPINB3 in ferroptosis, SW576[B3-WT] and SW576[B3-KO] cells were treated with erastin[77] and the fluorescent

**Fig. 4 Hypotonic stress induces lysoptosis-like death in human tumor cell lines null for SERPINB3. a, b** HT3[B3-WT] (blue) or HT3[B3-KO] (red) cells were incubated with DMSO, 2 μM E64d (**a, b**), or 10 μM DEVD-CHO (DEVD; **b**) for 1 h prior to exposure to 100% or 10% DPBS for 4 h, and stained with SG and H33342. Percent dead was calculated as (# Sytox positive nuclei/# of blue nuclei) × 100. **c–n** Representative confocal images (scale bar = 25 μm) of HT3[B3-WT] or HT3[B3-KO] cell lines treated with 100% DPBS or 10% DPBS for 4 h and then stained with H33342 (blue; **c, f, i, l**), Lysotracker™ deep red (white; **d, g, j, m**), and PI (red; **e, h, k, n**). **o–r** To quantify LMP, HT3[B3-WT] (**o, q**), or HT3[B3-KO] (**p, r**) cells were incubated with 3 kDa Cascade blue, 10 kDa Alexa488, 40 kDa TMR, and 70 kDa Texas red conjugated dextrans prior to exposure to 100 or 10% DPBS and imaged using live-cell resonance scanning confocal microscopy (≥10 fields, ≥20 z-planes). The number of lysosomes at each time point (LC$_t$) were normalized to the initial lysosome count at time zero (LC$_{t0}$). **s** Area under the curve (AUC) for each dextran over time for both HT3[B3-WT] (left) or HT3[B3-KO] (right) cells treated with 100 or 10% PBS. Statistical significance was determined using a two-way ANOVA with Tukey's multiple comparisons test. **t–w** Representative TEM images (scale bars = 500 nm) of HT3[B3-WT] or HT3[B3-KO] cells treated with 100 or 15% DPBS for 4 h. Vacuolization (closed arrowheads) was noted in the HT3[B3-KO], and to a lesser extent in the HT3[B3-WT] cells, treated with 15% DPBS. **x** Blinded scoring of the percentage of necrotic appearing cells in 2000x magnification TEM images (≥20 fields) of HT3[B3-WT] or HT3[B3-KO] cells treated with 100 or 15% DPBS for 4 h. Unless otherwise noted, a representative of ≥3 replicates is shown, and the means ± SD were compared using a two-way ANOVA with Tukeys' multiple comparisons (ns not significant, ***P < 0.001, **P < 0.01). Original data files can be found at https://data.mendeley.com/datasets/gnwb39t76k/1.

ROS indicator, H2-DCFDA. Both cell lines showed a comparable increase in fluorescence that peaked ~10–12 h after erastin treatment (Fig. 8a–e). As indicated by Sytox™ Orange uptake, the degree of erastin-induced death in the SW756[B3-KO] and SW756[B3-WT] cell lines was similar (Fig. 8f). Death was blocked completely by fer-1, but not by E64d (Fig. 8g). While both cell lines similarly showed a slight decrease in LysoTracker® Red staining with erastin treatment (Fig. 8h–l), minimal LMP was detected, with partial loss of both the 3 and 10 kDa dextran markers in only the SW756[B3-KO] cells (Fig. 8m–q). Both cell lines showed morphological features of ferroptosis, with shrunken mitochondria, dense membranes and loss of cristae (Fig. 8r–w)[75]. We concluded that the ferroptosis RCD pathway neither triggered significant LMP nor activated the lysoptosis-like pathway even in the absence of SERPINB3.

**Necroptosis.** Necroptosis is executed via its core machinery; receptor-interacting serine/threonine-protein kinase 1 (RIPK1), RIPK3, and mixed-lineage kinase domain-like pseudokinase (MLKL)[80]. While necroptosis is thought to be an executioner protease-independent pathway[81], in macrophages, lysosomal cysteine peptidases can cleave RIPK1 and impair necroptosis signaling[82]. Given our findings in FIECs, we determined whether LMP and SERPINB3 loss altered necroptosis, induced by TNFα, a SMAC mimetic (BV6), a pan-caspase inhibitor (qVD-Oph), and cycloheximide (T + S + Q + C). HT3[B3-KO] cells demonstrated more cell death than HT3[B3-WT] cells (Fig. 9a, r). Nec-1 completely blocked cell death in the HT3[B3-WT] cells, but only partially in HT3[B3-KO] cells (Fig. 9b). In contrast, treatment of E64d had no effect on the death of HT3[B3-WT] cells, whereas it inhibited death in HT3[B3-KO] cells to a greater extent than that of nec-1 (Fig. 9b). Immunoblotting with anti-phospho-RIPK1 demonstrated that necroptosis was activated by T + S + Q + C in both cell lines (Fig. 9c). Similarly, immunofluorescence staining with anti-phospho-MLKL showed that MLKL was phosphorylated in HT3[B3-WT] and HT29 positive control cells after necroptosis induction, but very little staining was observed in HT3[B3-KO] cells (Fig. S16). Staining with LysoTracker® Deep Red, indicated that the endolysosomal compartment was compromised more extensively in HT3[B3-KO] cells than HT3[B3-WT] cells (Fig. 9d–h). In addition, LMP was increased in HT3[B3-KO] compared to HT3[B3-WT] cells, demonstrated by the significant lysosomal leakage of the 3–70 kDa dextrans (Fig. 9i–m). By TEM, HT3[B3-WT] cells were intact with some heterochromatin clumping (Fig. 9n, o). In contrast, more HT3[B3-KO] cells were swollen with cytoplasmic vacuolization and clearing, and occasional vacuolar and plasma membrane breaks (Fig. 9p, q, r). This morphology was consistent with a lysoptosis-like phenotype. We concluded that HT3 cells underwent RIPK1 and MLKL-dependent cell death routine consistent with necroptosis in response to T + S + Q + C, but LMP did not contribute

appreciably to their demise. In the absence of SERPINB3, however, LMP was more extensive with cells converting to a lysoptosis-like death before a necroptosis phenotype emerged.

**Pyroptosis.** The activation of cytosolic pattern recognition receptors, such as inflammasomes, leads to CASP1 activation, processing of inflammatory cytokines (pro-IL-1β and pro-IL-18), and cleavage of gasdermin D (GSDMD)[83]. N-terminal GSDMD cleavage fragments polymerize and form pores within the plasma membrane leading to mature IL-1β and IL-18 release and associated lytic cell death[83]. The role of LMP in pyroptosis is confounded by studies showing that cytosolic cysteine peptidase activity can both activate and inhibit pyroptosis signaling[84–86]. We hypothesized that if increased LMP contributed to pyroptosis-induced lytic death, then the absence of SERPINB3 should augment cell death in response to a nucleotide-binding domain and leucine-rich repeat-containing (NLR) protein 3 (NLRP3) inflammasome activation. HT3[B3-WT] and HT3[B3-KO] cell lines were treated with several different NLRP3 inflammasome activators after LPS priming[87]. Nigericin-induced cell death (Fig. S17a). Sytox™ Orange uptake was significantly increased in the HT3[B3-KO] compared to HT3[B3-WT] cells. Live-cell confocal imaging indicated that there was a loss of 10 kDa TMR-labeled dextran staining in both HT3[B3-WT] and HT3[B3-KO] cell lines, with more Sytox positive cells in HT3[B3-KO] (Fig. S17a, b). Moreover, LMP was observed equally in both HT3 cell lines as demonstrated by equivalent loss of 3–70 kDa labeled dextrans (Fig. S17c–g). Unlike the pyroptosis-prone THP cells, neither GSDMD nor caspase-1 cleavage was detected in either HT3 cell line exposed to varying concentrations of nigericin (Fig. S17h, i)[88]. Interestingly, both HT3 cell lines were protected by incubation with E64d, but minimally by z-VAD or YVAD (Fig. S17i, j, k). As expected, THP1 cells were protected by these caspase inhibitors and E64d (Fig. S17l). These data suggested that nigericin-induced LMP and cell death, reminiscent of lysoptosis, but not pyroptosis, in both HT3 cell lines, with HT3[B3-KO] cells being more susceptible.

The cell lines were also incubated with an alternative NLRP3 inducer, *Staphylococcus aureus* α-hemolysin (Hla)[89,90]. HT3[B3-WT] cells showed increased SG uptake compared to HT3[B3-KO] cells (Fig. S18a). However, while the THP1 control cell line showed GSDMD cleavage, immunoblots from cell lysates from both HT3 cell lines detected cleavage of GSDME, but not GSDMD (Fig. S18b), supporting an alternative signaling pathway in HT3 cells. GSDME is cleaved to yield a functional N-terminal domain by CASP3, not CASP1[91]. This interpretation was consistent with protease inhibitor data, where z-VAD-fmk (pan-caspase inhibitor), but not VX735 (CASP1 inhibitor) or E64d, protected the cells from Hla-induced death (Fig. S18c–e). The lack of protection by E64d suggested that LMP was not involved in GSDME-associated cell death, which was underscored by the minimal loss of

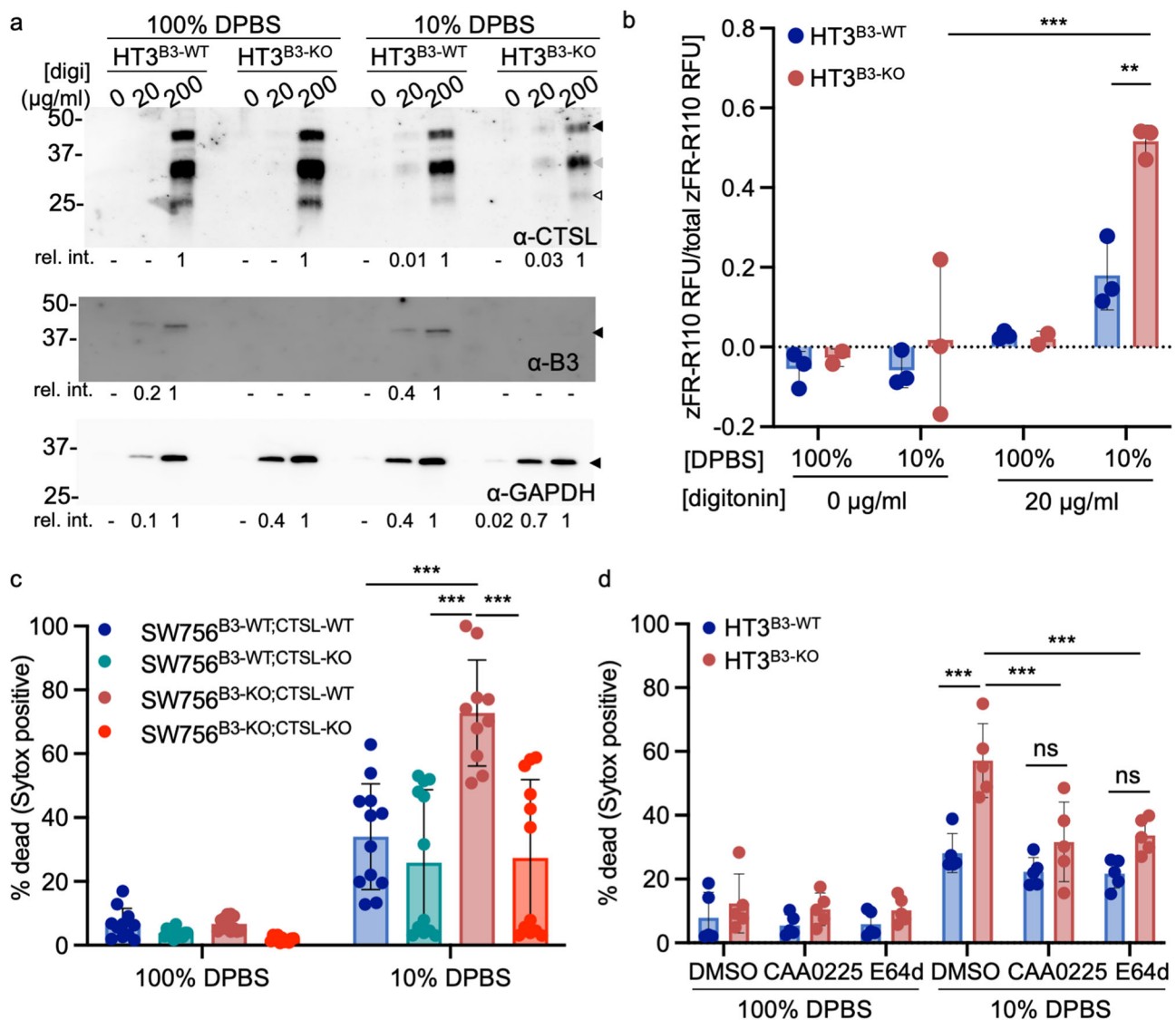

**Fig. 5 The lysoptosis-like cell death pathway involves promiscuous cytoplasmic proteolytic activity. a** Western blot analysis of lysosomal CTSL (goat polyclonal cathepsin L, Novus, AF952), nucleocytoplasmic SERPINB3 (α-B3; rabbit polyclonal SERPINB3/B4), and cytosolic GAPDH () in supernatants of HT3[B3-WT] and HT3[B3-KO] cell lines treated with 100 or 10% DPBS for 30 min at 37 °C, 5% $CO_2$ prior to treatment with DPBS containing 0, 20, or 200 µg/ml digitonin. The relative band intensities (rel. int.) for each lane were calculated (Image Lab, v6.1, Bio-Rad) and are shown under the corresponding lane. Note the detection of CTSL in the 20 µg/ml digitonin supernatants of both HT3[B3-WT] and HT3[B3-KO] treated with 10% DPBS but not the 100% DPBS controls. Black arrowheads indicate full-length proteins, gray and open arrowheads demark processed active forms of CTSL. **b** Pan-lysosomal cysteine protease activity (zFR-R110). Activity in supernatants of HT3[B3-WT] and HT3[B3-KO] cell lines treated with 100 or 10% DPBS for 30 min at 37 °C, 5% $CO_2$ prior to treatment with DPBS containing 0, 20, or 200 µg/ml digitonin. The activity was normalized to the 200 µg/ml control to account for total lysosomal cysteine protease protein levels. Note, the significantly enhanced cysteine protease activity in HT3[B3-KO] cytosolic supernatants (20 µg/ml digitonin) compared to HT3[B3-WT] in 10% DPBS cells treatment. **c** Bulk CTSL-KO SW756[B3-WT] or SW756[B3-KO] cell lines were generated by CRISPR/cas9 methodology. The percentage of the bulk population containing indels in the CTSL gene was confirmed by NGS sequencing (Table S1) and CTSL protein levels are shown in ref. [114]. Cell lines were then treated with either 100 or 10% DPBS in the presence of SG for 4 h. The % dead was calculated as the number of SG positive cells/the total number of cells as determined by high contrast brightfield imaging × 100. Analyses were compared using a two-way ANOVA with Tukeys' multiple comparisons ($n \geq 10$ replicates; *** $P < 0.001$). Note, the statistically significant reduction in % dead between SW756[B3-KO;CTSL-WT] and SW756[B3-KO;CTSL-KO]. **d** HT3[B3-WT] or HT3[B3-KO] cells were incubated with DMSO, 10 µM E64d, or 10 µM cathepsin L selective inhibitor, CAA0225 for 1 h prior to exposure to 100 or 10% DPBS for 14 h, and stained with SG. Percent dead was calculated as ((Sytox positive nuclei/total cells calculated by brightfield microscopy) × 100) ($n = 5$ replicates; ns not significant, *** $P < 0.001$). Uncropped immunoblots can be found in Supplementary Fig. 21.

Lysotracker[TM] staining observed over 12 h (Fig. S18f, g). We concluded that HT3 cell lines were resistant to canonical NLRP3 inflammasome activation, but could undergo GSDME-induced pyroptosis in response to Hla[92]. Although HT3[B3-KO] cell lines appeared to be more susceptible than HT3[B3-WT] cells to this form of pyroptosis, death did not involve lysoptosis.

Non-canonical inflammasome activation by cytoplasmic LPS is commonly activated by direct activation of CASP4/5 in human epithelial cells leading to GSDMD-mediated pyroptosis[93]. HT3[B3-WT] and HT3[B3-KO] cell lines were exposed to cytoplasmic LPS by nucleofection[93,94]. These lines showed comparable amounts of cell death based on SG uptake, and death was blocked by the CASP1

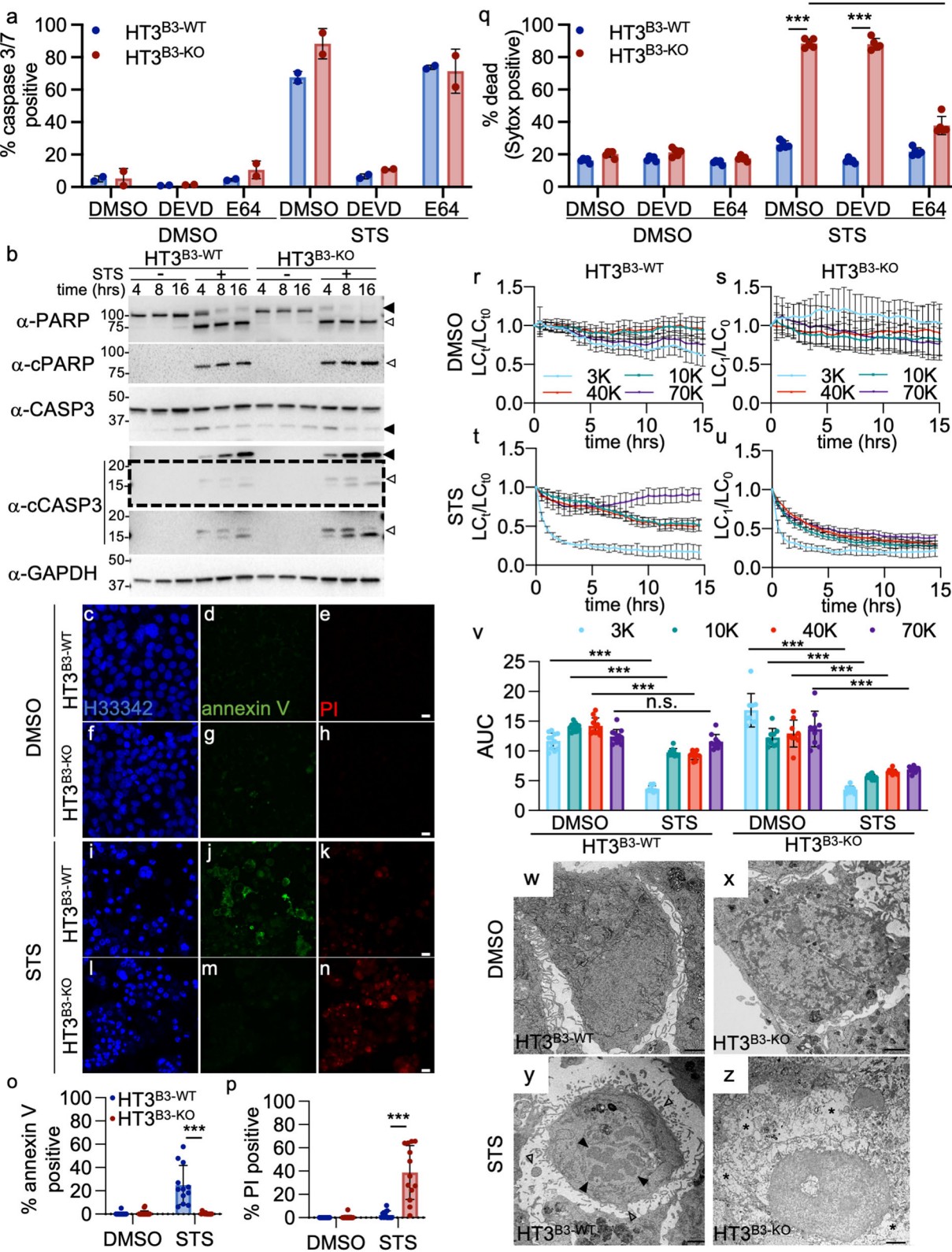

inhibitor, VX765, but not E64d (Fig. 10a). By immunoblotting, GSDMD, but not GSDME, N-terminal domains were detected ~1 h after nucleofection (Fig. 10b). There was no difference in the TEM morphology of HT3^B3-WT and HT3^B3-KO cell lines dying by this form of pyroptosis (Fig. S19). While the lack of protection from E64d, suggested that increased cytosolic cysteine peptidase activity did not contribute to GSDMD-mediated pyroptosis, to

directly assess LMP, we examined the cells lysosomal content by flow cytometry using the metachromatic dye, acridine orange (AO)[95]. To confirm that AO could detect LMP by flow cytometry, HT3^B3-WT and HT3^B3-KO cells were resuspended in a medium containing the lysosomal destabilizing compound, L-leucyl-L-leucine methyl ester (LLOMe)[96,97]. Confocal imaging of adherent HT3^B3-WT cells showed that LMP could be detected early after

**Fig. 6 Tumor epithelial cell lines null for SERPINB3 undergo lysoptosis-like death when exposed to the apoptosis inducer, staurosporine. a** HT3[B3-WT] or HT3[B3-KO] cells were pretreated with DMSO, 10 μM DEVD-CHO (DEVD), or E64 for 1 h prior to treatment with DMSO or 10 μM staurosporine (STS) in the presence of NucView caspase-3/7 activity detection reagent for 16 h (% caspase-3/7 positive = (#Nucview positive/# of nuclei) × 100), (Note, this caspase data was performed in conjunction with the Sytox staining shown in **q**). **b** Immunoblot analysis of PARP, cleaved PARP (cPARP), caspase-3 cleavage, and actin (α-act) in HT3[B3-WT] or HT3[B3-KO] cell lines treated with DMSO (−) or 10 μM STS (+). The dashed area is shown with contrast enhancement below (black arrowheads = full length, open arrowheads = cleavage product). **c–n** Representative confocal images (scale bar = 25 μm) of HT3[B3-WT] or HT3[B3-KO] cells treated with DMSO or 10 μM STS for 18 h and then stained with H33342 (blue), annexin V (green), and PI (red). **o, p** Quantification of the percent annexin V and PI-positive cells from $n \geq 10$ fields shown in (**c–n**). **q** HT3[B3-WT] or HT3[B3-KO] cells were treated as in (**a**) and incubated with Sytox™ orange for 16 h (percent dead = (# of Sytox™ positive/# of nuclei) × 100). **r–u** To quantify lysosomal content, HT3[B3-WT] or HT3[B3-KO] cells were incubated with fluorescently labeled different molecular weight dextrans as above, prior to exposure to DMSO or 10 μM STS. Live-cell confocal microscopy was used to determine the number of lysosomes at each time point ($LC_t$) normalized to the initial lysosome count at time zero ($LC_{t0}$) over $n \geq 9$ fields. **v** AUC for each dextran over time for the experiment in **r–u** (means compared using two-way ANOVA with Tukey's multiple comparisons test). **w–z** Representative TEM images (scale bars = 500 nm) of HT3[B3-WT] or HT3[B3-KO] cells treated with DMSO or 10 μM STS for 16 h. Apoptotic morphology indicated with open (membrane budding) and closed arrows (chromatin fragmentation), and necrotic morphology (vacuolization, cytoplasmic clearing) by an asterisk. Unless otherwise noted, a representative of ≥3 replicates is shown, and the means ± SD were compared using a two-tailed $t$-test (ns not significant, ***$P < 0.001$). Original imaging data files can be found at https://data.mendeley.com/datasets/6g2typdf63/1. Uncropped immunoblots can be found in Supplementary Fig. 21.

LLOMe treatment, with complete loss of lysosomal staining by 240 min with all of the cells dying, as indicated by Sytox™ Red uptake (Fig. S20). Flow cytometry using AO and Sytox™ Blue, followed a similar pattern (Fig. 9c, e, f). Both HT3[B3-WT] and HT3[B3-KO] exposed to 5 mM LLOMe showed that lysosomal staining in live cells (lysosome⁺-live cells; Quadrant 1 (Q1); Fig. 10c, e, f) decreased over time. This decrease was associated with an increase in lysosome⁻-live cells at 20 min (Q4; Fig. 10c, e, f). By 120 min, the percent of lysosome⁻-live cells decreased (Q4; Fig. 10c, e, f). As expected, there was a progressive increase in lysosome⁻-dead cells over time (Q3; Fig. 10c, e, f). To summarize (purple arrow, Fig. 10e), most of the cells were lysosome⁺-live cells at 20 min into the experiment (Q1 in Fig. 10c, e, f). With LLOMe treatment, cells shifted to the lysosome⁻-live-cell compartment (Q4 in Fig. 10c, e, f), and then to the lysosome⁻-dead-cell compartment over time (Q3 in Fig. 10c, e, f). Very few cells were detected in the lysosome⁺-dead cell compartment (<2%), even after 120 min of LLOMe treatment (Q2 in Fig. 10c, e, f). We concluded that LMP occurred prior to cell death after treatment with LLOMe and flow cytometry could be utilized to study the relationship of LMP to cytoplasmic LPS-induced pyroptosis in these tumor cell lines. Treatment with LPS via nucleofection yielded a different flow-cytometry pattern over time (green arrow, Fig. 10e). Induction of pyroptosis by LPS resulted in the emergence of a lysosome⁺-dead-cells and not a lysosome⁻-live-cell population (compare Q2 vs. Q4 in Fig. 10d, e, g). Ultimately, lysosome⁺-dead-cells were lost, and only lysosome⁻-dead cells were detected (Q3 in Fig.10d, e, g). Taken together, we concluded that non-canonical inflammasome activation leads to GSDMD-mediated pyroptosis in these tumor cell lines. However, detectable LMP was mostly a postmortem event and explained why the loss of SERPINB3 did not increase the susceptibility of HT3 cells to death induced by either GSDMD or GSDME N-terminal domains.

## Discussion
Since Christian De Duve's discovery of lysosomes in 1955, there has been a long-standing debate on whether LMP and the displacement of lysosomal hydrolases into the cytosol compromise cell survival or serve as the prelude to postmortem autolysis[28]. While data suggest that LMP contributes to the cytopathic changes associated with many RCD pathways, death is not lysosomal-dependent unless the cells are treated harshly or exposed to lysosomotropic detergents that disrupt lysosomal membranes[1]. These observations support the notion that LMP and lysosomal hydrolases are conscripted by other cell death routines to augment their killing, but LDCD per se does not signify the action of a molecularly defined, stand-alone RCD pathway analogous to others described[1]. Previously, we characterized a form of LDCD in *C. elegans* that challenges this hypothesis[41]. Since the necroptosis and pyroptosis pathways are not conserved in *C. elegans*, and neither apoptosis nor entosis are active in the adult soma, this virulent cell death phenotype represents a clear example of a stand-alone LDCD pathway in metazoans[41,47,98]. We designated this cell death process "lysoptosis" to distinguish it from those forms of LDCD that are not negatively regulated by cytoplasmic serpins (e.g., SRP-6 or other types of protease inhibitors), and whose attributes are limited only to the detection of LMP and cathepsin activity[1].

The objective of this study was to determine whether lysoptosis was evolutionarily conserved in higher eukaryotes by identifying its presence among the more commonly described RCD pathways. Since *mSerpinb3a* (mouse) and *SERPINB3* (human) are orthologous intracellular/Clade B lysosomal cysteine peptidase inhibitors homologous to *C. elegans srp-6*, we used a similar genetic loss-of-function strategy to enhance the detection of lysoptosis in mice and human cell lines.

As in *C. elegans*, intestinal cultures derived from *mSerpinb3a⁻/⁻* mice were more sensitive to hypoosmotic stress, compared to those from *mSerpinb3a⁺/⁺* mice. The killing kinetics were rapid, with the majority of cells dying within 30 min of exposure. The FIECs showed extensive LMP prior to plasma membrane permeabilization and demonstrated the necrotic features that were similar to those of *srp-6(-) C. elegans*[41]. Cell death was inhibited by E64d, and there was no evidence of apoptosis or necroptosis. Similarly, the human squamous cell cancer lines null for *SERPINB3* were more sensitive than controls upon exposure to hypotonic stress. Moreover, assessment of LMP by the release of fluorescently labeled dextrans in the HT3[B3-KO] cells phenocopied that observed in the *C. elegans srp-6(-)* intestinal cells. This response was blunted in wild-type animals and cells, which attests to the role of the intracellular serpins in moderating LMP and lysoptosis-induced cell death. Analysis of the lysosomal cysteine proteases in the human cervical cancer cell lines indicated that this lysoptosis-like death resulted in the release of active CTSL into the cytoplasm of SW756[B3-KO] cells. Genetic or pharmacologic reduction of CTSL activity resulted in significant protection of the cells to hypotonic stress. This result suggested that, as in the *C. elegans* model[41], CTSL is one of the major cysteine proteases involved in the lysoptosis pathway in mammalian cells. However, additional studies in different cell types and/or organisms are required to

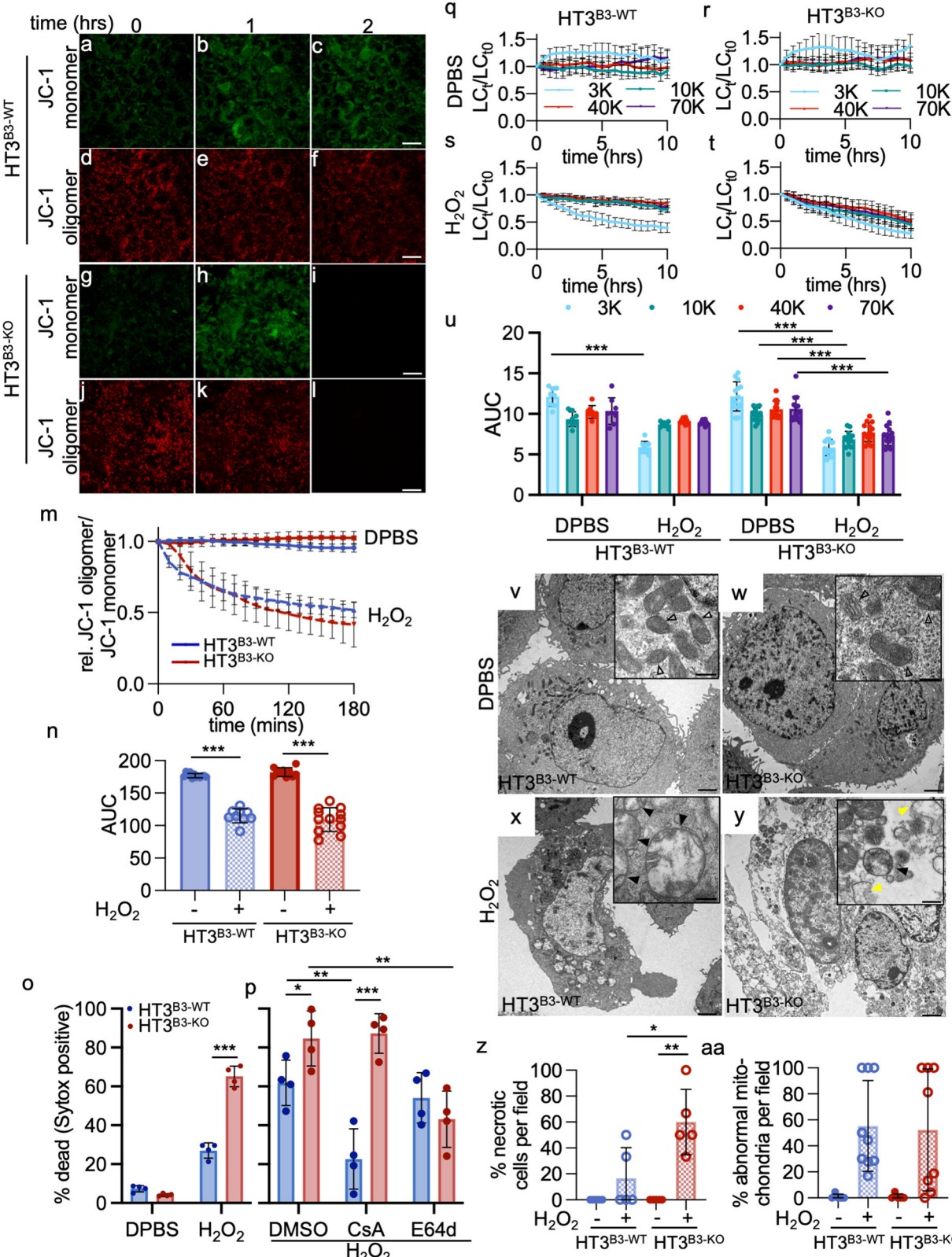

determine whether other types of CTSs also play a prominent role in cell-specific lysoptosis cell death routines.

One means to study evolutionary conservation of function is to express mammalian proteins in more primitive loss-of-function models and assess for transgenic rescue. Several of the mammalian intracellular serpins including human SERPINB3 and -B13, as well as mSerpinb3a possess lysosomal cysteine protease

inhibitory activity in vitro[50,65,70]. SERPINB3 and -b3a expressing transgenic *srp-6(-) C. elegans* showed protection against hypoosmotic stress, suggesting that these serpins replaced the function of SRP-6 in *C. elegans*. Additionally, the A341R (P14) mutation of SERPINB3, which blocks serpin protease inhibition, suppressed the rescue[50]. Of note, SERPINB3, but not SERPINB3[A341R], was also found to protect the human HT3[B3-KO]. These

**Fig. 7 Initiation of mTP-dependent necrosis by $H_2O_2$ triggered lysoptosis-like death in HT3[B3-KO] cells. a–n** Representative live-cell confocal images of HT3[B3-WT] and HT3[B3-KO] cells stained with JC-1, then treated with 5 mM $H_2O_2$. Images were collected using Ex 488 nm with detection windows of Em 498–532 nm (JC-1 monomer, green) and 566–606 nm (JC-1 oligomer, red) every 5 min over 3 h (select time points shown). **m** Fluorescence intensity of JC-1 oligomer/JC-1 monomer normalized to time 0 (rel. JC-1 oligomer/JC-1 monomer) of the experiment in panels **a–n**. Time points with no fluorescence in either channel (dead cells) were eliminated from the analysis. **n** Area under the curve (AUC) for multiple fields for experiment in panels **a–n** (compared using one-way ANOVA with Tukey multiple comparisons). **o** Percent dead ((#Sytox™ positive/# of nuclei) × 100) of HT3[B3-WT] or HT3[B3-KO] cells treated with 5 mM $H_2O_2$ or DPBS for 4 h. **p** Percent dead HT3[B3-WT] or HT3[B3-KO] cells pretreated with DMSO, 10 µM cyclosporin A (CsA), or E64d for 1 h prior to exposure to 5 mM $H_2O_2$ or DPBS for 4 h. **q–t** To quantify lysosomal content, HT3[B3-WT] or HT3[B3-KO] cells were incubated with fluorescently labeled different molecular weight dextrans as above and imaged using live-cell confocal microscopy. The number of lysosomes at each time point ($LC_t$) were normalized to the lysosome count at time zero ($LC_{t0}$). **u** AUC for each dextran over time for an experiment in **q–t** (compared by two-way ANOVA with Tukey's multiple comparisons test). **v–y** Representative TEM images (scale bars = 2 µm) of HT3[B3-WT] or HT3[B3-KO] cells treated with DPBS or 5 mM $H_2O_2$ for 4 h. Insets show normal or abnormal mitochondria (open and dark arrowheads, respectively; scale bars = 500 nm). Yellow arrowheads indicate enlarged vesicle and cytoplasmic clearing. **(z–aa)** Quantification of TEM images from the experiment in **v–y** showing the number of necrotic cells per field (**z**) and abnormal mitochondria per field (**aa**) (≥5 fields). Unless otherwise noted, a representative of ≥3 replicates is shown, and the means ± SD were compared using a two-tailed $t$-test (ns not significant, ***$P < 0.001$, **$P < 0.01$, *$P < 0.05$). Original data files can be found at https://data.mendeley.com/datasets/tpp7xnmwxg/1.

results confirmed that the protease inhibitory activity of SERPINB3 was required to block lysoptosis. SERPINB4, which inhibits cathepsin G and to a lesser extent, lysosomal cysteine proteases, also rescued the hypoosmotic stress lethality. We did not determine if the decreased activity of SERPINB4 in this system was due to its inhibitory profile or the high levels of protein expression[51]. However, both SERPINB1 and -B6 are inhibitors of cathepsin G[99], but they did not protect the *srp-6(-)* animals, suggesting the serine protease inhibitory activity of these serpins was not required. Moreover, the expression of SERPINB4 in the human HT3[B3-KO] cell line did not rescue the loss of SERPINB3 against hypotonic stress in these cells. Taken together, these data suggest that SERPINB4 does not afford protection. SERPINB13 is another Clade B member that inhibits lysosomal cysteine proteases[65]. This serpin also did not suppress the hypoosmotic stress phenotype in *C. elegans*, potentially due to the mosaic nature of the expression patterns associated with the use of extrachromosomal arrays, or that this serpin neutralizes nematode cysteine peptidase with a lower second-order rate constant than that of SERPINB3 or mSerpinb3a[70,100]. We concluded that mammalian intracellular serpins demonstrated the ability to impair lysoptosis in *C. elegans*, which suggested that this function has been conserved evolutionarily in vertebrates.

Plants, which lack caspases and cannot undergo apoptosis, undergo a limited number of RCD pathways relative to their mammalian counterparts[101]. Vacuolar-mediated cell death (VCMD) is triggered by the loss of tonoplast (the membrane surrounding the central vacuole) integrity, the release of vacuolar proteases into the cytosol, and extensive destruction of the plant cell. In *Arabidopsis thaliana*, the release of the vacuolar papain-like cysteine protease, RD21, into the cytosol mediates VCMD[102]. RD21 is neutralized by AtSerpin1 and overexpression or loss-of-function of this serpin enhances or protects against cell death, respectively[102]. AtSerpin1 is an intracellular serpin with a high degree of amino acid sequence homology with that of the human clade B serpins, including SERPINB3[103]. VCMD, the major RCD in plants, is analogous to lysoptosis pathway in animals and underscores the long evolutionary history of this cell death program, which may date back to the origins of Eukaryota over ~1000 Ma[104].

We detected low-level lysoptosis in wild-type *C. elegans* intestinal cells, mouse FIECs, and human squamous cell carcinoma cell lines in response to hypotonic stress. However, the phenotype was markedly exacerbated by the loss of *srp-6*, *mSerpinb3a*, and *SERPINB3*, respectively. This observation prompted us to determine whether limited LMP or the more extreme lysoptosis was active in other RCD pathways. Endolysosomal

staining remained relatively intact in FIECs from wild-type animals induced to undergo apoptosis or necroptosis. In contrast, $mSerpinb3a^{-/-}$ FIECs demonstrated a complete loss of endolysosomal staining followed by the breakdown of plasma membrane integrity. Thus, in the absence of mSerpinb3a, necroptosis and apoptosis inducers activated lysoptosis. Since the FIECs are a fastidious system, we turned to human tumor cell lines null for SERPINB3 to determine whether the predominance of lysoptosis was reproducible even after appropriate activation of another RCD. Indeed, HT3[B3-WT] cells exposed to STS showed both biochemical and morphological features of apoptosis. In contrast, HT3[B3-KO] cells showed a dramatic increase in LMP and a lysoptosis-like phenotype despite executioner caspase activation and PARP cleavage. Moreover, death proceeded in the presence of DEVD-CHO and could only be blocked by a papain-like cysteine protease inhibitor. Similarly, wild-type cells died in response to the necroptosis or MPT-DN pathway-specific inducers and were blocked by pathway-specific inhibitors. In contrast, *SERPINB3* null cells showed evidence of necroptosis and MPT-driven necrosis activation (RIPK1 phosphorylation and JC-1 mitochondrial leakage, respectively). However, LMP was accentuated, and the cells exhibited extensive necrotic features consistent with lysoptosis. In both cases, death could only be prevented by E64d. The association of three RCD pathways; apoptosis, necroptosis, and MPT-DN, with LMP, suggested that there is an important biological link between these events. When we examined LMP using lysosomes loaded with 3–70 kDa fluorescently labeled dextrans, a different graded pattern of release was detected between HT3[B3-WT] and HT3[B3-KO] cells. Wild-type cells showed a graded, slower rate of release of the fluorescently labeled dextran, particularly with the lower molecular mass species (3 kDa » 10 kDa). In contrast, the *SERPINB3* null cells showed a similar ordered response, but the release was more rapid and always included the higher molecular weight species. Taken together, we concluded that apoptosis, necroptosis, and MPT-DN generated a limited degree of LMP that permitted the release of divalent cations and/or <10 kDa molecular mass species and possibly marginal amounts of proteolytic enzymes. Thus, these lower molecular mass species may serve as more important co-factors in the activation of these RCD pathways; with lysosomal cysteine proteases merely occupying a secondary role in the disintegration phases of dying or dead cells. However, in the absence of SERPINB3 or other cytoplasmic cysteine proteinase inhibitors (e.g., cystatins), we hypothesize that the cytosolic release of even trivial amounts of lysosomal cysteine peptidases may allow for the promiscuous catalysis of lysosomal integral membrane proteins from the cytoplasmic face of the

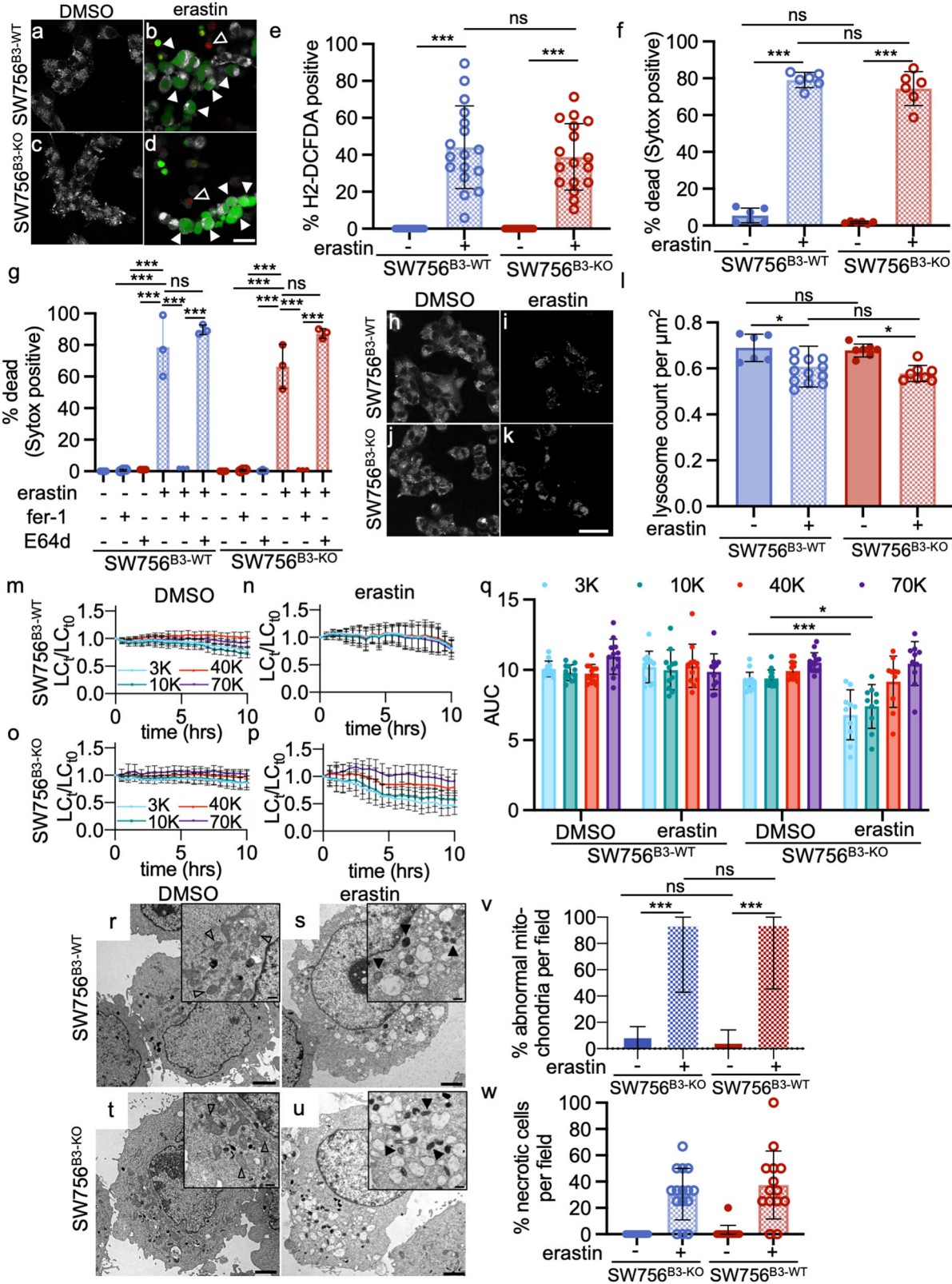

organelle, analogous to the cleavage of the lysosomal integral membrane protein, LAMP2, by calpains[105]. In turn, this catalysis could accentuate LMP, and lead to a feedback loop triggering the explosive propagation and disruption of the endolysosomal compartment. A positive feedback loop involving a protease-mediated mechanism could explain the rapid and volatile nature of lysoptosis also induced by other noxious stimuli (e.g., lysosomotropic

detergents, excessive free radicals) and the vacuolated necrotic morphology commonly observed in these cells[106]. Other possibilities include the sequential occlusion or destruction of amino acid export channels, generation of an intraluminal hyperosmolar state, influx of fluid, and organellar swelling and disruption.

Unlike the other forms of RCD, neither pyroptosis nor ferroptosis co-activated lysoptosis in the absence of SERPINB3.

**Fig. 8 Erastin induces ferroptosis, but minimal LMP, in SW756[B3-WT] and SW756[B3-KO] cells. a–d** Representative confocal images (scale bar = 25 µm) of SW756[B3 WT] or SW756[B3 KO] cells treated with DMSO or 20 µM erastin for 12 h in the presence of Sytox™ Orange (red, open arrowheads), then stained with the ROS indicator, H2-DCFDA (green, white arrowheads) and Mitotracker™ Deep Red (white). **e** Quantification of the percentage of H2-DCFDA positive cells in SW756[B3 WT] or SW756[B3 KO] cells after treatment with DMSO (−) or 20 µM erastin (+) for 12 h (means were compared using a one-way ANOVA with Tukey's multiple comparisons). **f** The percent dead 12 h after treatment with erastin ((# Sytox positive nuclei/# blue nuclei) × 100, means compared using a one-way ANOVA with Tukey's multiple comparisons). **g** SW756[B3 WT] or SW756[B3 KO] cells were incubated for 1 h with DMSO (−), ferrostatin-1 (fer-1), or E64d, prior to addition of DMSO (−) or 20 µM erastin (+) for 12 h (% dead and statistical comparison calculated as in panel **f**). **h–k** Representative confocal images (scale bar = 25 µm) of SW756[B3 WT] or SW756[B3 KO] cells treated with DMSO or 20 µM erastin for 12 h and stained with Lysotracker™ Red. **l** Quantification of the lysosomal count per cellular µm² (≥6 fields). **m–p** SW756[B3 WT] or SW756[B3 KO] cells were incubated with fluorescently labeled different molecular weight dextrans as above prior to exposure to either DMSO or 20 µM erastin and imaged using live-cell confocal microscopy. The number of lysosomes were quantified as above. **q** Area under the curve (AUC) for each dextran over time. (two-way ANOVA with Tukey's multiple comparisons test). **r–u** Representative TEM images (scale bars = 2 µm) of SW756[B3 WT] or SW756[B3 KO] cells treated with either DMSO or erastin for 12 h. Insets show normal and abnormally appearing mitochondria (open and black arrowheads, respectively; scale bars = 500 nm). **v, w** Quantification of the number of abnormal mitochondria (**v**) and necrotic cells (**w**) in the experiment in panels **r–u** (≥10 fields). The means ± SD were compared using a two-tailed t-test (ns not significant, ***$P < 0.001$, **$P < 0.01$, *$P < 0.05$). Original data can be found at https://data.mendeley.com/datasets/9m9f4cfhxy/1.

However, we were unable to detect caspase-1 and GSDMD cleavage using the canonical NLRP3 pyroptosis inducer, nigericin, in the HT3 cell lines. Moreover, Hla-induced GSDME cleavage, not GSDMD in the HT3 cell lines, indicative of CASP3 activation[91]. We could detect GSDMD-mediated pyroptosis in cells activated by the non-canonical pathway, in which electroporated LPS directly activates caspase-4/5[93,107,108]. Under these conditions, little LMP occurred antemortem with most of the LMP occurring after the loss of plasma membrane integrity. These data suggested that pyroptosis neither required nor triggered extensive LMP in the HT3 cell lines.

The original description of ferroptosis did not ascribe a role for LMP, and death was not blocked by E64d[77]. Subsequently, a role for the lysosome in generating Fe(II) for Fenton-type reactions leading to lipid peroxidation have been recognized[10,79]. We were unable to detect any evidence of increased LMP after induction of ferroptosis in wild-type cancer cells. In the absence of SERPINB3, there was lysosomal leakage of the 3–10 kDa dextran species. However, cell death was not enhanced, and E64d, unlike fer-1, had no protective effect. We concluded that lysoptosis was not activated by the induction of ferroptosis.

Although the mSerpinb3a$^{−/−}$ FIECs were more susceptible to lysoptosis, the effects of serpin loss and predisposition to LMP had not been assessed comprehensively in vivo. In a mouse model system, M. tuberculosis-infected macrophages are more susceptible to protease-induced death in mSerpinb3a$^{−/−}$ animals[109]. Also, SERPINB3 transgenic overexpression protects bax$^{−/−}$ bak$^{−/−}$ baby mouse kidney cells from the damaging effects of LMP[110].

Since the discovery of the SERPINB3 (a.k.a. squamous cell carcinoma antigen (SCCA)) in 1977, numerous clinical reports show that elevated circulating levels of this protein are a poor prognostic factor for patients with advanced squamous cell carcinomas and other epithelial malignancies[111,112]. We hypothesize that tumor cells with increased expression of this cytoprotective serpin are afforded a distinct survival advantage[113]. We recently reported that high circulating levels of SERPINB3 are also an early indicator of a poor response to chemoradiation therapy in patients with cervical squamous cell carcinomas[61]. Moreover, as we show in a companion paper to this study, the HT3[B3-KO] and SW756[B3-KO] cell lines demonstrated enhanced sensitivity and a lysoptosis-like cell death phenotype in response to irradiation in vitro and in vivo[114]. These clinical observations raise the intriguing possibility of exploiting the discovery of small-molecule inhibitors that incapacitate specific serpins to enhance the response of SERPINB3-expressing tumor cells to cancer therapeutics, irradiation, or lysoptosis inducers. Taken together, these studies provide evidence that SERPINB3/mSerpinb3a are cytoprotective in vivo.

The role of SERPINB3 as a pro-survival factor that protects against the toxicity of lysosomal proteases can be extended to other members of the intracellular/clade B serpin family. SERPINB1 is highly expressed in the myelocytic lineage and neutralizes neutrophil granule proteases including cathepsin G, neutrophil elastase, and proteinase 3[115]. Neutrophils from SERPINB1 null mice are markedly susceptible to LLOMe as compared to wild-type, and death could be suppressed by genetic loss of cathepsin G, but not by incubation with caspase inhibitors[116]. SERPINB9 is a potent inhibitor of granzyme B (GZMB), which is stored in the lytic granules (lysosome-related organelles) of CD8 + cytotoxic T cells (CTLs), regulatory T cells, and natural killer (NK) cells[117]. SERPINB9 has a pro-survival function in CTLs and NK cells by blocking activation-induced cell death, which is induced by GZMB leakage from lytic granules[118–120].

In the 1963 Ciba symposium, the role of lysosomes as "suicide bags" that participated in physiological involutions was proposed[121]. By 1983, this concept "waned considerably…as it proved difficult to distinguish clearly between the actual rupture of lysosomes and their mere fragilization…and between premortal injuries to the lysosomes, possibly causal of cell death, and postmortem alterations that would be no more than a consequence of cell death[28]." We suggest that over half a century later, Christian de Duve and his detractors were both correct.

## Methods

**C. elegans growth and maintenance.** All C. elegans strains were grown and maintained using standard procedures[122]. Both the wild-type N2 Bristol strain and mutant srp-6(ok319) were grown at 20–25 °C on standard nematode growth medium (NGM) plates seeded with E. coli OP50 as the food source[41].

**C. elegans plasmid construction.** The P$_{srp-6}$mammalianSERPIN$^{cDNA}$::GFP plasmids were constructed by first amplifying the srp-6 promoter from N2 genomic DNA using primers that contained SphI restriction sites (Forward: 5′-CGCGCATGCGGTCTCAACCACCTTTTCCTCCGC-3′; Reverse: 5′-GCGGCATGCCGAAATTGAAGAAAAGTGTTCAC-3′) and ligated into the SphI restriction site of pPD95.77, which contains GFP DNA and the unc-54 3′ UTR, to make the P$_{srp-6}$GFP[123]. The cDNA sequences for the mammalian serpins were amplified using the following primer sets: SERPINB3 (Forward: 5′-GGGGTCGACAAAAATGAATTCACTCAGTGAAGCC-3′; Reverse: 5′-TTGGATCCCGGGGATGAGAATCTGCC-3′), SERPINB4 (Forward: 5′- GGGGTCGACAAAAATGAATTCACTCAGTGAAGCC-3′; Reverse: 5′-TTGGATCCTGGGGATGAGAATCTGCC-3′), Serpinb3a (Forward: 5′-GTCAGTCGACAAAAATGCATTTGTTTGCTGAAGCCAC-3′; Reverse: 5′- ATGGATCCAGGGGAGGAGTTCTGCC-3′), SERPINB1 (Forward: 5′-CCTTCTGCAGAAAATGGAGCGCTGAGC-3′; Reverse: 5′-AAGCGGATCCTCTAAGGGGAAGAAAATCTCCC-3′), SERPINB6 (Forward: 5′-ATCGGTCGACAAAAATGGATGTTCTC-3′; Reverse: 5′-CCCGGAGAGGGAAAAGCGG-3′), SERPINB13 (Forward: 5′- GTCAGTCGACAAAAATGGATTCACTTGGCGCCG-3′; Reverse: 5′-ATGGATCCAGGAGAAGAAAATCTGCCGAAG-3′). SERPINB3, -B4, -B3a, and -B13 were ligated into the SalI-BamHI restriction enzyme sites of P$_{srp-6}$GFP, SERPINB1 was ligated into the PstI-BamHI restriction

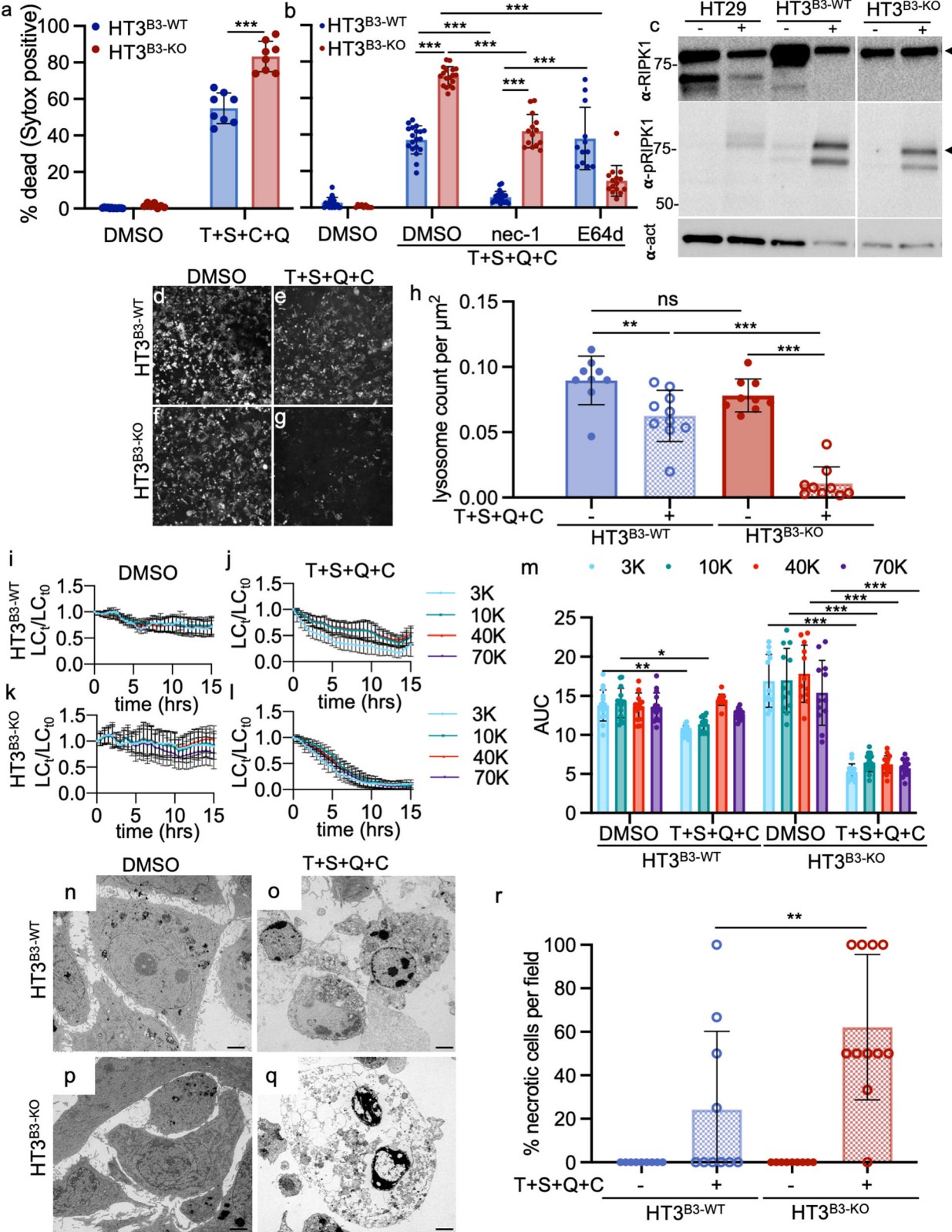

enzyme sites of P_{srp-6}GFP and SERPINB6 was ligated into the SalI-SmaI restriction enzyme sites of P_{srp-6}GFP to create the plasmids P_{srp-6}SERPINB3 ^{cDNA}::GFP, P_{srp-6}SERPINB4 ^{cDNA}::GFP, P_{srp-6}Serpinb3a ^{cDNA}::GFP, P_{srp-6}SERPINB1 ^{cDNA}::GFP, P_{srp-6}SERPINB6 ^{cDNA}::GFP and P_{srp-6}SERPINB13 ^{cDNA}::GFP. A P14 mutation (V341R) was generated in P_{srp-6}SERPINB3::GFP construct using the Quikchange™ site directed mutagenesis kit (Agilent Technologies, Santa Clara, CA) with the primers forward: 5′-GGTTACAGAGGAGGGACGAGAAGCTGCAGCTGCC-3′; reverse: 5′-GGCAGCTGCAGCTTCTCGTCCCTCCTCTGTAACC-3′ per manufacturer's instructions to generate the P_{srp-6}SERPINB3(V341R) ^{cDNA}::GFP construct. All plasmid sequences were confirmed by Sanger sequencing.

**C. elegans transgenic strain generation.** The gonads of adult srp-6(ok319) mutant C. elegans were injected with 50 ng/μl of the individual P_{srp-6}mammalianSERPIN::GFP construct plus 10 ng/μl of the co-injection marker P_{myo-2}mCherry and 40 ng/μl of pBluescript SK(-) filler DNA to create the following lines: VK1276 (srp-6(ok319);vkEx1276[P_{srp-6}SERPINB3^{cDNA}::GFP; P_{myo-2}mCherry]), VK1279 (srp-

**Fig. 9 Tumor epithelial cell lines null for SERPINB3 undergo lysoptosis-like death under conditions that induce RIP1K-dependent necroptosis. a** Percent dead ((# Sytox positive nuclei/# of blue nuclei) × 100) of HT3$^{B3-WT}$ or HT3$^{B3-KO}$ cells 12 h after incubation with DMSO or 10 ng/μL human TNFα + 0.1 μM BV6 SMAC mimetic + 0.5 μM qVD-OPh + 1 μM cyclohexamide (T + S + Q + C). **b** Percent dead (calculated as in panel **a**) of HT3$^{B3-WT}$ or HT3$^{B3-KO}$ cells incubated with DMSO, 50 μM nec-1 or 10 μM E64d for 1 h prior to exposure to T + S + Q + C for 8 h. **c** Immunoblot analysis of RIP kinase 1 (α-RIPK1), phosphorylated RIP kinase 1 (α-pRIPK1), and actin (α-act) in HT29 (control cell line), HT3$^{B3-WT}$, and HT3$^{B3-KO}$ treated with DMSO (−) or T + S + Q + C (+) for 12 h (black arrowheads denote bands for RIPK1 and pRIPK1 based on molecular mass). **d–g** Representative confocal images (scale bar = 25 μm) of HT3$^{B3-WT}$ or HT3$^{B3-KO}$ cells incubated with DMSO or T + S + Q + C and stained with Lysotracker Deep Red (white). **h** Quantification of the lysosomal count per cellular μm$^2$ for the experiment in **d–g** (≥9 fields, compared using a one-way ANOVA with Tukey's multiple comparisons). **i–l** To quantitate lysosomal content, HT3$^{B3-WT}$ or HT3$^{B3-KO}$ cells were incubated with 3 kDa Cascade blue, 10 kDa Alexa488, 40 kDa TMR, and 70 kDa Texas red labeled dextrans prior to exposure to either DMSO or T + S + Q + C and imaged using live-cell resonance scanning confocal microscopy (≥20 z-planes, ≥10 fields). The number of lysosomes at each time point (LC$_t$) were normalized to the lysosome count at time zero (LC$_{t0}$). **m** AUC for each dextran over time for an experiment in **i–l** (two-way ANOVA with Tukey's multiple comparisons test). **n–q** Representative TEM images (scale bars = 2 μm) of HT3$^{B3-WT}$ or HT3$^{B3-KO}$ cells treated with DMSO or T + S + Q + C. **r** Quantification of the number of necrotic cells per field (≥10 fields) for the experiment in **n–q**. Unless otherwise noted, a representative of ≥3 replicates is shown, and the presented means ± SD were compared using a two-tailed t-test (ns not significant, ***P < 0.001, **P < 0.01, *P < 0.05). Original imaging data can be found at https://data.mendeley.com/datasets/gvgnvkw4ct/1. Uncropped immunoblots can be found in Supplementary Fig. 21.

6(ok319);vkEx1279[P$_{srp-6}$SERPINB4$^{cDNA}$::GFP; P$_{myo-2}$mCherry]), VK1282 (srp-6(ok319);vkEx1282[P$_{srp-6}$Serpinb3a$^{cDNA}$::GFP; P$_{myo-2}$mCherry]), VK2649 (srp-6(ok319);vkEx2649[P$_{srp-6}$SERPINB1$^{cDNA}$::GFP; P$_{myo-2}$mCherry]), VK2651 (srp-6(ok319);vkEx2651[P$_{srp-6}$SERPINB6$^{cDNA}$::GFP; P$_{myo-2}$mCherry]), VK2653 (srp-6(ok319);vkEx2653[P$_{srp-6}$SERPINB13$^{cDNA}$::GFP; P$_{myo-2}$mCherry]) and VK1277 (srp-6(ok319);vkEx1276[P$_{srp-6}$SERPINB3(A341R)$^{cDNA}$::GFP; P$_{myo-2}$mCherry]).

**SERPINB3 knockout cell line production and characterization.** HT3$^{B3-KO}$ and SW756$^{B3-KO}$ cell lines were generated as previously described in ref. [61]. In brief, the LentiCRISPR V2 (Addgene 52961, Addgene, Watertown, MA, USA) lentiviral vector containing hSpCas9 and either an empty sgRNA cassette (CRISPR-control cells) or a guide RNA targeting exon 5 of SERPINB3 gene was designed using standard cloning procedures. Lentivirus was packaged in HEK293T cells, concentrated using Lenti-X Concentrator (Clontech, Mountain View, CA, USA), and used to infect HT3 and SW756 parental cells (ATCC©) with polybrene for 24–48 h. Pools were selected with puromycin and single cell clones obtained through flow sorting into 96-well dishes. Target locus KO was confirmed in selected clones by targeted sequencing of the genomic locus, mRNA using qRT-PCR, and protein levels with Western blot (Fig. S5). For CRISPR-control cells, the puromycin-selected pool was used. Genomic DNA was isolated from cells using DNeasy Blood and Tissue Kit (Qiagen, Hilden, Germany), the SERPINB3 locus was amplified using PCR (forward primer: ATGGAAGAAAGCAAGGCAGA, reverse primer: TAAAGGGGAAGGGAGATGCT), and sequenced using Sanger sequencing (Primer: CTTCGGTGACCTCTTTCTTGC). Transcript levels were determined using qRT-PCR on cDNA isolated from each of the cell lines (forward primer: CGCGGTCTCGTGCTATCTG, reverse primer: ATCCGAATCCTACTA-CAGCGG). Western blot analysis was performed on whole-cell lysates with proteinase and phosphatase inhibitors, equal parts protein (25–30 μg) and Laemmli sample buffer (Santa Cruz Biotechnology) were boiled at 95 degrees Celsius for 10 min and gel electrophoresed on 4–20% gradient gels (Mini-Protean TGX; Bio-Rad), transferred to PVDF blot using the Trans-Blot Turbo Transfer System (Bio-Rad), blocked with 5% milk:TBS-Tween and incubated with 1:4000 anti-SER-PINB3/4 antibody (NBP2, Novus International, Saint Louis, MO, USA) or anti-SERPINB4 monoclonal antibody (8H11), overnight at four degrees Celsius. 1:100,000 anti-Actin (A5441, Santa Cruz Biotechnology, Dallas, TX, USA) was incubated for 2 h at room temperature. Anti-mouse or anti-rabbit HRP-conjugated secondary antibody was used for detection with ECL chemiluminescent reagent (GE Healthcare Life Sciences), visualized, and quantified using the Bio-Rad ChemiDoc MP imaging system and Image Lab software (Bio-Rad).

**Antibodies.** PolyADP-ribose polymerase (PARP) (9542), cleaved PARP (5625), caspase-3 (14220), cleaved caspase-3 (9664), caspase-8 (4790), cleaved caspase-8 (9748), β-actin (3700), RIP1 (3493), phospho-RIP1 (65746), and anti-rabbit IgG, HRP-linked secondary antibody (7074) were purchased from Cell Signaling (Danvers, MA). Gasdermin D domain-containing protein 1 (GSDMDC1) (NBP2-33422) polyclonal, cathepsin L polyclonal (AF952), and SERPINB3/4 monoclonal antibodies (NBP2-45788) were purchased from Novus Biologicals (Centennial, CO). Gasdermin E (GSDME) (ab215191) was purchased from Abcam (Cambridge, MA). Monoclonal anti-cathepsin B Antibody (H-5; sc-365558) and mouse-IgGκ BP-HRP-linked secondary antibody (sc-516102) was purchased from Santa Cruz Biotechnology (Santa Cruz, CA). SERPINB3 (8H11), SERPINB4 (10C12) monoclonal antibodies, and SERPINB3/B4 polyclonal were used as previously described[124]. All primary antibodies were used at 1:1000, except the SERPINB3/B4 polyclonal, and secondary antibodies were used at 1:2000 dilution.

**Cell culture.** Normocin (ant-nr-1) and zeocin (ant-zn-1) were purchased from InvivoGen (San Diego, CA). Fetal bovine serum (FBS), Iscove's Modified Dulbecco's Medium (IMDM) (12440079), Dulbecco's modified essential medium (DMEM) (11965126), RPMI 1640 (11875093), L-glutamine (25030081), Opti-MEM™ (31985070), sodium pyruvate (11360070), GlutaMax™ (35050061), Dulbecco's phosphate-buffered saline (DPBS) without magnesium and calcium (14190136) and DPBS with magnesium and calcium (14040117) were purchased from Thermo Fisher Scientific (Waltham, MA). Penicillin-streptomycin solution (25-512) and N-2-hydroxyethylpiperazine-N'-2-ethanesulphonic acid (HEPES) (25-534) were purchased from Genesee Scientific (El Cajon, CA). StableCell™ trypsin solution (T2610) was purchased from Sigma Aldrich (St. Louis, MO). Dimethyl sulfoxide (DMSO) (sc-358801) was purchased from Santa Cruz Biotechnology.

Except where indicated, all cell lines were obtained from the American Type Culture Collection (ATCC, Manassas, VA). The human squamous cell carcinoma epithelial cell lines HT3$^{B3-WT}$, HT3$^{B3-KO}$, SW756$^{B3-WT}$, and SW756$^{B3-KO}$ were generated by CRISPR/Cas9 as previously described[61]. HT3 derived lines were maintained in complete IMDM media (containing 15% FBS and 2 mM L-glutamine) and sub-cultured every 3–5 days. SW756 derived lines were maintained in complete DMEM media (containing 10% FBS and 2 mM sodium pyruvate) and sub-cultured every 2–3 days. The CRISPR-generated cell lines were assessed for authenticity using STR profiling (ATCC). The human colorectal adenocarcinoma epithelial line HT29 (ATCC® HTB-38™) were maintained in complete DMEM media (containing 10% FBS and 2 mM sodium pyruvate). The cells were sub-cultured every 3–5 days. The human connective tissue fibrosarcoma epithelial cell line HT-1080 (ATCC® CCL-121™) were maintained in complete OptiMEM™ media containing 4% FBS, 20 mM HEPES, and 10 mM GlutaMax™ (Thermo Fisher Scientific). The cells were sub-cultured every 1–2 days. The human monocyte cell line THP1-HMGB1-Lucia™ (thp-gb1lc) was obtained from InvivoGen. The cells were maintained in complete RPMI 1640 media (containing 10% FBS, 25 mM HEPES, 2 mM L-Glutamine, 100 μg/ml Normocin, and 100 μg/ml of Zeocin). The cells were sub-cultured every 2–3 days. All cell lines were cultured at 37 °C in 5% CO$_2$. Except where indicated, all media contained 100 units/ml penicillin and 100 μg/ml streptomycin.

**Cell death inhibitors.** Prior to experiments in FIECSs or cell lines, a dose-response of the different inhibitors was conducted to determine effective concentrations and toxicity. The pan-cysteine protease inhibitor, E64 or E64d (Bachem, Torrance, CA) was used at a final concentration of 10 μM in human cell lines and 1 μM in FIECS. The caspase-3/7 inhibitor, DEVD-CHO (BD Biosciences), was used at a final concentration of 10 μM in all cultures. The pan-caspase inhibitor, z-VAD-fmk (Enzo Life Sciences, Farmingdale, NY) was used at a final concentration of 10 μM in human cell lines and 1 μM in FIECS. The RIP1K inhibitor, necrostatin-1 (UBPBio, Aurora, CO), was used at a final concentration of 5 μM in FIECS and 10 μM in human cell lines. All inhibitors were dissolved in the diluent, in dimethyl sulfoxide (DMSO). Unless otherwise stated, inhibitors were added to DMEM plus 20% FBS at 37 °C in 5% CO$_2$ for 1 h prior to induction of death.

**Chemicals.** (2 S,3 S)-3-[[(2 S)-1-[4-(diaminomethylideneamino)butylamino]-4-methyl-1-oxopentan-2-yl]carbamoyl]oxirane-2-carboxylic acid (E64) (A2576), ethyl (2 S,3 S)-3-[[(2 S)-4-methyl-1-(3-methylbutylamino)-1-oxopentan-2-yl]carbamoyl]oxirane-2-carboxylate (E64d) (A1903), (S,S,2 S,2'S)-N,N'-((2 S,2'S)-(hexane-1,6-diylbis(azanediyl))bis(3-oxo-1,1-diphenylpropane-3,2-diyl))bis(1-((S)-2-cyclohexyl-2-((S)-2-(methylamino)propanamido)acetyl)pyrrolidine-2-carboxamide) (BV6) (B4653), and quinolyl-valyl-O-methylaspartyl-[-2, 6-difluorophenoxy]-methyl ketone (Q-VD-OPh hydrate) (A1901) were purchased from ApexBio (Boston, MA). Staurosporine (STS; J67198) and cyclosporin A (CsA;

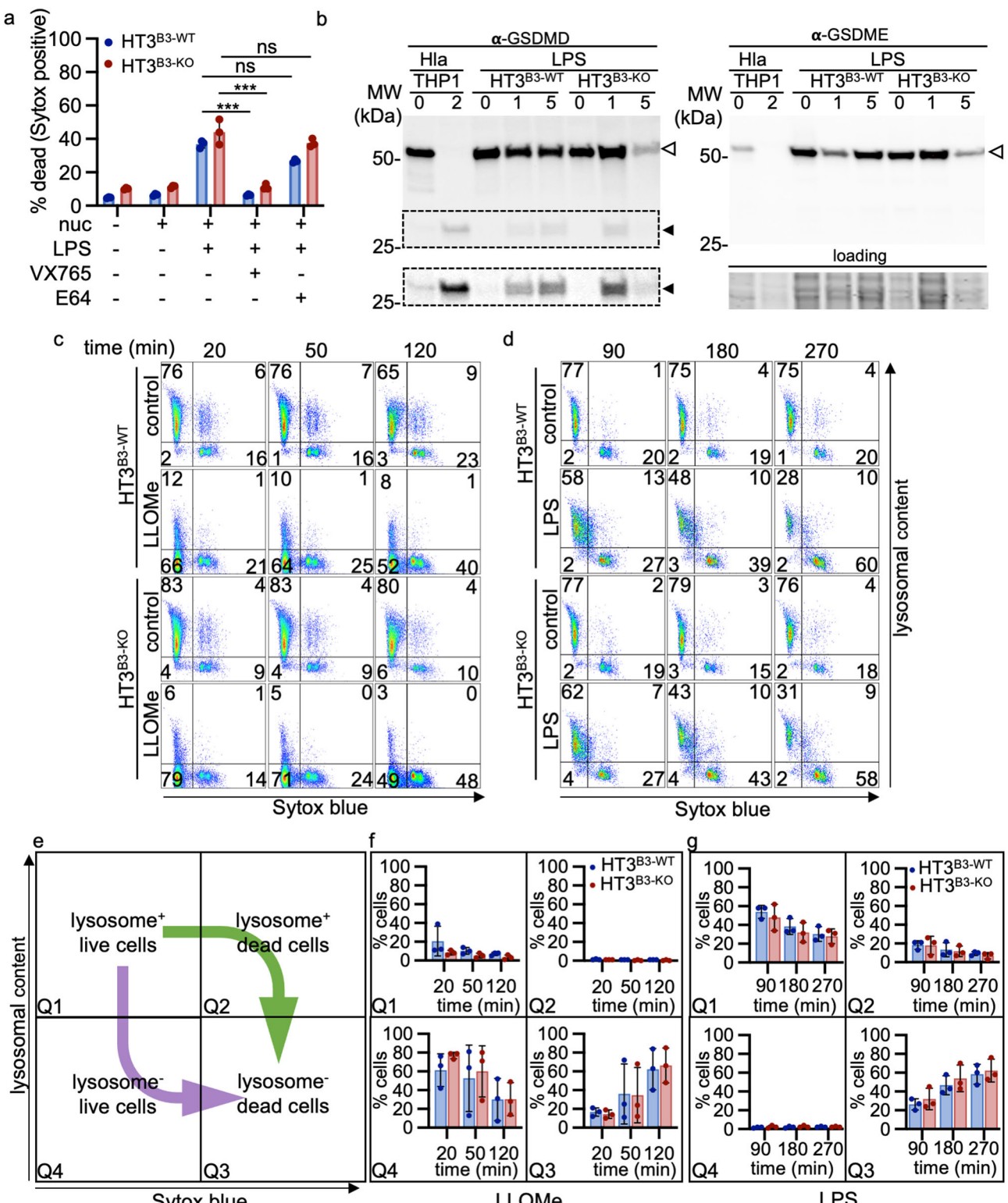

J63191) were purchased from Alfa Aesar (Ward Hill, MA) and TOCRIS Bioscience. Acetyl-Asp-Glu-Val-Asp-aldehyde (Ac-DEVD-CHO; 556465) was purchased from BD Biosciences. Z-Val-Ala-Asp (OMe)-CH₂F (z-VAD-fmk; 627610), (4-[(2 R)-2-[(1 S,3 S,5 S)-3,5-Dimethyl-2-oxocyclohexyl]-2-hydroxyethyl]piperidine-2,6-dione (cycloheximide; CHX; 239765), digitonin (CHR103) and α-hemolysin from *Staphylococcus aureus* (Hla; H9395) were purchased from Millipore Sigma (St. Louis, MO). Belnacasan (VX765) (S2228), 2-[1-[4-[2-(4-Chlorophenoxy)acetyl]-1-piper-azinyl]ethyl]-3-(2-ethoxyphenyl)-4(3H)-quinazolinone (erastin) (S7242), 7-Cl-O-5-(1H-indol-3-ylmethyl)-3-methyl-2-sulfanylideneimidazolidin-4-one (nec-1s) (S8641), and 3-Amino-4-cyclohexylaminobenzoic acid ethyl ester (ferrostatin-1) (S7243) were purchased from Selleckchem (Houston, TX). Ammonium iron (III)

citrate (FAC) (61221) was purchased from ACROS Organics (Fair Lawn, NJ). Standard LPS, *Escherichia coli* K12 (LPS-EK; tlrl-eklps) was purchased from Invivogen. Tumor Necrosis Factor alpha (TNFα) (300-01 A) was purchased from Peprotech (Rocky Hill, NJ). H-Leu-Leu-OMe•HBr (LLOMe) (4000725) was purchased from Bachem. Cathepsin L Inhibitor, CAA0225, (21-950-21MG) and hydrogen peroxide (H₂O₂) (H325) were purchased from Fisher Scientific (Pittsburgh, PA).

**Fetal intestinal explant cultures (FIECs)**. Mouse experiments were approved by the University of Pittsburgh and Washington University Animal Care and Use

**Fig. 10 Cytosolic LPS induces pyroptosis but not LMP in both HT3$^{B3-WT}$ and HT3$^{B3-KO}$ cell lines. a** HT3$^{B3-WT}$ (blue) or HT3$^{B3-KO}$ (red) cells were subjected to nucleofection (nuc, +) in the absence (−) or presence (+) of 5 ug/ml LPS-EK (LPS), caspase-1 inhibitor, VX765, or lysosomal cysteine protease inhibitor, E64. An aliquot of cells were also not nucleofected (nuc, −). After 3 h, cells were stained with SG and Hoescht 33342 and imaged. Quantification: (# Sytox positive nuclei/# of blue nuclei) × 100. Means ± SD were compared using a two-tailed $t$-test (*$P < 0.05$). **b** Immunoblot of gasdermin D (GSDMD) and gasdermin E (GSDME) cleavage in THP1 cells treated with α-hemolysin (HlA); positive control. HT3$^{B3-WT}$/HT3$^{B3-KO}$ were nucleofected with 0, 1, or 5 μg of LPS and left to recover for 1 h. Open arrowheads: full-length GSDMD and GSDME based on molecular mass. Black arrowheads: cleaved GSDMD. Dashed box: area contrast enhanced for clarity. Note, the positive control for GSDME cleavage is provided in Fig. S18b. **c, d** Flow cytometry analysis of HT3$^{B3-WT}$/HT3$^{B3-KO}$ cells treated with a lysosomotropic agent, LLOMe (**c**) or 1 μg/ml LPS (**d**). Cells were stained with the lysosomotropic dye, acridine orange (Y-axis), and the cell viability dye, Sytox™ blue (X-axis). Lines indicate the threshold fluorescence levels (gate) to determine each quadrant. Numbers indicate cell percentage in each quadrant. **e** Schematic representation of each quadrant. Quadrant 1 (Q1) contained live cells positive for lysosomal staining (acridine orange positive, Sytox blue negative). Quadrant 2 (Q2) contained dead cells with positive lysosomal staining (acridine orange positive, Sytox blue positive). Quadrant 3 (Q3) contained dead cells negative for lysosomal staining (acridine orange negative, Sytox blue positive). Quadrant 4 (Q4) contained live cells negative for lysosomal staining (acridine orange negative, Sytox blue negative). Arrows indicate the timing of fluorescence loss for different cell death pathways. If the lysosomal loss occurred prior to plasma membrane permeabilization, then the percentage of cells will increase from Q1 to Q4 to Q3 over time. If the lysosomal loss occurred after plasma membrane loss, then the percentage of cells will increase from Q1 to Q2 to Q3 over time. **f, g** Graphical representation of the average of three separate experiments indicating the percentage of cells treated with LLOMe (**f**) or nucleofected LPS (**g**) in HT3$^{B3-WT}$ (blue) and HT3$^{B3-KO}$ (red) cells. Error bars represent means ± SD. Uncropped immunoblots can be found in Supplementary Fig. 21.

---

Committees protocols 12020207 and 19-102, respectively. FIECs were generated as described in ref. [56]. Embryos from timed pregnant wild-type Balb/C females (mSerpinb3$^{+/+}$) and Serpinb3a homozygous null (mSerpinb3a$^{−/−}$) mice were harvested on day E13.5–E15.5 and the intestines removed[57]. Fetal intestinal tissue was cut into ~1 mm sections and then dissected to expose the epithelial layer (Fig. S2a). Intestinal sections were cultured in 200 μl of Dulbecco's Modified Eagle Medium (DMEM) (Thermo Fisher Scientific) plus 20% FBS using fibronectin (BD Biosciences) coated glass bottom 35 mm dishes (MatTek Corporation, Ashland, MA) at 37 °C in 5% CO$_2$ for 18 h. An additional 2 ml of prewarmed DMEM (Thermo Fisher Scientific) plus 20% FBS was added to the dishes, and the FIECs were cultured at 37 °C in 5% CO$_2$ for a further 7-11 days with daily media changes.

**Fluorescent staining reagents.** Invitrogen™ H33342 (H3570), propidium iodide (PI; P3566), Sytox™ blue (S11348), Sytox™ red (S34859), Sytox™ green (S7020), Sytox™ orange (S11368), Lysotracker™ deep red (L12492), Lysotracker™ red DND-99 (L7528), Mitotracker™ red (M22425), Mitotracker™ deep red FM (M22426), CM-H2-DCFDA (C6827), BODIPY™ 581/591 C11 (D3861), Cascade Blue™ dextran 3 kDa (D7132), Alexa Fluor™ 488 dextran 10 kDa (D22910), tetramethylrhodamine (TMR) dextrans 40 kDa (D1842) and 70 kDa (D1818), and Texas red labeled dextrans 3 kDa (D3328), 10 kDa (D1863), 40 kDa (D1829), and 70 kDa (D1864) were purchased from Thermo Fisher Scientific. Acridine orange (AO) (40039) and NucView® 530 (10406) were purchased from Biotium. FITC-annexin V was purchased from Southern Biotech or FITC-annexin V apoptosis detection kit (556547) was purchased from BD Pharmingen.

**Light microscopy.** Multi-well plate assays were detected for cell death by staining with the indicated Sytox™ dyes using a Biotek Cytation 5 multimode imaging plate reader (Agilent Technologies) fitted with a Biotek BioSpa8 live-cell incubator (Agilent Technologies) at the indicated excitation and emission wavelengths at 37 °C in 5% CO$_2$. The total number of cells were calculated using either H33342 staining or high contrast brightfield imaging as described in https://www.biotek.com/resources/application-notes/live-cell-imaging-of-apoptosis-and-necrosis/ using a 4x high contrast objective. The percentage of dead cells was calculated as the number of Sytox™ positive cells/the total number of cells × 100 using Biotek Gen5 software (Agilent technologies).

Widefield fluorescence imaging was performed using a Nikon Eclipse Ti inverted microscope fitted with DIC optics and either a 20 × 0.5 N.A. air objective or a 40 × 1.3 N.A. oil objective, using a 100 W halogen fluorescent light source and DAPI HC HISN Zero Shift, GFP HC HISN Zero Shift, DS Red HC HISN Zero Shift and CY5 HYQ filter sets (Nikon Instruments, Inc, Melville, NY). Images were acquired using a Retiga 2000R camera (Qimaging, Surrey, Canada) and Volocity software (v6.3.5, Quorum, Canada).

Spinning disk confocal images was performed using a Nikon Eclipse Ti inverted microscope fitted with a Nipkow spinning disk and 488 nm and 568 nm excitation laser lines. Images were acquired using a CoolSnap HQ camera (Photometrics, Tucson, AZ) and NIS elements AR software (Nikon Instruments).

Line scanning confocal images were collected using a Leica SP8 tandem scanning confocal fitted with 20 × 0.4 N.A. air and 40 × 1.3 N.A. oil objectives using a 488 nm argon gas laser line and 405, 561, and 633 nm diode laser lines or a Leica SP8 tandem scanning confocal fitted with 25 × 0.95 N.A. water, 40 × 1.3 N.A. and 63 × 1.4 N.A. oil objectives, a supercontinuum white light laser and 405 nm diode laser lines. Live-cell imaging was performed at 37 °C in 5% CO$_2$ using either a stage top incubator with a lens heater (Tokai Hit, Shizuoka, Japan) or a cage incubator (OkoLabs, PA). Images were collected by HyD avalanche detectors using

LASX software (Leica Microsystems, Buffalo Grove, IL). All images were visualized and analyzed using Volocity software (v6.3.5, Quorum) or LASX (v6.3, Leica Microsystems).

**Western blotting.** Pierce™ Halt™ protease and phosphatase inhibitor cocktail (78446), Coomassie Plus (Bradford) assay reagent (23238), and BCA protein assay kit (23227) was purchased from Thermo Fisher Scientific. RIPA Buffer (786-490) was purchased from G-Biosciences (St. Louis, MO). AnykD™ Criterion™ TGX Stain-Free™ precast gels (5678123, 5678124, and 5678125), 10x Tris/Glycine Buffer (1610771), 10x tris/glycine/SDS buffer (1610772), Precision Plus Protein™ all blue prestained protein standard (1610373), Clarity Max™ Western ECL substrate (1705062), and Clarity™ Western ECL substrate (1705061) were purchased from Bio-Rad (Hercules, CA). Amersham™ Protran™ 0.2 μm nitrocellulose (106000011) was purchased from Cytiva (Marlborough, MA). 2-Mercaptoethanol (773648) was purchased from MP Biomedicals (Irvine, CA). Methanol (A412SK-4) and Tween® 20 (BP337-500) were purchased from Thermo Fisher Scientific.

For analysis of C. elegans proteins, ~100 transgenic srp-6(ok319);vkEx [P$_{srp-6}$mammalianSERPIN::GFP; P$_{myo-2}$mCherry] C. elegans were placed directly into 2x Laemmli Sample Buffer (62.5 mM Tris-HCl, pH 6.8, 25% (w/v) glycerol, 2% SDS, 0.01% bromophenol blue, 355 mM 2-mercaptoethanol; Bio-Rad) and boiled for 20 min. Samples were centrifuged at 500×g for 2 min and the supernatant separated by AnyKD™ gradient SDS-PAGE (Bio-Rad) under reducing conditions, electrotransferred via wet transfer at 100 V in Tris/Glycine with methanol (25 mM Tris, 192 mM glycine, 20% v/v methanol, pH 8.3) onto nitrocellulose (Cytiva) and blocked in 5% nonfat milk in TBST (50 mM Tris-HCl, 150 mM NaCl, 0.1% Tween® 20, pH 7.5) for 1 h at room temperature or overnight at 4 °C. Proteins were detected using an anti-GFP polyclonal antibody (1:1000 dilution, Santa Cruz Biotechnology) and tubulin detected using an anti-tubulin monoclonal antibody (1:2500 dilution, Santa Cruz Biotechnology) was used as a loading control. The primary antibody binding was detected with peroxidase-conjugated bovine anti-mouse or peroxidase-conjugated anti-rabbit secondary antibodies (Santa Cruz Biotechnology). Signal was detected using Super Signal West Pico Chemiluminescent Substrate (Thermo Fisher Scientific) and autoradiography.

For analysis of mammalian cellular proteins, media was aspirated from the plates and they were rinsed once with DPBS. Plates were placed on ice and the cells were disrupted by adding RIPA lysis buffer supplemented with Pierce™ Halt™ protease and phosphatase inhibitors for 10 min. Cell lysates were transferred to microcentrifuge tubes and placed at −80 °C. Thawed samples were centrifuged at 21,000 × g. Supernatants were then transferred to new microcentrifuge tubes and the protein concentration was determined by Pierce™ BCA assay. Equal amounts of protein (~30 μg of total protein) were mixed with 2x Laemmli buffer with 2-mercaptoethanol and heated to 100 °C for 5 min. Protein samples were centrifuged at 15,000 × g for 2 min, analyzed by PAGE, and transferred onto nitrocellulose (Cytiva) as described above. Nitrocellulose membranes were blocked in 5% nonfat milk in TBST for 1 h at room temperature or overnight at 4 °C. Proteins were detected using the specified primary antibody with the appropriate secondary antibody at the indicated dilutions and visualized using Clarity chemiluminescent or Clarity-max chemiluminescent substrate (Bio-Rad) using a Chemidoc MP (Bio-Rad).

**Transmission electron microscopy.** Cells were incubated in six-well plates for 3 days. On the day of the experiment, cells were treated with death inducers for the indicated amount of time. For all cell death assays, except pyroptosis, cells were detached from the plate, centrifuged at 300×g for 4 min, and fixed in 2%

paraformaldehyde/2.5% glutaraldehyde (Polysciences Inc., Warrington, PA) in 100 mM sodium cacodylate buffer, pH 7.2 for 1 h at room temperature. Sample processing was performed by the Washington University School of Medicine Molecular Microbiology Imaging and Washington University Center for Cellular Imaging facilities. Samples were washed in sodium cacodylate buffer at room temperature and post-fixed in 1% osmium tetroxide (Polysciences) for 1 h. Samples were rinsed in ddH$_2$0 prior to en bloc staining with 1% aqueous uranyl acetate (Ted Pella, Inc., Redding, CA) for 1 h. Following several rinses in ddH$_2$0, samples were dehydrated in a graded series of ethanol solutions and embedded in Eponate 12 resin (Ted Pella). Sections were cut with a Leica Ultracut UCT ultramicrotome (Leica Microsystems), stained with uranyl acetate and lead citrate, and viewed on a JEOL 1200 EX (JEOL USA, Inc., Peabody, MA) equipped with an AMT 8-megapixel digital camera (Advanced Microscopy Techniques, Woburn, MA) or JEM-1400 TEM (JEOL USA) equipped with an AMT XR111 high-speed 4k × 2k pixel phosphor-scintillated 12-bit CCD camera (Advanced Microscopy Techniques). Images were captured using AMT Image Capture Engine software (Advanced Microscopy Techniques).

### Cell death assays

*Apoptosis*. FIECs were washed twice in ice-cold DPBS, once in cold annexin V binding buffer. FITC-annexin V was diluted 1:10 in annexin V binding buffer and incubated with FIECs for 15 min at 4 °C in the dark. Cells were stained with 500 nM PI and 2 µg/ml H33342 for 5 min in the dark prior to visualization. For necroptosis assays, FIECs were stained with 500 nM PI and 2 µg/ml H33342 for 5 min in the dark prior to visualization.

Apoptosis was induced in HT3$^{B3-WT}$ and HT3$^{B3-KO}$ cells using 5 µM STS for 16-20 h at 37 °C in 5% CO$_2$. Cell death was measured using 500 nM Sytox™ orange (Thermo Fisher Scientific) and imaged on the BioTek Cytation™ 5. Annexin V staining was measured using the FITC-annexin V apoptosis detection kit (BD Pharmingen, San Jose, CA) per the manufacturer's instructions. Cells were imaged by confocal microscopy. For determination of caspase-3/7 activation, cells were stained with 1 µM Nucview™ 530 (Biotium, Fremont, CA) and imaged on the BioTek Cytation™ 5.

*Ferroptosis*. Cells were stimulated using 1 µM FAC and 10 µM erastin for 10–12 h; notably, ferroptosis was only able to be induced when the cells were ≤50% confluence. Cell death was measured by 500 nM Sytox™ orange (Thermo Fisher Scientific) uptake using a Biotek Cytation 5 multimode imaging plate reader. For measurement of ROS production and mitochondrial function during ferroptosis, 100,000 SW756 cells were seeded into a well of a 35 mm optical-bottom dish. Prior to imaging, cells were stained with 10 µM H2-DCFDA and 1 µM MitoTracker™ red (Thermo Fisher Scientific) for 1 h at 37 °C in 5% CO$_2$. Cells were incubated in their respective media with 10 µM erastin for 10–12 h in the presence of 5 nM Sytox™ red (Thermo Fisher Scientific). Images were collected by line scanning confocal microscopy.

Hypotonic stress was performed in *C. elegans* as described in ref. [41]. The experiments were repeated three times and the average percentage recorded. Statistical significance was calculated using a two-tailed *t*-test.

For cell lines, either 15,000 cells/well were seeded into a Nunc™ optical-bottom 96-well plate (Thermo Fisher Scientific), 150,000 cells/well were seeded into a 24-well plate (Genesee Scientific), or 250,000 cells seeded into an imaging dish (Cellvis, Mountain View, CA) ~24 h before prior to assay. For FIECs, the medium was aspirated and replaced with prewarmed to 37 °C 100% or 10–25% DPBS as indicated, with magnesium and calcium added for the indicated time points. Assays performed with or without inhibitors at the designated concentrations were preincubated in the appropriate medium for 30–60 min prior to the assay.

*MPT-DN*. Death was induced in HT3$^{B3-WT}$ and HT3$^{B3-KO}$ cells using 1–5 mM H$_2$O$_2$ for 8 h[72]. Cell death was measured using 100 nM Sytox™ green imaged on the BioTek Cytation™ 5 with brightfield and GFP (Ex 488 nm/Em 535 nm) LED cubes. To measure mitochondrial function, cells were incubated with H$_2$O$_2$ inducers for 16 h. Prior to imaging, cells were incubated for 30 min with 500 nM MitoTracker™ Red and 5 nM Sytox™ red (Thermo Fisher Scientific) and imaged using confocal microscopy.

*Necroptosis*. Death was induced in FIECs by incubation with 1 µM z-VAD-fmk and 1 µM staurosporine (TOCRIS Bioscience) in DMEM plus 20% FBS at 37 °C in 5% CO$_2$ for 8 h[59]. FIECs were stained with 500 nM PI and 2 µg/ml H33342 for 5 min in the dark prior to visualization by confocal microscopy.

RIP1K-dependent necroptosis was induced in HT3$^{B3-WT}$ and HT3$^{B3-KO}$ cells using by pretreating cells with 100 nM BV6, 500 nM qVD-OPh, and 1 µM cyclohexamide at 37 °C in 5% CO$_2$[125]. After 1 h incubation, human TNFα was added to a final concentration of 10 ng/ml and incubated for 12–16 h at 37 °C in 5% CO$_2$. Cell death was measured by uptake of 100 nM Sytox™ green imaged on the BioTek Cytation™ 5.

*Pyroptosis*. Death was attempted to be induced by treating HT3$^{B3-WT}$ and HT3$^{B3-KO}$ with either nigericin (4312/10; Tocris Bioscience, Bristol, UK) or α-hemolysin (Hla; H9395; Millipore Sigma) at 37 °C in 5% CO$_2$ at the indicated concentrations and time points. Pyroptosis was induced in HT3 cells after detaching them from culture

plates using trypsin-EDTA. Nucleofection was performed according to the manufacturer's protocol (Lonza, Valparaiso, IN)[94]. Approximately 500,000 cells per treatment were resuspended in 100 µL of supplemented Lonza SE cell line Nucleofector™ solution and nucleofected with 5 µg of LPS-EK using the HeLa protocol on the Lonza 4D-Nucleofector™ System. Cells were resuspended in 1 ml of OptiMEM™ without antibiotics and incubated in a humidified tissue culture incubator. At the indicated times, an aliquot of cells was removed and either measured by flow cytometry, lysed for subsequent immunoblotting, or fixed for analysis by transmission electron microscopy.

### Lysosome content and quantification

For single time point measurement of lysosomes via fluorescent imaging, FIECs, HT3, or SW756 cells/well were seeded into 35 mm optical-bottom dishes. On the day of the experiment, cells were incubated in their respective media with their respective death inducers for the indicated amount of time. After treatment, cells were stained using H33342, Sytox™ dyes (5 nM) and Lysotracker™ deep red (200 nM) were added and incubated for 30 min prior to imaging using light microscopy. Widefield images were analyzed for lysosomal area per cell and confocal images were analyzed for lysosomal count per cellular area using threshold-based object identification of the different fluorophores in Volocity quantification (v6.3.5, Quorum).

For measurement of lysosomes via flow cytometry, HT3$^{B3-WT}$ and HT3$^{B3-KO}$ cells were detached using trypsin and treated with the indicated treatment. Cells were subsequently stained with 200 ng/ml acridine orange and 1 µM Sytox™ blue (Thermo Fisher Scientific) for 30 min and then analyzed with a Miltenyi MACSQuant® Analyzer 10.

For the measurement of lysosomal cysteine protease activity in cell lysates, after cell growth and indicated treatments, cells were placed on ice and the medium was aspirated. Cells were then washed with 1 ml of cold DPBS with calcium and magnesium (Thermo Fisher Scientific; 14040117) prior to addition of the indicated concentrations of digitonin (Millipore Sigma; CHR103) diluted in DPBS. Cells were then incubated on ice for 15 minutes. The digitonin solution was then assayed for protein and cysteine protease content as described above, or 50 µl of the extract was then mixed with equal amounts of cathepsin reaction buffer (50 mM sodium acetate, pH 5.5; 4 mM EDTA, 8 mM dithiothreitol) in a 96 well black-walled microtiter plate and incubated for 5 minutes at 30 °C. The cysteine protease substrate, zFR-R110 (Fisher Scientific; PKCA70710209), was then added to a final concentration of 5 µM and fluorescence intensity was measured using a Varioskan lux (Thermo Fisher Scientific) fluorescence plate reader at Ex 498 nm, Em 525 nm every 30 seconds for 30 minutes. The cysteine proteolytic activity (RFU) over time was calculated using SkanIT RE software (v4.1; Thermo Scientific).

### Lysosomal membrane permeability measurements using fluorescently labeled dextrans

The *C. elegans* protocol was modified from that described earlier in refs. [41,126]. Young adult N2 or *srp-6(ok319)* animals (*n* ≥ 200) were incubated in PBS containing 1 mg/ml of Texas red labeled dextrans 3, 10, 40, or 70 kDa and heat-killed OP50 at room temperature overnight in the dark. Animals were washed once with PBS and then placed onto a seeded NGM plate for 4 h to allow trafficking of the fluorescently labeled dextrans through the endolysosomal system and to chase the excess dextran from the intestinal lumen. Animals were collected from the plate in 1 ml of PBS in an Eppendorf tube and allowed to settle at 25 °C for 5–10 min. Animals were washed with 1 ml of ddH$_2$O and allowed to settle again. Excess ddH$_2$O was aspirated from above the animals with a remaining volume of 50–100 µl. Animals were loaded into an L4-YA vivoChip®-2x (Newormics, Austin, TX) using a 5 ml syringe. Animals were guided into the microfluidic channels using ddH$_2$O and the automatic function of the *vivo*Cube following the manufacturer's instructions (Newormics). Animals (≥10 nematodes) were visualized using a 40×1.3 N.A. PlanApo oil objective on an SP8X tandem scanning confocal microscope at an excitation wavelength of 594 nm and emission detection window of 605–640 nm with a HyD avalanche detector, every 15 min for 4 h over ≥20 z-planes. Images were rendered with LASX and lysosomal fluorescence was analyzed using Volocity software (v6.3.5, Quorum).

For cell lines, the protocol was adapted from that previously described in refs. [53,127]. Approximately 200,000 HT3$^{B3-WT}$ and HT3$^{B3-KO}$ or SW756$^{B3-WT}$ and SW756$^{B3-KO}$ cells were seeded into an optical-bottom imaging dish (Cellvis Mountain View, CA) and incubated at 37 °C in 5% CO$_2$ until ~50–80% confluency. Cells were treated with 10 µM of each of the following dextrans: 3 kDa Cascade Blue™ dextran, 10 kDa Alexa Fluor™ 488 dextran, 40 kDa tetramethylrhodamine (TMR) dextran, and 70 kDa Texas red labeled dextran, for 16 h at 37 °C in 5% CO$_2$. After labeling, cell lines were washed with 37 °C DPBS and chased for 4 h with the appropriate medium. Cells were exposed to different cell death inducers were visualized in XYZ planes over time by live-cell confocal microscopy using a Leica SP8X confocal microscope fitted with a 63×1.4 N.A. PlanApo oil objective at 37 °C in 5% CO$_2$. The 3 kDa Cascade Blue™ dextran was excited at 405 nm and the emission was detected at 420–460 nm, 10 kDa Alexa Fluor™ 488 dextran was excited at 488 nm and the emission detected at 500–540 nm, 40 kDa TMR dextran was excited at 561 nm and the emission detected at 570–585 nm and 70 kDa Texas red labeled dextran was excited at 594 nm and the emission detected at 605–635 nm. Cell plasma membrane permeability was measured using Sytox™ deep red using 647 nm excitation and 660–700 nm emission. Images were rendered with

LASX (Leica Microsystems) and lysosomal fluorescence was analyzed using Volocity software (v6.3.5, Quorum).

Lysosomes in FIECs were labeled with 100 µg/ml TMR- or Alexa Fluor® 647-labeled 10 kDa dextran in DMEM plus 20% FBS at 37 °C in 5% $CO_2$. After 18 h, FIECs were washed three times with 2 ml of prewarmed DPBS then incubated in DMEM plus 20% FBS at 37 °C in 5% $CO_2$ for at least 4 h prior to visualization or experimentation to chase excess label.

LMP analysis of fluorescently labeled lysosome images had their background subtracted and noise reduced using a fine filter, if necessary. Three-dimensional z-planes were merged by maximum intensity projection using Volocity (v6.3.5, Quorum). Merged images were analyzed using Volocity Quantification (v6.3.5, Quorum) to detect the number of lysosomes per field of view. The lysosomal number in each of the channels representing the different fluorophores was calculated using intensity-based object identification. Large-sized identified objects were excluded from the lysosome number (>25 µm). The lysosomal count (LC) in each field was normalized to the LC at time 0 to generate a relative number of lysosomes ($LC_{t0}/LC_t$). LMP changes for the different conditions, cell lines, C. elegans, and dextrans were compared by analyzing the area under the curve (AUC) for each individual time series and compared by means of one-way ANOVA analysis with Tukey multiple comparisons using GraphPad Prism (v8.4.2; GraphPad Software, LLC).

**Generation of cathepsin L knockout cell lines.** Cathepsin L (CTSL) knockouts were generated in both SW756[B3-WT] and SW756[B3-KO] using the lentiGuide-Puro system as previously described in ref. [128]. Briefly, guide RNAs for cathepsin L (TGAGGAATCCTATCCATATG) was cloned into the BsmB1 restriction enzyme site in the lentiGuide-Puro vector (a gift from Feng Zhang; Addgene plasmid # 52963; http://n2t.net/addgene:52963; RRID:Addgene_52963). Lentivirus producing the guide RNA and CAS9 protein from lentiCAS9-Blast (a gift from Feng Zhang; Addgene plasmid # 52962; http://n2t.net/addgene:52962; RRID:Addgene_52962) was made in HEK293T cells with packaging vectors psPAX2 (a gift from Didier Trono; Addgene plasmid # 12260; http://n2t.net/addgene:12260; RRID:Addgene_12260) and pCMV-VSV-G (a gift from Bob Weinberg; Addgene plasmid # 8454; http://n2t.net/addgene:8454; RRID:Addgene_8454). SW756[B3-WT] and SW756[B3-KO] cells lines were transduced with the virus and cultured in DMEM + 10% FBS under blasticidin and puromycin selection. The genomes of these cells were then examined for indels in the CTSL gene by NGS sequencing by the Genome Engineering and iPSC Center at the Washington University in St. Louis, School of Medicine (Table S1), and CTSL protein levels were examined by western blotting[114].

**Statistics and reproducibility.** All experiments were repeated multiple (≥3 replicates unless stated) times to ensure reproducibility. For multi-well plate assays, replicates were also contained within the plate to determine well to well variability. Statistics were performed using either a two-tailed t-test, one-way ANOVA, or two-way ANOVA with Tukey multiple comparisons using GraphPad Prism (v8.4.2; GraphPad Software, LLC).

**Reporting summary.** Further information on research design is available in the Nature Research Reporting Summary linked to this article.

## Data availability

Original imaging data files can be found at https://data.mendeley.com/datasets indicated at the end of each appropriate figure legend. Individual chart data can be found at https://data.mendeley.com/datasets/h4n6sz9334/1. Uncropped blots and gel images can be found as Supplementary Fig. 21.

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

## Acknowledgements

This work was supported by NIH grants R01DK104946 (G.A.S.), R01DK114047 (C.J.L. and G.A.S), R01DK118568 (M.G.), K08CA237822 (S.M.), K08AI144033 (A.O.), The Society for Pediatric Research Physician Scientist Award (A.O.), American Society for Clinical Oncology (ASCO) Career Development Award (S.M.), The Children's Discovery Institute of St. Louis Children's Hospital Foundation (C.J.L., S.C.P., and M.G.). J.A.J.F is supported by the Chan Zuckerberg Initiative as an Imaging Scientist (2020-225726). Washington University Center for Cellular Imaging (WUCCI) is supported by Washington University School of Medicine, The Children's Discovery Institute of Washington University and St. Louis Children's Hospital (CDI-CORE-2015-505 and CDI-CORE-2019-813), and the Foundation for Barnes-Jewish Hospital (3770). We would like to thank Dr's Ekkehard Weber, Martin Luther University, Germany and Stuart Kornfeld, Washington University in St. Louis, School of Medicine, USA for the kind gifts of CTSB and -D antibodies, respectively.

## Author contributions

M.G., S.M., I.E.W., B.J.T., A.O., S.C.P., G.A.S., and C.J.L. conceived and designed the experiments. M.G., S.M., B.J.T., I.E.W., J.M.L., W.E.L., O.K., M.R.C., M.T.M., S.W., L.C., Q.G., A.A., M.C.G., and C.J.L. performed the experiments. M.G., S.M., B.J.T., I.E.W., J.M.L., M.T.M., W.L.B., K.N.H., F.V.W., J.A.J.F., and C.J.L. interpreted and analyzed the data. G.A.S. and C.J.L. supervised the study and wrote the manuscript. All authors read, edited, and approved the manuscript.

## Competing interests

The authors declare no competing interests.
