## [Transparent Peer Review File · Communications Biology]

Reviewers' comments:

Reviewer #1 (Remarks to the Author):

The manuscript by Good and colleagues reports a lysosome-dependent cell death termed lysoptosis is an early eukaryotic regulated cell death program that is evolutionarily conserved, and predominates when intracellular protease-inhibitor balance is compromised. They found that intracellular lysosomal cysteine peptidase inhibitors mSerpib3a (mouse) and SERPINB3 (human) were found to impair lysoptosis in mouse fetal intestinal explant cultures (FIECs) and human tumor cell lines respectively in response to hypotonic stress. They further conducted experiments to determine whether lysosomal membrane permeabilization compromises cell survival or is an onset of postmortem autolysis in response to different stress and cell death inducers. Specifically, they found that lysosomal membrane permeabilization is required to induce lysoptosis-like death in mSerpib3a^{-/-} FIECs in response to hypotonic stress. Furthermore, apoptosis inducer staurosporine induced lysosomal membrane permeabilization and lysoptosis-like death in SERPINB3 deficient human cervical carcinoma cell line HT3B3. They further investigated association of human SERPINB3 in various necrotic forms of cell death including mitochondrial permeability transition-driven necrosis (MPT-DN), necroptosis, ferroptosis and pyroptosis. Interestingly, the authors observed that loss of SERPINB3 in human tumor cell HT3B3 accentuated lysosomal membrane permeabilization and lysoptosis-like cell death in response to inducers of necroptosis and MPT-DN but not in response to inducers of pyroptosis and ferroptosis. Lastly, the current study provides insights into the physiological relevance of the lysoptosis pathway using a neonatal mouse model of intestinal injury. The data suggested that mSerpib3a has a cytoprotective role in the intestinal epithelium of mice. In conclusion, the author demonstrated the role of SERPINB3 as a pro survival factor that protects against the toxicity of lysosomal proteases in higher eukaryotes.

Major comments:

1. One of the major aims of the manuscript is to determine whether lysosomal membrane permeabilization (LMP) compromises cell survival or is an onset of postmortem autolysis in response to different stress and cell death inducers. The authors have comprehensively utilised different molecular weight response to dextrans and lysotracker dye to determine LMP, however an additional evidence is to do flow cytometry analysis (similar to Fig 9E-G) using acridine orange in response to inducers of necroptosis and MPT-DN. This will answer the question whether LMP is preceding or succeeding cell death.
2. Since Clade B Serpins namely SERPINB3 and mSerpib3a are known to inhibit cysteine proteases including cathepsins K, L, S and V; immunoblots against the mentioned cathepsins will enhance support to the major claim of this manuscript.
3. The authors propose that loss of SERPINB3 and mSerpib3a leads to enhanced LMP and lysoptosis in human and mouse cells in response to hypotonic stress. Additional evidence is required to support this claim. Can hypotonic stress induced lysoptosis in SERPINB3 and mSerpib3a deficient cells be rescued by using gene complementation of SERPINB3 and mSerpib3a in SERPINB3 and mSerpib3a deficient cells.
4. Please quantify loss of lysosomal content and sytox uptake in Figure 1 (C-L) similar to Figure 1 (Y and Z).
5. Please provide statistical p value for following groups and figures:
 - Figure 2Y- statistics between WT and KO in E64d treatment group.
 - Figure 2NN- statistics between DMSO and Nec-1 treated WT cells.
 - Figure 3Y- statistics between DMSO and Nec-1 treated KO cells.

- Figure 8B- statistics between Nec-1 and E64d treated KO cells.
- Figure 8H- statistics between WT and KO in T+S+Q+C treated group.
- Figure 9A- statistics between LPS nuc and LPS nuc+ E64 treated WT cells.
- Figure 9A- statistics between LPS nuc and LPS nuc+ E64 treated KO cells.

These statistical values are important because the authors refer to these observations to infer their conclusions in the main text.

6. Following blots need to be revised:

- Figure 5B- uneven loading control β -actin e.g. 16 mins WT and KO lysate lack actin band. Please revise the blot.
- Figure 8C- uneven loading control β -actin e.g. T+S+Q+C treated WT and KO lysate have reduced β -actin signal. Please revise the blot. In addition, the positive control HT29 cells do not show a robust pRIPK1 band. Please revise since it is not convincing that the experimental conditions are working.
- Figure 9B- Please add a loading control below the GSDMD blot similar to GSMDE blot. In addition, please revise the GSDMD blot due to visibly less protein in LPS nucleofection treated KO 5h sample. Alternatively, please provide detail reason in the discussion. In addition, please revise the loading blot below the GSDME blot due to uneven protein samples e.g. Hla treated THP1 cells have no loading control bands in 2h timepoint. Furthermore, the positive control THP1 cells treated with Hla do not show a robust GSDME band. Please revise since it is not convincing that the experimental conditions are working. Previous study used glucocorticoid triamcinolone acetonide in the human T-lymphoblastic leukemia cell line CEM-C7 (Rogers, Nature Communications volume 10, Article number: 1689; 2019)
- Supplementary figure 11D- uneven loading control β -actin e.g. considerably low protein samples in the 50 mins KO lysate observed with lack of β -actin band. Please revise the blot.
- Supplementary figure 12B- Please revise the loading blot below the GSDMD and GSDME blot due to uneven protein samples e.g. Hla treated THP1 cells have no loading control bands in 2h timepoint.

7. The interpretation of Supplementary figure 11B,C (text line numbers: 411-412) is not justified. Author need to demonstrate further support by showing the time-course data of Sytox positive cells similar to the loss of TMR graph (Supplementary figure 11C).

8. In the text line number 416, the authors conclude that 'nigericin induced a lysoptosis-like but not pyroptosis.' This argument requires additional experiments listed below:

- The authors should consider running immunoblots against caspase-1 in HT3 and THP1 cell lines in response to nigericin and Hla treatments in presence and absence of inhibitors including E64d, z-VAD-fmk and YVAD-CHO.
- The authors should use caspase-1 inhibitor VX765 in nigericin and Hla treated HT3 and THP1 cell lines similar to LPS nucleofection experiment in Figure 9A. This is important because activation of the canonical NLRP3 inflammasome culminates in activation of the effector caspase-1 and protein level of pro-caspase-1 will dictate the downstream processes including gasdermin D cleavage and activation. What if HT3 cells have defect in pro-caspase-1? In contrast, during the non-canonical NLRP3 inflammasome activation LPS induced activation of caspase 4,5 which can directly cleave gasdermin D prior to activation of NLRP3 and caspase-1 (Swanson et al. Nature Reviews Immunology volume 19, pages477-489; 2019).
- The authors only show loss of 10kDa TMR -dextran, it is necessary to show data of other low to high molecular weight dextran (similar to figures: 5R-V, 6Q-U, 7M-Q, 8I-M) to further support the current argument.
- Since HT3 cells did not respond to activators (nigericin and Hla) of canonical NLRP3 inflammasome, it is intriguing to investigate whether SW756 cells respond to activators of the canonical NLRP3 inflammasome.

9. The GSDMD data observed in LPS transfected cells in Figure 9B is not convincing. In addition, TEM images doesn't showcase morphological characteristics of a pyroptotic cell (Webster et al. J Immunol, 2010, 185 (5) 2968-2979). As mentioned the immunoblot samples were collected 1h after transfection (text line 441) however, based on previous studies activation of the non-canonical NLRP3

inflammasome by cytosolic LPS requires 8-16h (Viganò et al. Nature Communications volume 6, Article number: 8761, 2015; Knodler et al. Cell Host Microbe. 2014 Aug 13; 16(2): 249–256). The author should repeat the experiment where the cells are incubated with cytosolic LPS for 16h consistent with previous literature and perform GSDMD immunoblots.

10. Please revise the conclusion in text lines (588-589) indicating that death could only be prevented by E64d whereas, there is a partial protection in presence of Nec-1 inhibitor in response to T+S+Q+C treatment (Figure 8B) suggesting that both necroptosis and lysoptosis are active cell death pathways in response to T+S+Q+C treatment in SERPIN3B null cells.

Minor comments:

1. Methods and material:

- Experimental conditions of LPS nucleofection is missing the following details:
 - o Figure 9A: the amount of LPS used for transfection and the duration of activation is missing. In addition, for ease of readers, please include the time of incubation with LPS for Figure 9B in the legend; currently it is mentioned in line 441.
 - o Figure 9C: the amount of LPS used for LPS transfection
 - o Supplementary figure 13G,H is missing the duration of LPS activation.
- Supplement material is missing experimental conditions used for activating cell lines with nigericin.

2. Multiple figure legends are missing experimental conditions and details which are highlight below:

- Figure 9 legend: missing experimental detail regarding the loading control blot below GSDME.
- Supplementary figure 11A-C: can the authors please confirm if HT3 cells were pre-treated with LPS before activating with nigericin. If yes, please add the experimental conditions used for nigericin activation.

3. Please define (+) and (-) controls in Supplementary figure 1D.

Reviewer #2 (Remarks to the Author):

In the manuscript entitled „Lysoptosis is an ancient and evolutionarily-conserved cell death pathway moderate by intracellular serpins” the authors describe a novel type of a lysosome-dependent cell death mechanism which has not been previously considered by Nomenclature Committee on Cell Death. This mechanism is called lysoptosis, and it depends on the lysosomal interplay between cathepsins and serpins. Moreover the authors demonstrate that this mechanism is evolutionarily conserved, as it was identified and examined in *C. elegans*, rodents and human cancer cell lines. Overall, the manuscript is very well-written and easy to follow. The results and methods sections are comprehensively described. All the research tools and methodology that were applied to dissect the differences between lysoptosis and other cell deaths are sufficiently justified and strongly support the research hypothesis. Although the concept of lysoptosis as a stand-alone and distinct cell death mechanism carries some risk, the Reviewer recommends this article to be published in *Communications Biology*. Although, some points should be addressed prior to publication:

1. The whole concept of lysoptosis as independent cell death machinery strongly relies on the interactions between lysosomal enzymes (mostly cathepsins) and serpins (SERPINB3 in humans). However, do authors have any suggestions which lysosomal enzymes play a dominant role in this cell death? Is it cathepsin B, L or S? And do the cathepsins have to be active in the cytoplasm to trigger cell demise, or they rather induce lysoptosis from the lysosomes? The concept of lysoptosis is very interesting, and can make a significant impact on the field, however, this work lacks a detailed mechanistic insight into interactions between SERPINB3 and lysosomal enzymes at the level of single

proteins.

2. According to the manuscript, the lack of SERPINB3 sensitizes cells for lysoptotic stimulants. Is it also true for other lysosomal proteases endogenous inhibitors, or this cell death mechanism is exclusively associate with SERPINB3? Therefore authors may consider testing this with HT3 cell lines lacking other lysosomal proteases inhibitors.

3. The main stimulant to trigger lysoptosis was "hypotonic stress". Do authors are aware of any other putative stimulants, that can trigger this cell death in HT3B3-KO system?

4. The justification for the selection of HT3 cell lines is clearly presented. HT3 is a cervical carcinoma cell line, and the over-expression of SERPINB3 correlates with poor prognosis of patients with squamous cell cancer, as such cells are more resistant to classical chemotherapy. However, it would be beneficial to the manuscript if authors could demonstrate that the lysoptosis occurs also in other cancer cell lines.

Reviewer #3 (Remarks to the Author):

In the present paper of Misty et al the authors try to establish the concept of lysoptosis in dependency of Serpin 3 expression as an independent regulated cell death mechanism.

Considering that the role of Serpin3 in the induction of lysosomal cell death is published by the same authors already in the model of c.elegans, novelty is provided by transferring the concept of lysoptosis to new model systems (mouse and cancer cells). However, the mechanism of lysoptosis requires a more careful and detailed characterisation, as it seems to differ from classical LCD, where Cathepsin act downstream of LMP. In this respect the paper completely lacks any mechanistical aspects on the role of Serpin3-deficiency in lysoptosis and results dealing with lysoptosis as a stand-alone cell death mechanism or a bystander cell death mode are not clearly seperated and mixed up.

Overall the paper lacks clarity and the manuscript needs to be more down to the point. The common theme is not very obvious and the pieces of the puzzle need to be better put in place to give the complete picture of lysoptosis in the absence of Serpin3.

Major comments

1. Hypotonic stress induces cell death in the mSerpin3a -/- cells. To further validate that observed cell death represents lysosomal cell death authors should include the analysis of the classical hallmarks of lysosomal cell death like Cathepsin translocation to the cytosol, Cathepsin activity in the cytosol. In addition the apoptotic features observed in Serpin3a wt cells are confusing and probably not helping to tell the story- at 60 min there are only 20% dead compared to 80% in the knock out.

2. Results are not always in a clear order. There is a jump between Figure 2 and 3 and back to Figure 2. Also clarity is missing at the level of the models used: results start with mouse then cell lines and mouse in the end. Further authors state that Serpin3 is expressed in intestine, lung and skin- what is the rational of taking a cervical cancer cell line as a model system? Authors should explain.

3. The canonical pathway of lysosomal cell death includes LMP- followed by the release of Cathepsins into the cytosol- cleavage of tBID by cytosolic Cathepsins- activation of Caspases, which then at least contribute to cell death execution. To exclude the involvement of Casapases authors should increase the concentration of DEVD-CHO as 1uM seems very low, a better inhibitor of Caspases is zVAD-fmk, which is also later also used in the paper. Results should be backed up by western blots demonstrating the absence of Caspase cleavage in order to establish the concept Caspase-independent lysosomal cell death in the absence of Serpin3. Serpins are described to inhibit Caspase-independent lysosomal cell death involving ROS (Liu et al 2004). Possible involvement of ROS in cell death induction should at

least be explained. Further use of RIPK1 inhibitor NEC-1 is not easy to understand as necroptosis can also occur in the absence of RIPK1. The exclusion of necroptosis requires a more careful approach.

4. In relation to point 3: Authors claim that inhibition of Cystein proteases inhibit LMP. Consequently Cathepsins would act upstream of LMP, in lysoptosis and Serpin3 would execute their inhibitory role in the lysosomes? This finding needs further investigation as it is in contrast to the current knowledge of lysosomal cell death.

5. In order to support the hypothesis of lysosomal cell death under hypotonic stress situations in Serpin3 k.o. cells, other cell death pathway should be excluded in the hypotonic stress model. The switch to an apoptotic system by treatment of cells with apoptptosis- inducer STS does not support lysoptosis as an independent regulated form of cell death. Further authors claim that STS (apoptosis inducer) / STS + zVAD also triggers lysoptosis in Serpin3 k.o. cells. This needs to be validated by more experiments (Dunai et al 2012). Further Dunai et al demonstrated that STS-induced necroptosis can be blocked by Cathepsin inhibitor Ca-074 – leaving the lysosomes in this cell death modality in a central position. How can this be combined with the proposed lysoptosis cell death model? Same is true for the HT3 model.

9. Do the authors have an explanation why there is 70 % Caspase-activity in wt HT3 cells when there is only 20 % Annexin positive cells detected (Figure 5)? There is a discrepancy in death induction between Figure 5Q and 5I-K. FACS analysis of Annexin/PI stained cells would give some clarity here. Further discrimination of secondary apoptosis and necrotic cells can be provided by the inclusion of zVAD into the Annexin/PI assay. As well as the inclusion of multiple different time points. Caspase-Blots need redoing- there is nothing loaded (at 16h cells are not secondary apoptotic yet – the absence of bands cannot be explained by loss of protein content due to advanced cell death). Also lysosomal cell death should be much better characterised. The release of the different dextran sizes is more confusing than helpful- release of LysoTracker or acridine orange which again can be analysed by FACS answers the question of LMP sufficiently. How do the authors explain that the Cathepsin inhibitor E64d has no effect on Caspase-activity in HT3B3-k.o. cells? The order of events in LCD is believed to be Cathepsins released in cytosol- cleavage of tBid- activation of Caspases.

10. What is the rational between Figure 5 and 6- the paper seems brakes here. Authors jump from trying to establish Serpin 3 as key regulator in lysoposis which is a form of regulated Caspase independent cell death- to a sensitizing role of Serpin3 in the execution of alternative regulated cell death pathways. In addition ferroptosis and pyroptosis are not blockable by the Cathepsin inhibitor E64d. This might be better placed in supplemental information.

11. In vivo experiments include NEC which is based on TLR4-induced necroptosis. The role of lysoptosis in receptor-mediated necroptosis is not addressed in the paper and evidence that this mouse model is switching to lysoptosis in the absence of serpin3 is not sufficient.

Minor points

1. Please change the title. The main focus of this paper is not that lysoptosis is an evolutionary mechanism- this is a conclusion from the previous paper published by the authors in 2007 and the current one. Could the authors also please include a section about Serpins into the introduction.

2. Labelling of the fluorescence images is not always easy to follow, for example labelling in figure 4 T-W is difficult to read.

3. First result section line 121-151 just contains supplemental information Those result should be part of a figure as they establish the system and validate the futher investigation of mouse/ human Serpin

4. Line 132: Figure 1A is not identical with Figure 1A
5. Figure 1C quantification is missing
6. Figure 3G-L should show representatives of Figure 3S
7. Fluorescence images do not always represent the according graphs.
8. Figure 2MM and NN miss legends
9. Line 276 N2 wild type *c.elegans* are not in Figure 1T

Response to referees

Reviewers' comments are numbered, our response is italicized underneath:

Reviewer #1

Major comments:

1. One of the major aims of the manuscript is to determine whether lysosomal membrane permeabilization (LMP) compromises cell survival or is an onset of postmortem autolysis in response to different stress and cell death inducers. The authors have comprehensively utilised different molecular weight response to dextrans and lysotracker dye to determine LMP, however an addition evidence is to do flow cytometry analysis (similar to Fig 9E-G) using acridine orange in response to inducers of necroptosis and MPT-DN. This will answer the question whether LMP is preceding or succeeding cell death.

While we had shown qualitatively that LMP occurs before plasma membrane permeabilization during lysoptosis (Fig. 1C-L. and Fig. S7), we had not quantified this analysis. While FACS analysis as performed in Fig 9E-G provides some context to the extent of LMP prior to plasma membrane damage, this technique does not tell you the events of individual cells in real time, as such would not necessarily answer the reviewer's question. We attempted performed real-time live-cell analyses using the varying molecular weight dextrans in the presence of Sytox FarRed to assess the temporal relationship between LMP and plasma membrane permeability. However, due to the limitations of the live-cell microscopy, our microscope and the difficulty in identifying individual cells in real time (due to cellular movement, long-term fluorescent cellular markers in a wavelength outside the dextran fluorescence, etc.), we could not obtain cellular quantification on this data. We will defer to the editors discretion on this matter.

2. Since Clade B Serpins namely SERPINB3 and mSerp3a are known to inhibit cysteine proteases including cathepsins K, L, S and V; immunoblots against the mentioned cathepsins will enhance support to the major claim of this manuscript.

*Our previous studies in *C. elegans* (Luke et al., 2007) show that cathepsins are essential for lysoptosis. However, we agree that analyses of the cathepsins in the human cell lines (the FIEC system is impractical due to the small number of cells) would provide additional evidence that LMP is crucial to death by lysoptosis. Thus, we analyzed the cathepsin content in the human tumor cell lines HT3 and SW756 by qRT-PCR and western blot analysis (supplementary figure 12) and made additional manuscript text changes to reflect this new data (pp12-13, lines 324-369 and pp22-23, lines 653-664)*

3. The authors propose that loss of SERPINB3 and mSerp3a leads to enhanced LMP and lysoptosis in human and mouse cells in response to hypotonic stress. Additional evidence is required to support this claim. Can hypotonic stress induced lysoptosis in SERPINB3 and mSerp3a deficient cells be rescued by using gene complementation of SERPINB3 and mSerp3a in SERPINB3 and mSerp3a deficient cells.

*Our previous studies in *C. elegans* (Luke et al., 2007) show that intestinal expression of SRP-6 rescues the lysoptosis phenotype in *srp-6* nulls. Moreover, we showed that *srp-6* mutations known to eliminate its lysosomal cysteine peptidase inhibitory activity failed to suppress the death phenotype. These data indicated that the serpin inhibition of cysteine peptidases was required to block lysoptosis. In Fig. S2, we showed that transgenic expression of SERPINB3, -B4 and Serpin3a rescued *srp-6* deficit *C. elegans* from*

hypotonic stress-induced lysoptosis. We therefore transduced HT3^{B3-WT} and HT3^{B3-KO} cells with SERPINB3, SERPINB4, and the inactive P14 mutation, SERPINB3^{A341R} and measured their response to hypotonic stress (Supplemental Fig. 7). rescued the HT3^{B3-KO} phenotype in the cancer cell lines with SERPINB3 but not SERPINB4, nor the inactive P14 mutation, SERPINB3^{A341R} (Supplemental Fig. 7). We included new manuscript text changes to reflect this new data (p11, lines 294-303 and p23, lines 672-684)

4. Please quantify loss of lysosomal content and sytox uptake in Figure 1 (C-L) similar to Figure 1 (Y and Z).

Obtaining an accurate lysosomal count using the FIEC system is difficult technically due to their small cell size and the lower resolution of the spinning disk system used for examination of these cultures. However, we have now provided supplemental movies to show qualitatively the loss of LMP prior to sytox uptake.

5. Please provide statistical p value for following groups and figures:

- **Figure 2Y- statistics between WT and KO in E64d treatment group.**
- **Figure 2NN- statistics between DMSO and Nec-1 treated WT cells.**
- **Figure 3Y- statistics between DMSO and Nec-1 treated KO cells.**
- **Figure 8B- statistics between Nec-1 and E64d treated KO cells.**
- **Figure 8H- statistics between WT and KO in T+S+Q+C treated group.**
- **Figure 9A- statistics between LPS nuc and LPS nuc+ E64 treated WT cells.**
- **Figure 9A- statistics between LPS nuc and LPS nuc+ E64 treated KO cells.**

These statistical values are important because the authors refer to these observations to infer their conclusions in the main text.

We have added these statistics to the appropriate figures.

6. Following blots need to be revised:

- **Figure 5B- uneven loading control b-actin e.g. 16 mins WT and KO lysate lack actin band. Please revise the blot. Done**

Time courses following the initiation of cell death induces cellular protein breakdown as cells begin to die (for examples see Karki et al., Cell, 2020 and McComb et al. Sci Adv, 2019). Evidence of proteome degradation is seen in almost every time course and the red arrows highlight some of these. The degradation of protein over time can be offset in immunoblotting by normalization to total protein loading, but this strategy may obscure the transitions that many cells undergo during the death process. However, we have repeated the blot in Figure 5B and normalized the protein loading. While the actin band is now consistent across the time points 4, 8 and 16 hours, these results were consistent with the data originally submitted and do not change the results nor conclusions from the

original manuscript.

- Figure 8C- uneven loading control b-actin e.g. T+S+Q+C treated WT and KO lysate have reduced b-actin signal. Please revise the blot. In addition, the positive control HT29 cells do not show a robust pRIPK1 band. Please revise since it is not convincing that the experimental conditions are working.

Like 5B, the blot in Fig. 8C, was included to provide molecular evidence of necroptosis by detecting the phosphorylation of RIPK1. However, we agree with this reviewer and reviewer 3 (*vide infra*) that RIP1 alone does not provide all the evidence that necroptosis is induced. Necroptosis also involves the phosphorylation of RIP3K and finally MLKL. However, attempts to detect these by western blotting were unsuccessful. Several reasons may account for this. Biochemical evidence is a static assay (*i.e.* one or a few time points at any given time). We have observed that in the HT3 cell lines, often cells can behave individually in terms of the timing of their induction of cell death and thus may be of relatively low concentration at a particular time point. Additionally, the SW756 and HT3

cells may contain low amounts of RIP3K and MLKL levels below the threshold of detection for immunoblots. Thus, we have now included additional data using immunofluorescence of phospho-MLKL staining to provide additional evidence that at least small amounts of necroptosis can be detected (Fig S16). We feel that this additional data should be convincing enough that necroptosis is being induced in the HT3^{B3-WT} cells, but was undetectable in the HT3^{B3-KO} cells, suggesting that lysoptosis predominates over necroptosis in the absence of SERPINB3.

- **Figure 9B- Please add a loading control below the GSDMD blot similar to GSMDE blot. In addition, please revise the GSDMD blot due to visibly less protein in LPS nucleofection treated KO 5h sample. Alternatively, please provide detail reason in the discussion. In addition, please revise the loading blot below the GSDME blot due to uneven protein samples e.g. Hla treated THP1 cells have no loading control bands in 2h timepoint. Furthermore, the positive control THP1 cells treated with Hla do not show a robust GSDME band. Please revise since it is not convincing that the experimental conditions are working. Previous study used glucocorticoid triamcinolone acetonide in the human T-lymphoblastic leukemia cell line CEM-C7 (Rogers, Nature Communications volume 10, Article number: 1689; 2019)**

The immunoblots in Fig. 9B are from a representative experiment in which all cells were nucleofected at the same time point with different amounts of LPS. Thus, the loading control depicted below the GSDME panel is applicable to both blots as they are from the same cell lysate used to load all the gels. The THP-1 monocytic line is extremely sensitive to Hla and hence, as above, most of the cells have died by the 2h timepoint and thus, have less protein. However, the cleavage band is clearly visible on slightly longer exposure. The THP-cell line treated with Hla simply to served as an immunoblot control for GSDME cleavage.

The GSDME blot was included to demonstrate that were the HT3 cell lines were not undergoing GSDME cleavage under the nucleofection conditions with LPS as had been observed with Hla (Fig. S18 (previously Fig. S12)). The Rogers, Nature Communications volume 10, Article number: 1689; 2019 utilized an additional cell line (not THP1 cells) at 48 h as a positive control for GSDME cleavage. We did however show that HLA induced GSDME cleavage in HT3 cells (Fig. S18) and this data served as a positive control for the GSDME cleavage reagents. We will refer to this control by referring to Fig. S18 in the figure legend of Fig. 9.

- **Supplementary figure 11D- uneven loading control b-actin e.g. considerably low protein samples in the 50 mins KO lysate observed with lack of b-actin band. Please revise the blot.**

The intent of Fig. S11D is to show that varying concentrations (not incubation time) of nigericin does not induce GSDMD cleavage in the HT3 cell lines. We included THP-1 as a positive control for nigericin induced GSDMD cleavage. The THP-1 cells, as well as the 50 μ M nigericin treated HT3^{B3-KO} cells, have high levels of cellular death, which would result in the decreased amount of the actin band for the reasons stated above. However, to eliminate any confusion about the decreased actin band at 50 μ M we will eliminate this data point from the analysis and just report the HT3 cells response from 0-40 μ M.

- **Supplementary figure 12B- Please revise the loading blot below the GSDMD and GSDME blot due to uneven protein samples e.g. Hla treated THP1 cells have no loading control bands in 2h timepoint.**

THP1 cells were include in this analysis to serve as a control for HLA induction of pyroptosis, and to show that our reagents could detect GSDMD cleavage. THP-1 cells are highly sensitized to Hla treatment with 2µg/ml, and total protein content had diminished significantly by 2h after treatment. Although the total protein of the sample was diminished, GSDMD cleavage was still detected, suggesting pyroptosis was induced. Since this was a qualitative study, there was not a quantitative comparison of the response to HLA treatment between THP1 and HT3 cell repeating the study to enhance the detection of the loading control will not change the initial conclusion.

- 7. The interpretation of Supplementary figure 11B,C (text line numbers: 411-412) is not justified. Author need to demonstrate further support by showing the time-course data of Sytox positive cells similar to the loss of TMR graph (Supplementary figure 11C).**

We have addressed the limitation of this approach in comment 1.

- 8. In the text line number 416, the authors conclude that ‘nigericin induced a lysoptosis-like but not pyroptosis.’ This argument requires additional experiments listed below:**
 - **The authors should consider running immunoblots against caspase-1 in HT3 and THP1 cell lines in response to nigericin and Hla treatments in presence and absence of inhibitors including E64d, z-VAD-fmk and YVAD-CHO.**

We want to stress, that only nigericin treatment, NOT Hla or LPS treatment, resulted in a lysoptosis-like instead of a pyroptosis. This finding was based on the absence of GSDMD cleavage (despite robust expression), the presence of LMP, and the ability of E64d (but not zVAD or YVAD) to protect the cells from death. We have detected capase-1 in both the HT3 and SW756 cell lines by immunoblotting (shown below).

However, we now have shown that while caspase-1 is cleaved in the positive THP-1 control cells, there is no detectable caspase-1 cleavage with treatment of nigericin in HT3 cells (Fig. S17)

- The authors should use caspase-1 inhibitor VX765 in nigericin and Hla treated HT3 and THP1 cell lines similar to LPS nucleofection experiment in Figure 9A. This is important because activation of the canonical NLRP3 inflammasome culminates in activation of the effector caspase-1 and protein level of pro-caspase-1 will dictate the downstream processes including gasdermin D cleavage and activation.

We showed that HT3 cell death induced by nigericin was not inhibited by YVAD (a caspase-1 inhibitor like VX765), As expected, YVAD did protect THP-1 cells treated with nigericin (Fig. S11G). Moreover, VX765 failed to protect Hla treated HT3 cells (Fig. S18D) Of note, there was a typographical error on the original Fig. S12D (now Fig. S18D), which should read VX765 instead of VX735. We corrected this in the figure.

What if HT3 cells have defect in pro-caspase-1? In contrast, during the non-canonical NLRP3 inflammasome activation LPS induced activation of caspase 4,5 which can directly cleave gasdermin D prior to activation of NLRP3 and caspase-1 (Swanson et al. Nature Reviews Immunology volume 19, pages477–489; 2019).

We appreciate the reviewer's concern. It is conceivable that procaspase-1 activation might be defective in HT3 cells. For this reason, we also employed the non-canonical inflammasome pathway and show that indeed, GSDMD-mediated pyroptosis could be induced in HT3 cells and LMP was not detected. Thus, under both conditions that we could elicit pyroptosis (one by Hla-induced GSDME cleavage and one by intracellular LPS-induced GSDMD cleavage) lysoptosis was not activated. We were unable to assess the relationship between lysoptosis and classical NLRP3-caspase-1-GSDMD induced pyroptosis. This conclusion is clarified in the discussion (p26, lines 782-785) so as to not overstate the results.

- The authors only show loss of 10kDa TMR -dextran, it is necessary to show data of other low to high molecular weight dextran (similar to figures: 5R-V, 6Q-U, 7M-Q, 8I-M) to further support the current argument.

We have now included this data in the supplemental Fig. S17C-G. Consistent with the 10K-TMR dextran there was no difference in the amount of LMP between HT3^{B3-WT} and HT3^{B3-KO} cells, however more death is observed in the SERPINB3 deficient cells, suggesting that a lysoptosis-like cell death pathway is induced under nigericin conditions.

- Since HT3 cells did not respond to activators (nigericin and Hla) of canonical NLRP3 inflammasome, it is intriguing to investigate whether SW756 cells respond to activators of the canonical NLRP3 inflammasome.

We performed experiments on SW756 cells and showed that cell death activated by nigericin could not be suppressed by zVAD, YVAD, VX765, E64d nor Ca-74-oMe. We concluded that nigericin activated a necrotic cell death phenotype in SW756 cells but did not pursue this pathway further.

9. The GSDMD data observed in LPS transfected cells in Figure 9B is not convincing. In addition, TEM images doesn't showcase morphological characteristics of a pyroptotic cell (Webster et al. J Immunol, 2010, 185 (5) 2968-2979). As mentioned the immunoblot samples were collected 1h after transfection (text line 441) however, based on previous studies activation of the non-canonical NLRP3 inflammasome by cytosolic LPS requires 8-16h (Viganò et al. Nature Communications volume 6, Article number: 8761, 2015;

Knodler et al. Cell Host Microbe. 2014 Aug 13; 16(2): 249–256). The author should repeat the experiment where the cells are incubated with cytosolic LPS for 16h consistent with previous literature and perform GSDMD immunoblots.

The Webster et al. J Immunol, 2010, 185 (5) 2968-2979 paper referenced shows TEM images of monocytes infected with different bacterial strains, which have a completely different morphology than the HT3 cells. The TEM images of cells incubated with the Lonza Se buffer was enough to produce some cytoplasmic vacuolization (Fig. S19C and D). However, the cellular morphology including cytoplasmic vacuolization and organelle disintegration was enhanced markedly in the presence of LPS (Fig. S19 G and H). Thus, we respectfully disagree with the reviewer on this point since, to our knowledge, there are no published TEM images of any cell line nucleofected with LPS. Moreover, we are unaware of any “standard” morphological description of pyroptosis.

The data reported by Viganò et al. 2015 Nature Comms 6: 8761 showed robust IL1B release at 5h after direct LPS treatment with 1ug of LPS. Most experiments were performed at 5h on monocytes, not epithelial cells, which did not undergo cell death by an LDH assay. Knodler et al. Cell Host Microbe. 2014 Aug 13; 16(2): 249–256 nucleofected S. typhimurium LPS into Caco-2 C2Bbe1 colorectal adenocarcinoma cells whereas we used LPS derived from E. coli K12 strain into HT3 cervical cells. As the experimental conditions are different and there was no quantification of LPS transfection efficiency, strict quantitative comparisons may not be feasible. For example, the efficiency of nucleofection is greater than that of standard electroporation, is unlikely to affect 100% of cells. Thus, the amount of GSDMD cleavage will never be complete in this system. Moreover, different cell lines will have different killing kinetics. By flow cytometry, we detected ~80% killing at 270 min (4.5h). The cellular constituents would be degraded by secondary necrosis if we examined them at 8-16h. Also, different cell types respond differently to cell death signals (for example several cell types do not undergo necroptosis or pyroptosis due to absent cellular machinery), and the source and type of LPS can dramatically affect the cellular response. Since the HT3 epithelial cell line is adherent in nature, and nucleofection requires trypsinization of the cells, it is unlikely that these cells will survive for 16h even in the absence of LPS. Thus, we conclude that the extended time course will not yield any significant information that will alter the original observation that non-canonical pyroptosis was activated in HT3 cells.

10. Please revise the conclusion in text lines (588-589) indicating that death could only be prevented by E64d whereas, there is a partial protection in presence of Nec-1 inhibitor in response to T+S+Q+C treatment (Figure 8B) suggesting that both necroptosis and lysoptosis are active cell death pathways in response to T+S+Q+C treatment in SERPIN3B null cells.

Good point, thanks, manuscript revised.

Minor comments:

1. Methods and material:

- **Experimental conditions of LPS nucleofection is missing the following details:**
- **Figure 9A: the amount of LPS used for transfection and the duration of activation is missing. In addition, for ease of readers, please include the time of incubation with LPS for Figure 9B in the legend; currently it is mentioned in line 441. done**
- **Figure 9C: the amount of LPS used for LPS transfection. done**

- Supplementary figure 13G,H is missing the duration of LPS activation.
- Supplement material is missing experimental conditions used for activating cell lines with nigericin.

These changes have been made to the revised manuscript.

2. Multiple figure legends are missing experimental conditions and details which are highlight below:

- Figure 9 legend: missing experimental detail regarding the loading control blot below GSDME.

These changes have been made to the revised manuscript.

- Supplementary figure 11A-C: can the authors please confirm if HT3 cells were pre-treated with LPS before activating with nigericin. If yes, please add the experimental conditions used for nigericin activation.

We tried both activation and non-activation of these cells and so no quantitative differences, thus the cells were not activated with LPS before activating with nigericin in the data shown.

3. Please define (+) and (-) controls in Supplementary figure 1D.

These changes have been made to the revised manuscript.

Reviewer #2

Major comments:

1. The whole concept of lysoptosis as independent cell death machinery strongly relies on the interactions between lysosomal enzymes (mostly cathepsins) and serpins (SERPINB3 in humans). However, do authors have any suggestions which lysosomal enzymes play a dominant role in this cell death? Is it cathepsin B, L or S? And do the cathepsins have to be active in the cytoplasm to trigger cell demise, or they rather induce lysoptosis from the lysosomes? The concept of lysoptosis is very interesting, and can make a significant impact on the field, however, this work lacks a detailed mechanistic insight into interactions between SERPINB3 and lysosomal enzymes at the level of single proteins.

This issue is being addressed by the experimental revisions described in response to Reviewer 1, comment #2 (above). In addition, we have conducted a CRISPR screen of LLOMe-induced lysoptosis. These unpublished data show that loss of CatL suppresses the cell death pathway. RNAseq data from human cervical squamous cell carcinomas show an increase in lysosomal peptidases: CatD>CatB>CatH>CatL.

2. According to the manuscript, the lack of SERPINB3 sensitizes cells for lysoptotic stimulants. Is it also true for other lysosomal proteases endogenous inhibitors, or this cell death mechanism is exclusively associate with SERPINB3? Therefore authors may consider testing this with HT3 cell lines lacking other lysosomal proteases inhibitors.

We show that only SERPINB3, -B4 and Serpinb3a protected C. elegans from lysoptosis. Murine Serpinb3a inhibits both serine and cysteine proteases and is orthologous to both

SERPINB3 and -B4. Since HT3 and SW756 cells do not contain SERPINB4 (Fig. S5) we focused on SERPINB3 in these cells. The only other serpin that has robust published lysosomal cysteine protease inhibitory activity is SERPINB13, which did not rescue lysoptosis in C. elegans. Another family of cysteine protease inhibitors are the cystatins. These proteins are competitive reversible inhibitors but are unable to permanently eliminate cytosolic lysosomal cysteine proteases like covalently binding serpins. Intracellular cystatins (e.g., stefin/cystatin A and stefin/cystatin B) do not regulate lysoptosis in C. elegans, as these genes, unlike the serpins, are not conserved in the nematode lineage. We analyzed the expression levels of other functionally relevant protease inhibitors, SERPINB1, -B6 and -B13, as well as cystatin A and B, in HT3 cells by qRT-PCR and western blotting and showed that only SERPINB1 was expressed in all cell lines but this serpin failed to block lysoptosis in C. elegans, nor were levels altered in B3-KO cell lines compared to B3-WT. Moreover, we showed that we could get complete protection from lysoptosis with SERPINB3 but not the serpin inactivating mutation, A341R. Thus this data would suggest that SERPINB3 alone is responsible for protection against lysoptosis in these cell lines.

3. The main stimulant to trigger lysoptosis was “hypotonic stress”. Do authors are aware of any other putative stimulants, that can trigger this cell death in HT3B3-KO system?

In the manuscript, we show that hypotonic stress, STS, H₂O₂, T+S+Q+C and Nigericin induce LMP in B3-KO cells but not B3-WT cells. In addition, we induced direct LMP using lysosome destabilizing agents, such as LLOMe, and show that HT3^{B3-KO} and SW756^{B3-KO} cells have increased cell death in response to LMP as compared to WT controls. These data are part of manuscript in preparation that describes a high-content drug screen for compounds that block lysoptosis. In a companion study ready to be submitted, we also show that both SW756 and HT3 SERPINB3 KO cells are more sensitized to irradiation and die by lysoptosis as compared to WT cells. Since cervical cancer patients expressing high levels of SERPINB3 have a poor prognosis and their tumors are resistant to radiotherapy, this highlights the biological importance of both the serpin and the lysoptosis pathway.

4. The justification for the selection of HT3 cell lines is clearly presented. HT3 is a cervical carcinoma cell line, and the over-expression of SERPINB3 correlates with poor prognosis of patients with squamous cell cancer, as such cells are more resistant to classical chemotherapy. However, it would be beneficial to the manuscript if authors could demonstrate that the lysoptosis occurs also in other cancer cell lines.

We showed in the manuscript that two SERPINB3 expressing cervical cell lines HT3 and SW756, which are different in that one is HPV- and one is HPV+. Unfortunately, non-cervical tumor cell lines that express detectable levels of SERPINB3 are scant. A head and neck squamous carcinoma cell line, SCC090, expresses relatively high levels of SERPINB3. We attempted to generate SERPINB3 knockout or knockdown in this cell however, these attempts have been unsuccessful thus far and may attest to the role of SERPINB3 as a critical prosurvival factor in certain tumors.

Reviewer #3

Major comments:

1. Hypotonic stress induces cell death in the mSerpin**3a **-/-** cells. To further validate that observed cell death represents lysosomal cell death authors should include the analysis**

of the classical hallmarks of lysosomal cell death like Cathepsin translocation to the cytosol, Cathepsin activity in the cytosol.

This issue is addressed in response to Reviewer 1, comment #2 (see above).

In addition the apoptotic features observed in Serpinb3a wt cells are confusing and probably not helping to tell the story- at 60 min there are only 20% dead compared to 80% in the knock out.

We understand the reviewer's confusion. However, it is well documented that no cell death pathway operates in isolation. Rather, certain stressors will induce several cell death routines with one pathway dominating or emerging if another pathway is blocked (this is how necroptosis was discovered). In WT cells exposed to hypotonic stress, very few cells die: some demonstrate morphological features of apoptosis (but some cytosolic vacuolization consistent with necrosis); others with features of necrosis. In contrast, Serpinb3a^{-/-} cells are much more susceptible to hypotonic stress with lysoptosis becoming the predominant pathway in the absence of the inhibitor. This result is consistent with the major theme of the data showing that SERPINB3-KO cell lines are sensitized to lysoptosis and underscores the that the high concentration of the intracellular serpin prevents this pathway from being fully activated in WT cells. Moreover, the co-activation of cell death pathways explains why E64d markedly, but not completely, blocks cell death even in Serpinb3a^{-/-} FIECs. We believe that removing this portion of the figure, while perhaps leading to less confusion, obscures the true complexity of interacting cell death pathways.

2. Results are not always in a clear order. There is a jump between Figure 2 and 3 and back to Figure 2.

We appreciate the reviewer's concern and we have modified the text such that there is no longer a need to refer back to Fig. 2 after Fig. 3 is discussed.

Also clarity is missing at the level of the models used: results start with mouse then cell lines and mouse in the end.

The reviewer is correct, we started with an in vitro model using mouse FIECs, and then moved to another in vitro system using human squamous cell carcinoma cell lines for evidence of conservation and ease of manipulation. We concluded the manuscript by showing, in vivo, that serpin loss accentuated whole animal death in a mouse model of necrotizing enterocolitis. We believe this progression of in vitro work to in vivo work makes for a more compelling story for the physiological role of lysoptosis. We defer to the editor's preference.

Further authors state that Serpin3 is expressed in intestine, lung and skin- what is the rational of taking a cervical cancer cell line as a model system? Authors should explain.

The cervical cancer cell lines are squamous cell carcinomas (SCCs) derived from skin of the cervix (stratified squamous epithelia) and like many SCCs are high expressors of SERPINB3. Interestingly, reviewer 2 stated "The justification for the selection of HT3 cell lines is clearly presented." Nonetheless, this explanation is now included in the Results section where the cell lines were introduced.

3. The canonical pathway of lysosomal cell death includes LMP- followed by the release of Cathepsins into the cytosol- cleavage of tBID by cytosolic Cathepsins- activation of Caspases, which then at least contribute to cell death execution. To exclude the involvement of Caspases authors should increase the concentration of DEVD-CHO as 1uM seems very low, a better inhibitor of Caspases is zVAD-fmk, which is also later also used in the paper. Results should be backed up by western blots demonstrating the absence of Caspase cleavage in order to establish the concept Caspase-independent lysosomal cell death in the absence of Serpin3.

LDCD is an ill-defined necrotic cell death pathway and many different mechanisms/pathways implicated. The cell death nomenclature committee operationally defines “lysosome-dependent cell death as a form of regulated cell death demarcated by primary LMP and precipitated by cathepsins, with optional involvement of MOMP and executioner caspases.” Thus, the triggering of apoptosis by cathepsin cleavage of tBID is one of many cell death routines involving LMP and in itself does not define a canonical lysosomal cell death pathway. For example, others have shown that cathepsins are also directly responsible for cleavage of multiple proteins (e.g. RIP1K, McComb et al., J. Immunol. 2014) to induce other cell death mechanisms. Indeed, LMP and cytosolic cathepsin activity has been associated with most cell death forms (reviewed in Galluzi, et al., Cell Death Diff 2018). However, whether this activity is associated with the execution of these pathways or is a consequence of cellular demise is in question. DEVD-CHO is marketed as a potent reversible caspase 3/7 inhibitor; however analysis has shown that it can inhibit other caspases (Garcia-Calvo et al. JBC, 1998). 1µM DEVD-CHO completely suppressed caspase activity (Fig 5A). zVAD.fmk is an irreversible pan-caspase inhibitor which is often used to block the apoptosis pathway and switch cells to a necroptosis cell death mechanism. However, the fmk moiety on caspase inhibitors has been shown to inhibit cathepsins (Schotte et al., 1999) and thus, may not be the best choice in this case. We showed that even though caspases are activated by STS treatment using fluorogenic substrate and western blot, lysoptosis occurred in the presence of this pathway.

Serpins are described to inhibit Caspase-independent lysosomal cell death involving ROS (Liu et al 2004). Possible involvement of ROS in cell death induction should at least be explained.

There is little doubt that ROS induce or exacerbate lysoptosis. C. elegans treated with H₂O₂ and unpublished work with cell lines treated with t-BOOH (tert-Butyl hydroperoxide) show marked lysoptosis. Moreover, induction of mPT-dependent necrosis by H₂O₂ in WT cell lines triggers lysoptosis in SERPINB3 nulls (Fig. 6.). These data confirm that some types of free radical injury triggers lysoptosis. In contrast, ROS generated during ferroptosis, which results in lipid peroxidation, does not to trigger lysoptosis. Thus, ROS production is not an obligate trigger for lysoptosis. However, we now include data showing that the ROS indicator, H₂-DCFDA, does not significantly increase in cells treated with hypotonic stress, while death is still exacerbated in HT3^{B3-KO} cells (Supplemental Fig. 10, text pp11-12, lines 307-316). These data are consistent with the ferroptosis data that ROS is not induced during this cell death pathway.

Further use of RIPK1 inhibitor NEC-1 is not easy to understand as necroptosis can also occur in the absence of RIPK1. The exclusion of necroptosis requires a more careful approach.

We showed that RIP1K was phosphorylated in T+S+Q+C treated cells and that death associated with this treatment in HT3^{B3-WT} was inhibitable with Nec-1. As such, we concluded that RIP-1 dependent necroptosis was induced in the HT3^{B3-WT} cells treated with T+S+Q+C. However, we did not show RIP3K or MLKL data. We now include immunofluorescence analysis of cells treated with our necroptosis induction cocktail and show that HT3^{B3-WT} has a small but significant increase in phospho-MLKL staining but not in the HT3^{B3-KO} cell lines (Supplemental Fig. 16, text p17, lines 481-484). Moreover, the phospho-MLKL staining was repressed with the necroptosis inhibitor, necrostatin-1 (Supplemental Fig. 16, text p17, lines 481-484). These data, combined with the original data, support the concept that the necroptosis pathway is induced in HT3 cells but lysoptosis-like cell death predominates in the HT3^{B3-KO} cell line.

4. In relation to point 3: Authors claim that inhibition of Cystein proteases inhibit LMP. Consequently Cathepsins would act upstream of LMP, in lysoptosis and Serpin3 would execute their inhibitory role in the lysosomes? This finding needs further investigation as it is in contrast to the current knowledge of lysosomal cell death.

Based on the ability of E64 to block lysoptosis and LMP, the reviewer may have inferred that cathepsins would act upstream of LMP. However, based on the C. elegans model using genetic epistasis experiments and live-animal imaging, we previously showed that an influx of cytosolic calcium and calpain activation triggers lysoptosis and subsequent lysosomal cysteine peptidase release (Luke et al., Cell 2007). In turn, the cytosolic release of cathepsins secondarily exacerbates LMP leading to death (multiple examples of calpain-cathepsin hypothesis the literature). Since E64 (and SERPINB3) inhibits both calpains and lysosomal cysteine peptidases, there is no need to speculate on cathepsins triggering the loss of lysosomal membrane integrity or for a role for SERPINB3 functioning inside the lysosome.

5. In order to support the hypothesis of lysosomal cell death under hypotonic stress situations in Serpin3 k.o. cells, other cell death pathway should be excluded in the hypotonic stress model.

We agree completely. This is the reason we tested multiple cell death programs concurrently. It is extremely difficult to prove a negative, i.e., that apoptosis, for example, is not occurring in the cell lines exposed to hypotonic stress in the Serpinb3 nulls, unless you can prove you can detect apoptosis in the same cell lines under apoptosis inducing conditions. This was the case, and none of the inhibitors of apoptosis necroptosis, MPT-necrosis, ferroptosis, or pyroptosis could block hypotonic stress-induced lysoptosis. We had showed that lysoptosis in mSerpib3a^{-/-} FIEC cultures induced by hypotonic stress could not be inhibited by DEVD.cho nor necrostatin. Moreover, we have data showing that the caspase inhibitor DEVD-CHO does not inhibit HT3^{B3-KO} cell death when exposed to hypotonic stress (see below):

However, as mentioned above, many cell death inhibitors can also inhibit cysteine proteases. Additionally, we have evidence that preincubating wild-type cells with other cell death pathway inhibitors can sensitize even B3-KO cells to lysoptosis, by mechanisms we did not investigate. We have developed a high-content screening to determine inhibitors of lysoptosis that may be able to address these concerns, however these data are outside the scope of this current study. Thus we will defer to the editors discretion on this point.

The switch to an apoptotic system by treatment of cells with apoptosis- inducer STS does not support lysoptosis as an independent regulated form of cell death. Further authors claim that STS (apoptosis inducer) / STS + zVAD also triggers lysoptosis in Serpin3 k.o. cells. This needs to be validated by more experiments (Dunai et al 2012).

The STS-apoptosis inducing experiments were conducted in the cell lines to show that the cells were capable of undergoing apoptosis and that this form of cell death could be blocked by caspase inhibitors. The fact that lysoptosis dominated the cell death phenotype in the Serpinb3a nulls, even in the presence of executioner caspase activation, only supports the notion that lysoptosis can be activated concurrently by apoptosis inducing agents but is only manifest because there is no SERPINB3 to neutralize the consequences of the variable LMP that appears to occur in response to a multitude of cell stressors. In other words, in the absence of SERPINB3, lysoptosis is exacerbated in response to STS and death can no longer be blocked by caspase inhibitors, only by E64d. Although the reviewer suggests that this observation does not support lysoptosis as an independent pathway, the argument is moot. No cell death pathway is truly independent. They are co-activated to variable degrees (e.g., apoptosis and necroptosis) with one dominating based on multiple conditions outside and inside the cell. Apoptosis and necroptosis are intimately related and share entire sets of signaling complexes (e.g., TNF-receptor complex I) and are co-regulated by multiple factors. Yet, cell death scientists recognize these interdependent cell death routines as independently regulated forms of cell death because there are still components specific to each pathway (e.g., caspase 3 vs MLKL). A similar relationship exists between apoptosis and lysoptosis (e.g., caspase 3/7 in apoptosis and Serpinb3a/SERPINB3 and cathepsins in lysoptosis).

Further Dunai et al demonstrated that STS-induced necroptosis can be blocked by Cathepsin inhibitor Ca-074 – leaving the lysosomes in this cell death modality in a central

position. How can this be combined with the proposed lysoptosis cell death model? Same is true for the HT3 model.

The Dunai paper shows that death induced by a necroptosis trigger (STS + zVAD) can be blocked by either Necrostatin or the cathepsin B inhibitor, Ca-074-OMe (the methyl ester of a synthetic analogue of E-64). The reviewer is actually making our point. Necroptosis triggered in U937 is partially blocked but never completely blocked by Necrostatin. Similarly, Ca-074-Ome partially, but not completely blocks death induced by STS-zVAD. Indeed, Ca-074-Ome has been shown to block most forms of necrotic cell death including pyroptosis, necroptosis and LDCD (Brojatsch et al., 2015). The key experiment in the Dunai reference was assessing lysosome content (through pH) by acridine orange (AO) fluorescence. AO staining is markedly diminished after STS-zVAD treatment and only partially blocked by Necrostatin, but almost completely blocked by Ca-074 (inhibiting lysosomal cysteine peptidases). These data clearly show that, like our study, STS-zVAD treatment triggers necroptosis and also an element of lysoptosis. Thus, those U937 cells not dying by necroptosis go on to die by lysoptosis. These findings support our contention that lysoptosis can be triggered concomitantly with some, but not all, cell death inducers. However, in our studies, lysoptosis predominates in the cells that loss SERPINB3.

9. Do the authors have an explanation why there is 70 % Caspase-activity in wt HT3 cells when there is only 20 % Annexin positive cells detected (Figure 5)? There is a discrepancy in death induction between Figure 5Q and 5I-K. FACS analysis of Annexin/PI stained cells would give some clarity here. Further discrimination of secondary apoptosis and necrotic cells can be provided by the inclusion of zVAD into the Annexin/PI assay. As well as the inclusion of multiple different time points.

As we have shown throughout the manuscript, the timing to cell death in multiple cell death activations can vary from experiment to experiment. Thus, a direct correlation between these experiments is not possible. The data in Fig. 5 represent the results from several different but reproducible experiments. Specifically, the caspase (Fig. 5A) and PI data (Fig. 5Q) were collected from the same set of cultures. In contrast, the annexin V data (Fig. 5O) represents the quantitative analyses of the images in Fig. 5C-N and was not optimized to coordinate with caspase activity. The purpose of the two datasets (Fig. 5A, Q and Fig. 5C-N, O, P) was to show that wild-type cells were capable of undergoing apoptosis. An early time course was selected to avoid the confounding effects of secondary necrosis (annexin V staining and lysosomal leakage), which would be impossible to differentiate from lysoptosis. The data show molecular features of apoptosis: caspase 3/7 activity, annexin V staining and minimal LMP early in the time course of STS treatment of wild-type cell. These results contrast markedly with the dominant lysoptosis phenotype in the SERPINB3-KO cells. We will amend the Figure legend indicating that Fig. 5A and Q are from the same experiment and that Fig. 5O and P represent the quantitation of the images depicted in Fig. 5C-N. The relationship between caspase 3 activity and annexin V positivity is unclear. While there have been some reports that caspase 3 activity is required for phosphatidyl serine externalization have been reported (e.g. Mandal et al., Febs letters, 2002), it has also shown that annexin V positive cells have been shown in both pyroptosis and necroptosis pathways due to the pore formation in the plasma membrane (Miao et al., 2011., Zagarian et al. PLoS Biology 2017). As such, secondary apoptotic cells, necroptotic cells, and pyroptotic cell, as well cells undergoing the latter stages of necrosis pathways, would all be Annexin V and PI positive. Thus, the proposed FACS experiment by the reviewer would not aid this discrimination.

Caspase- Blots need redoing- there is nothing loaded (at 16h cells are not secondary apoptotic yet – the absence of bands cannot be explained by loss of protein content due to advanced cell death).

This issue is addressed in response to Reviewer 1, comment #6 (see above).

Also lysosomal cell death should be much better characterised. The release of the different dextran sizes is more confusing than helpful- release of LysoTracker or acridine orange which again can be analysed by FACS answers the question of LMP sufficiently.

Lysotrackers and acridine orange (AO) are lysosomotropic fluorescent compounds that are used to assess LMP. Each has advantages and limitations. LysoTrackers and AO can both diffuse across lipid membranes and become trapped and concentrated in acidic environments. Thus, the loss of LysoTracker staining over time can indicate increased LMP with the presumption that the compound is leaked and diluted by the cytosol. However, LysoTracker fluorescence requires a low pH. Thus, a loss in signal could also indicate failure of the v-APTase to maintain a proton gradient across the lysosomal membrane. Therefore, the loss of staining per se is not an absolute indicator of LMP. AO is a metachromatic dye that aggregates and fluoresces red after blue-light (488nm) excitation in acidic organelles. If AO leaks into the cytosol, it becomes monomeric and fluoresces green. However, the loading of lysosomes with AO can interfere with their function and blue light excitation can lead to the generation of ROS that trigger LMP. Additionally, since LysoTracker and acridine orange are lysosomal pH sensitive, small amounts of LMP will cause changes in fluorescence in both these fluorophores. We showed that during apoptosis in HT3^{B3-WT} cells, the lower MW dextran was released but the higher MW dextrans were retained, which suggested a low level of LMP. Thus, experiments proposed by reviewer 3 above would show a decrease in fluorescence intensity during both apoptosis and lysoptosis, whereas the dextran labeling experiments are able to distinguish between these pathways. For these reasons, we used several different markers to assess LMP and avoided AO staining except for the FACS analysis after nucleofection of LPS in the pyroptosis assay, as this process prohibited live cell imaging. Since cells are labeled by a cocktail of lysosomal probes, we can measure simultaneously their individual decays over time. This technique gives us information on how fast LMP occurs after different treatments and also the extent of LMP by showing the progressive release of smaller to larger molecular mass dextrans. This technique clearly shows that the timing and degree of LMP varies depending on the cell death stimulus. The differential release of various MW dextrans is an established methodology for determining the severity of LMP provides a more elegant and quantifiable determination of lysosomal integrity. We are not certain why the reviewer finds this approach confusing as no rationale is provided. If we assume the reviewer is concerned about the LMP detected in the apoptosis assays in Fig. 5, the data clearly show that a small degree of LMP (leakage of the 3 kDa dextran species) is present in WT cells undergoing apoptosis. However, the SERPINB3 null cells, treated identically, show a marked increase in LMP characterized rapid loss of all the dextran markers (3-70 kDa). This is the exact same result we observed with LysoTracker, and consistent with all of the data presented in Fig. 5 showing that even under conditions to activate apoptosis, lysoptosis predominates in the absence of SERPINB3. We are hard pressed to understand how using AO and flow sorting would change this conclusion or add more clarity. Thus, we respectfully disagree with this reviewer on this matter.

How do the authors explain that the Cathepsin inhibitor E64d has no effect on Caspase-activity in HT3B3-k.o. cells? The order of events in LCD is believed to be Cathepsins released in cytosol- cleavage of tBid- activation of Caspases.

Apoptosis in Fig. 5 was induced by STS. Although, STS may induce apoptosis by more than one mechanism, activation of the intrinsic pathway has been implicated. Activation of the intrinsic pathway involves the mitochondrial release of cytochrome C and formation of the caspase 9-activating apoptosome. The intrinsic apoptosis pathway does not involve LMP and cathepsin release (at least initially). For this reason, we did not expect much inhibition in the WT or KO line with E64, as this inhibitor blocks calpains and lysosomal cysteine proteases, but not caspases. The reviewer also correctly states that a form of LCD results in apoptosis via LMP, cathepsin release, tBid cleavage and activation of the intrinsic apoptosis pathway. However, in our studies presented here, lysoptosis (inhibited by E64, but not DEVD) and not LCD as described above (inhibited partially by E64 and completely blocked by DEVD) is the predominant pathway. Since there was no evidence LCD (as the reviewer defines it) the reviewer confirms that a different form of lysosomal cell death is occurring in the SERPINB3-KO cells (i.e., lysoptosis).

10. What is the rationale between Figure 5 and 6- the paper seems brakes here. Authors jump from trying to establish Serpin 3 as key regulator in lysoposis which is a form of regulated Caspase independent cell death- to a sensitizing role of Serpin3 in the execution of alternative regulated cell death pathways. In addition ferroptosis and pyroptosis are not blockable by the Cathepsin inhibitor E64d. This might be better placed in supplemental information.

*The major premise of the manuscript was that lysoptosis, as defined in *C. elegans*, was evolutionarily conserved, and could account, in part, to the LMP associated with many different regulated forms of cell death. Since *srp-6*, *Serpinb3a* or *SERPINB3* loss sensitized, cells to lysoptosis, we had an experimental paradigm that allowed us to test this hypothesis. This paradigm was tested in STS-induced apoptosis and showed WT type cells treated with STS died by classical apoptotic death despite a relatively limited amount of LMP. In contrast and surprisingly, *SERPINB3* KO cells, demonstrated extensive LMP and death by lysoptosis despite caspase 3/7 activation. These data strongly suggest that STS triggered both apoptosis and lysoptosis, but death by lysoptosis was held in check by the intracellular serpin, reminiscent of the necroptosis pathway being activated by apoptotic stimuli in the presence of caspase inhibitors. Moreover, activation of multiple death routines by a single stimulus is a recurring theme detected throughout the death literature. For these reasons, we examined for the presence of lysoptosis in several well described cell death routine in which some degree of LMP has been implicated in the demise of the cell.*

11. In vivo experiments include NEC which is based on TLR4-induced necroptosis. The role of lysoptosis in receptor-mediated necroptosis is not addressed in the paper and evidence that this mouse model is switching to lysoptosis in the absence of serpin3 is not sufficient.

The first author of the manuscript as well of one of the senior authors are board certified neonatologists and can attest that the pathogenesis of neonatal necrotizing enterocolitis is still multifactorial and unknown. The process appears to begin with mucosal injury, death

of enterocytes (epithelial cells), bacterial invasion of the lamina propria and deeper structures and a vigorous inflammatory response. Eventually full thickness and irreversible necrosis of the bowel and a systemic infection leads to high mortality rates. The mouse models of neonatal NEC are imperfect (the disease incidence is highest in the most premature infants and the mouse models use term pups) and utilize different combinations of risk factors (e.g., formula tonicity, dysbiotic bacterial, feeding frequency, hypoxic atmosphere) to mimic the disease. The reviewer cites a study described by Wertz et al. 2019 showing that there was an up regulation of necroptotic genes and RIP3K phosphorylation using one method for inducing NEC. Other NEC induction strategies implicate apoptosis (Good et al 2015 Mucosal Immunology) and pyroptosis (Chen et al., 2020) as the cause of intestinal cell death. Nonetheless, even if we assume that NEC induction in premature infants is caused by TLR4 triggering of necroptosis, this conclusion would not preclude the participation of lysoptosis playing a role. The data in Fig 8. show that under conditions that induce necroptosis, LMP becomes more extensive and lysoptosis dominates the landscape in the absence of SERPINB3 (death is marginally blocked by Nec-1, with E64d providing most of the protection). In the studies Wertz, Nec-1 is only partially protective indicating the enterocytes are dying by additional cell death mechanisms, possibly lysoptosis, as intestinal cell levels of Serpinb3a are relatively low in newborn mice. If this were the case, intestinal injury should be more pervasive in the Serpinb3a null mice, which is exactly what occurs. **The major objective of performing the experiment was to show that Serpinb3 loss would exacerbate the degree of injury in vivo and that is exactly what occurred.** We did not make any further claims as it to lysoptosis being the direct cause of NEC per se. Further definition of lysoptosis in this model is beyond the scope of this manuscript and being pursued in our laboratory.

Minor points

1. Please change the title. The main focus of this paper is not that lysoptosis is an evolutionary mechanism- this is a conclusion from the previous paper published by the authors in 2007 and the current one.

We feel that the title clearly reflects the premise of the work and will defer to the editors discretion on this matter.

2. Labelling of the fluorescence images is not always easy to follow, for example labelling in figure 4 T-W is difficult to read.

We have adjusted some of the labelling to make it easier for the readers to see.

3. First result section line 121-151 just contains supplemental information Those result should be part of a figure as they establish the system and validate the futher investigation of mouse/ human Serpin.

While we agree with the reviewer on this, the figure limit of the journal means that we cannot put all the new data in the main figures. Thus, we will defer to the editors discretion on this matter.

4. Line 132: Figure 1A is not identical with Figure 1A.

We have corrected this in the manuscript

5. Figure 1C quantification is missing.

We have addressed this above (reviewer 1, comment 4).

6. Figure 3G-L should show representatives of Figure 3S.

The quantification of the % annexin V is from multiple fields of the representative images shown in G-L.

7. Fluorescence images do not always represent the according graphs.

We have addressed this above

8. Figure 2MM and NN miss legends.

The chart legends are the same as in Figure Y and Z. in order to not clutter the graphs we have utilized the same legends for Y and MM and Z and NN.

9. Line 276 N2 wild type c.elegans are not in Figure 1T.

We have corrected this in the revised manuscript.

Reviewers' comments:

Reviewer #1 (Remarks to the Author):

The authors have addressed many of my concerns. However, the following minor comments need to be addressed:

1. Line 292-293. The inference for Fig S10B doesn't make sense. The % dead cells of HT B3WT and B3KO cells is comparable to each other and less than 20%. Please revise.
2. Presentation of the following figures should be improved for ease of reading:
 - S13D - Please place WT and KO cells side by side for ease of comparison between 2 different treatments (0 vs 20 digitonin and 100% vs 10% DPBS) [similar to Fig 4B].
 - S13E, F- Please place WT and KO side by side for ease of comparison between different treatment 100% vs 10% DPBS (similar to Fig 4A).
 - S13B,C- please add an arrow similar to S12B-U showing the exact band size of the protein of interest.
 - S12B-U- Please define the meaning of black vs white arrows in the figure legend.
3. Line 202-203: the following statement 'Neither DMSO (diluent), E64d, nor DEVD-CHO had an effect on cell viability' was made without reference to any figure or data. In addition, there was no data demonstrating this inference in either the main or supplementary figures. Please revise.
4. Line 402- based on Fig 6P the cell death in the HT3 B3KO cells was partially rescued by the E64D; only 50% rescue of cell death in the presence of E64D in response to H₂O₂. Please revise the text.
5. Fig 8C: In the rebuttal authors provided the argument that 'cell death induces cellular protein breakdown', I agree. However, can the authors please explain why the beta-actin band in untreated (-) HT3 B3KO cells weaker than the HT B3WT control. The untreated cells are not expected to undergo cell death. Does it mean that at the start of experiment, the HT B3KO cells were unhealthy and undergoing cell death? Alternatively, was there uneven loading of protein lysate between the HT3 B3WT and HT3 B3KO cells and hence samples were run on two separate blots. Therefore, please provide the Sytox+ data for untreated HT3 B3WT and HT3 B3KO cells for Fig 8A. Please run the HT3 B3KO samples alongside the HT29 and HT3 B3WT samples on the same blot for all the proteins of interest.
6. Fig S17K- There seems to be a discrepancy in the % dead cells in HT3 B3WT vs HT3 B3KO in response to nigericin treatment compared to S17- J and A. Also, the % dead cells is just 40% compared to S17- J and A (~80%). In addition, data in S17- J and A suggest that B3KO cells underwent high % of dead cells compared to B3WT cells in response to nigericin. Thus, the experiment for S17K needs to be repeated.
7. Line 754- Please change P25 to P20 (casp1 cleavage).
8. Line 1382- mention the % of DPBS used in Fig 6U-O.

Reviewer #2 (Remarks to the Author):

All the reviewer comments have been sufficiently addressed. The authors have performed a set of new experiments and now the data strongly support the research hypothesis. The reviewer has no further comments.

Reviewer #3 (Remarks to the Author):

In this revised manuscript by Good et al, the authors have addressed some of my original concerns and additional experiments have been performed. However, the hypothesis of lysoptosis being an evolutionary conserved, independent regulated cell death mechanism still is not well supported.

Although different cell death routines might be induced by a single death trigger, lysoptosis /LMP has to initiate all downstream death events. If cell death is not inhibited by interference with lysoptosis – like it is currently the case – lysoptosis is only a bystander of death, but not the inducer. Authors claim Cathepsin L as the main executor of lysoptosis. This finding could give the authors the possibility to strengthen their findings (also from a mechanistically aspect) and validate their hypothesis. In addition, the *in vivo* model used, despite expert knowledge of the authors in NEC, does not support the hypothesis of lysoptosis being an independent RCD.

Major comments

1. Authors argue that E-64 cannot fully inhibit lysoptosis induced by STS, H₂O₂, TSQC and LPS (in this case the inhibitory effect is in particular minimal), as multiple cell death routines are induced in parallel. However, to claim that lysoptosis is an independent RCD, it is essential and crucial to demonstrate that lysoptosis is the initiator of death and not a bystander. Consequently inhibition of lysoptosis has to block all downstream death events (necroptosis, apoptosis, pyroptosis,...).

2. Authors claim Cathepsin L as the main executor of lysoptosis. However, active Cathepsin L is released from the lysosomes in both Serpin B wt and k.o. cells. Consequently, the conclusion that Cathepsin L executes lysoptosis cannot be drawn. To confirm the executor role of cathepsin L during lysoptosis experiments need to be repeated in the presence of a specific cathepsin L inhibitor. In addition, also the knock out cells for Cathepsin L should also be exposed to the other death triggers used in the paper.

3. Authors speculate that the death is based on the calpein-cathepsin hypothesis. If this is the mechanistical basis of lysoptosis triggered by all different death stimuli used (hypotonic stress, STS, H₂O₂, TSQC, LPS, Nigericin) then the author need to demonstrate initial calcium influx and calpein activity. The previous publication by Cliff et al, to which the author refer demonstrates calcium influx in nematode, but appears distinct to the death triggers used in this manuscript. In particular STS (Fig.5), caspase activity is not blocked by E-64. The author state correctly E-64 is also a calpein inhibitor. Again, a specific Cathepsin L inhibitor is the better choice.

4. Necroptotic cocktail induces lysoptosis. To clearly demonstrate that LMP is the initial event leading to Cathepsin L release to execute the necroptosis like phenotype, necroptotic hallmarks should be prevented in the k.o. Cathepsin L condition or use of the specific Cathepsin L inhibitor. Again, better inhibition of lysoptosis should be achieved as well.

5. The *in vivo* model used should underline the pro-survival role of Serpin3 in a disease model of NEC. The expertise of authors in the field of NEC was never questioned, rather the relevance for this model to support lysoptosis as an RCD. As the authors state correctly, NEC might involve different death pathways triggering the observed deadly phenotype. Hence, this model does not support the hypothesis of the paper that lysoptosis is an independent RCD. Rather it confirms the pro-survival role of Serpin3 in a mouse model – confirming earlier results obtained by the authors in *C. elegans* (Cliff et al., 2007).

Further authors assume that TLR4 signaling is involved preliminary in NEC. However, in Figure 9 authors demonstrate that TLR4 induced death signaling is only very minimal affected by E-64. This discrepancy should be explained.

Minor points:

- Check Sentence line 334 following; and spelling mistake
- Although Annexin positivity can occur in necroptosis or pyroptosis, zVAD can only prevent the PS externalization during apoptosis. Further since STS is an established apoptosis inducer, additional PS exposure by potential bystander necroptosis/pyroptosis should increase STS-apoptosis induced PS externalization and not replace it. Authors' argumentation does not explain the marked difference

between cell with active Caspase-3 and Annexin positivity.

- Caspase-3 blot in Fig. 5: Full arrowheads not full-length caspase-3
- Digitonin is not an LMP inducer as stated, but a detergent, permeabilizing at higher concentrations all cellular membranes

Reviewer #1 (Remarks to the Author):

The authors have addressed many of my concerns. However, the following minor comments need to be addressed:

1. Line 292-293. The inference for Fig S10B doesn't make sense. The % dead cells of HT B3WT and B3KO cells is comparable to each other and less than 20%. Please revise.

Thank you for pointing out this confusing statement. We corrected the manuscript.

2. Presentation of the following Figs should be improved for ease of reading:

• S13D - Please place WT and KO cells side by side for ease of comparison between 2 different treatments (0 vs 20 digitonin and 100% vs 10% DPBS) [similar to Fig 4B].

We changed the chart format per the reviewer's suggestion, it is now Fig. 5B.

• S13E, F- Please place WT and KO side by side for ease of comparison between different treatment 100% vs 10% DPBS (similar to Fig 4A).

We changed the chart format per the reviewer's suggestion, it is now Fig. 5C, D.

• S13B,C- please add an arrow similar to S12B-U showing the exact band size of the protein of interest.

We changed the Figs per the reviewer's suggestion, it is now Fig. 5A.

• S12B-U- Please define the meaning of black vs white arrows in the Fig. legend.

We clarified this issue in the Fig. legend

3. Line 202-203: the following statement 'Neither DMSO (diluent), E64d, nor DEVD-CHO had an effect on cell viability' was made without reference to any Fig. or data. In addition, there was no data demonstrating this inference in either the main or supplementary figures. Please revise.

We revised the statement to include a reference to Fig. 2Y. (Lines 206-207)

4. Line 402- based on Fig 6P the cell death in the HT3 B3Ko cells was partially rescued by the E64D; only 50% rescue of cell death in the presence of E64D in response to H2O2. Please revise the text.

There was no significant change in the cell death observed between HT3^{B3-WT} and HT3^{B3-KO} cell lines treated with H₂O₂ (Fig. 7P (previously Fig. 6P)). However, significant protection was observed between diluent control (DMSO) and E64d treatment in the HT3^{B3-KO} cell line. Thus, we feel our original statement of "In contrast, E64d failed to protect significantly HT3^{B3-WT}, whereas cell death was significantly decreased in HT3^{B3-KO} cells (Fig. 7P)." (Lines 420-422) is scientifically accurate. We refer to the editor's discretion on this matter.

5. Fig 8C: In the rebuttal authors provided the argument that ‘cell death induces cellular protein breakdown’, I agree. However, can the authors please explain why the beta-actin band in untreated (-) HT3 B3KO cells weaker than the HT B3WT control. The untreated cells are not expected to undergo cell death. Does it mean that at the start of experiment, the HT B3KO cells were unhealthy and undergoing cell death? Alternatively, was there uneven loading of protein lysate between the HT3 B3WT and HT3 B3KO cells and hence samples were run on two separate blots. Therefore, please provide the Sytox+ data for untreated HT3 B3WT and HT3 B3KO cells for Fig 8A. Please run the HT3 B3KO samples alongside the HT29 and HT3 B3WT samples on the same blot for all the proteins of interest.

Our initial explanation concerning actin degradation in dying cells was in reference to the apoptosis immunoblot (Fig. 6 (previously Fig. 5)). However, since there were diminished actin bands in HT3B3-WT untreated vs treated cells but not in HT3-B3KO samples, these differences can be explained by the expected variations in total protein loading when dealing with relatively small cell numbers.

6. Fig S17K- There seems to be a discrepancy in the % dead cells in HT3 B3WT vs HT3 B3KO in response to nigericin treatment compared to S17- J and A. Also, the % dead cells is just 40% compared to S17- J and A (~80%). In addition, data in S17- J and A suggest that B3KO cells underwent high % of dead cells compared to B3WT cells in response to nigericin. Thus, the experiment for S17K needs to be repeated.

These experiments were included to demonstrate that the HT3 cells lines were recalcitrant (at least in our hands) to the induction of pyroptosis via the canonical (LPS/nigericin) pathway, despite detecting significant non-gasdermin D-mediated cell death. These data were not used to support the hypothesis that lysoptosis was distinct from pyroptosis. This latter point was demonstrated in experiments represented in Fig. 10 using the non-canonical pathway (cytoplasmic LPS). We appreciate that there may be some confusion. However, these results provided important rationale for using the non-canonical pathway to study the distinction between pyroptosis and lysoptosis in HT3 cells.

7. Line 754- Please change P25 to P20 (casp1 cleavage).

We revised the text.

Reviewer #2 (Remarks to the Author):

All the reviewer comments have been sufficiently addressed. The authors have performed a set of new experiments and now the data strongly support the research hypothesis. The reviewer has no further comments.

Reviewer #3 (Remarks to the Author):

In this revised manuscript by Good et al, the authors have addressed some of my original concerns and additional experiments have been performed. However, the hypothesis of lysoptosis being an evolutionary conserved, independent regulated cell death mechanism still is not well supported. Although different cell death routines might be induced by a single death trigger, lysoptosis /LMP has to initiate all downstream death events. If cell death is not inhibited by interference with lysoptosis – like it is currently the case – lysoptosis is only a bystander of

death, but not the inducer. Authors claim Cathepsin L as the main executor of lysoptosis. This finding could give the authors the possibility to strengthen their findings (also from a mechanistically aspect) and validate their hypothesis. In addition, the in vivo model used, despite expert knowledge of the authors in NEC, does not support the hypothesis of lysoptosis being an independent RCD.

Major comments

1. Authors argue that E-64 cannot fully inhibit lysoptosis induced by STS, H₂O₂, TSQC and LPS (in this case the inhibitory effect is in particular minimal), as multiple cell death routines are induced in parallel. However, to claim that lysoptosis is an independent RCD, it is essential and crucial to demonstrate that lysoptosis is the initiator of death and not a bystander. Consequently inhibition of lysoptosis has to block all downstream death events (necroptosis, apoptosis, pyroptosis,.....).

*The pan-cysteine protease inhibitors E64 and E64d completely protected **SERPINB3 knockout cells** (i.e., there was no significant difference between B3-WT and B3-KO cell lines in the Figs) from cell death pathways where marked LMP was detected (apoptosis, Fig. 6), mTP-DN (Fig. 7), and necroptosis (Fig. 9); but not in those where LMP was minimal (ferroptosis (Fig. 8) and pyroptosis (Fig. 10)). We respectfully disagree with the premise that blocking one death pathway will completely abolish all other cell death pathways. As an example, look no further than classical necroptosis, which is induced when apoptosis is blocked by caspase 8 inhibition. As stated in the introduction and discussion, the role of cathepsins and LMP in other cell death pathways has been controversial, with published works indicating both positive and negative roles in multiple cell death pathways (apoptosis, necroptosis, ferroptosis, pyroptosis etc.). In part, this conundrum stems from the observation that inhibition of one cell death pathway in mammalian systems frequently activate secondary cell death routines (e.g., Panoptosis; Malireddi, et al., 2019 doi:10.3389/fcimb.2019.00406). For this reason, model organisms, such as *C. elegans*, with their more simplified physiology are a useful tool for identifying nascent components of biochemical pathways such as cell death. Indeed, the essential components of lysoptosis were revealed by taking advantage of nematode genetics and biochemistry, we suggest that it would be difficult to discern the presence of this pathway in higher vertebrates without the clues provided by simple model organisms.*

2. Authors claim Cathepsin L as the main executor of lysoptosis. However, active Cathepsin L is released from the lysosomes in both Serpin B wt and k.o. cells. Consequently, the conclusion that Cathepsin L executes lysoptosis cannot be drawn. To confirm the executor role of cathepsin L during lysoptosis experiments need to be repeated in the presence of a specific cathepsin L inhibitor. In addition, also the knock out cells for Cathepsin L should also be exposed to the other death triggers used in the paper.

The reviewer actually makes our point. Cathepsin L was released into the cytosol in both WT and SERPINB3 KO cells, but can only serve as an “executioner” if it is not neutralized by SERPINB3. In other words, lysoptosis appears to be initiated under many different forms of cell stress, but cathepsin L-induced proteolysis reaches a critical level only in cells lacking SERPINB3 or in those situations where the cathepsin activity exceeds the buffering capacity of cytosolic protease inhibitors. Specific data addressing this concern were included in the response to the editors (see page 3 , editor’s comment (1)).

3. Authors speculate that the death is based on the calpain-cathepsin hypothesis. If this is the mechanistical basis of lysoptosis triggered by all different death stimuli used (hypotonic stress, STS, H2O2, TSQC, LPS, Nigericin) then the author need to demonstrate initial calcium influx and calpain activity. The previous publication by Cliff et al, to which the author refer demonstrates calcium influx in nematode, but appears distinct to the death triggers used in this manuscript. In particular STS (Fig.5), caspase activity is not blocked by E-64. The author state correctly E-64 is also a calpain inhibitor. Again, a specific Cathepsin L inhibitor is the better choice.

We made reference to the calpain-cathepsin hypothesis in our previous work in C. elegans (Luke et al., 2007). However, we did not refer to this pathway by name in the current manuscript. Nor, was it our intent to determine whether this pathway was linked to lysoptosis in the current studies. We did refer to some of the active components (calcium, calpains and cathepsins) of this pathway in the discussion, but only to draw similarities and potential untested aspects of the lysoptosis pathway that appears to present in invertebrates, vertebrates and plants. We believe that points are relevant for the discussion and are not overtly speculative. We defer to the Editor's discretion on this point.

4. Necroptotic cocktail induces lysoptosis. To clearly demonstrate that LMP is the initial event leading to Cathepsin L release to execute the necroptosis like phenotype, necroptotic hallmarks should be prevented in the k.o. Cathepsin L condition or use of the specific Cathepsin L inhibitor. Again, better inhibition of lysoptosis should be achieved as well.

We did not suggest in any way that LMP or lysosomal cathepsins were required for necroptosis. We did show that a relatively limited amount of LMP occurred in necroptosis (consistent with the literature), but death was not blocked by E64, and therefore was inconsequential in mediating this form of cell death. However, in the absence of the serpin, "trivial" LMP appeared to rapidly progress to extensive LMP, with triggering of lysoptosis that superseded the necroptosis phenotype. Once again, these results suggested that, in the absence of the serpin, cells become primed to lysoptosis induction by multiple stressors, including those know to induce necroptosis.

5. The in vivo model used should underline the pro-survival role of Serpin3 in a disease model of NEC. The expertise of authors in the field of NEC was never questioned, rather the relevance for this model to support lysoptosis as an RCD. As the authors state correctly, NEC might involve different death pathways triggering the observed deadly phenotype. Hence, this model does not support the hypothesis of the paper that lysopotosis is an independent RCD. Rather it confirms the pro-survival role of Serpin3 in a mouse model – confirming earlier results obtained by the authors in C. elegans (Cliff et al., 2007).

Further authors assume that TLR4 signaling is involved preliminary in NEC. However, in Figure 9 authors demonstrate that TLR4 induced death signaling is only very minimal affected by E-64. This discrepancy should be explained.

We agree that the intestinal injury model did not provide proof that Serpinb3a was inhibiting lysoptosis in vivo. Thus, we will remove these data from the manuscript to provide space for the more essential cathepsin L knockout and inhibitor data. Moreover, in the companion paper (Squamous cell carcinoma antigen/SERPINB3 protects cervical cancer cells from chemoradiation by inhibiting cathepsin L and preventing lysoptosis), we show that

SERPINB3 protected cells from lysoptosis-like cell death after exposure to ionizing radiation. Therefore, we suggest that this in vivo data is more compelling than the mouse intestinal injury model initially submitted. The irradiation experiments are referred to in the discussion: "Moreover, as we show in a companion paper to this study, the HT3^{B3-KO} and SW756^{B3-KO} cell lines demonstrated enhanced sensitivity and a lysoptosis-like cell death phenotype in response to irradiation in vitro and in vivo." (Lines 730-732)

Minor points:

- **Check Sentence line 334 following; and spelling mistake**

Corrected.

- **Although Annexin positivity can occur in necroptosis or pyroptosis, zVAD can only prevent the PS externalization during apoptosis. Further since STS is an established apoptosis inducer, additional PS exposure by potential bystander necroptosis/pyroptosis should increase STS-apoptosis induced PS externalization and not replace it. Authors' argumentation does not explain the marked difference between cell with active Caspase-3 and Annexin positivity.**

A direct correlation between these experiments is not possible. The data in Fig. 6 (previously Fig. 5) represented the results from several different but reproducible experiments. Specifically, the caspase (Fig. 6A) and PI data (Fig. 6Q) were collected from the same set of cultures. In contrast, the annexin V data (Fig. 6O) represented the quantitative analyses of the images in Fig. 6C-N and was not optimized to coordinate with caspase activity. The purpose of the two datasets (Fig. 6A, Q and Fig. 6C-N, O, P) was to show that wild-type cells were capable of undergoing apoptosis. An early time course was selected to avoid the confounding effects of secondary necrosis (annexin V staining and lysosomal leakage), which would be impossible to differentiate from lysoptosis. The data showed molecular features of apoptosis: caspase 3/7 activity, annexin V staining and minimal LMP early in the time course of STS treatment of wild-type cells. These results contrasted markedly with the dominant lysoptosis phenotype in the SERPINB3-KO cells. We will amend the Fig. legend indicating that Fig. 6A and Q are from the same experiment and that Fig. 6 and P represent the quantitation of the images depicted in Fig. 6C-N.

- **Caspase-3 blot in Fig. 5: Full arrowheads not full-length caspase-3.**

The manuscript has been corrected (Fig. 6).

- **Digitonin is not an LMP inducer as stated, but a detergent, permeabilizing at higher concentrations all cellular membranes**

We modified the manuscript to reflect this point (Line 327).